# Telomere-to-telomere genome assembly of a male goat reveals variants associated with cashmere traits

Hui Wu[1,2,15], Ling-Yun Luo[1,15], Ya-Hui Zhang[1,15], Chong-Yan Zhang [3,15], Jia-Hui Huang[1], Dong-Xin Mo[1], Li-Ming Zhao[4], Zhi-Xin Wang[3], Yi-Chuan Wang[3], EEr He-Hua[5], Wen-Lin Bai[6], Di Han[7], Xing-Tang Dou[8], Yan-Ling Ren[9], Renqing Dingkao[10], Hai-Liang Chen[11], Yong Ye[12], Hai-Dong Du[12], Zhan-Qiang Zhao[12], Xi-Jun Wang[13], Shan-Gang Jia [14] ✉, Zhi-Hong Liu[3] ✉ & Meng-Hua Li [1] ✉

A complete goat (*Capra hircus*) reference genome enhances analyses of genetic variation, thus providing insights into domestication and selection in goats and related species. Here, we assemble a telomere-to-telomere (T2T) gap-free genome (2.86 Gb) from a cashmere goat (T2T-goat1.0), including a Y chromosome of 20.96 Mb. With a base accuracy of >99.999%, T2T-goat1.0 corrects numerous genome-wide structural and base errors in previous assemblies and adds 288.5 Mb of previously unresolved regions and 446 newly assembled genes to the reference genome. We sequence the genomes of five representative goat breeds for PacBio reads, and use T2T-goat1.0 as a reference to identify a total of 63,417 structural variations (SVs) with up to 4711 (7.42%) in the previously unresolved regions. T2T-goat1.0 was applied in population analyses of global wild and domestic goats, which revealed 32,419 SVs and 25,397,794 SNPs, including 870 SVs and 545,026 SNPs in the previously unresolved regions. Also, our analyses reveal a set of selective variants and genes associated with domestication (e.g., *NKG2D* and *ABCC4*) and cashmere traits (e.g., *ABCC4* and *ASIP*).

Goats (*Capra hircus*) were domesticated in the Fertile Crescent *ca.* 10,500 years before present (BP)[1]. Footprints of natural and artificial forces have shaped their genomes through the accumulation of genomic variants. A complete reference genome is crucial for improving the analysis of genomic variants involved in the genetics, selection and evolution of goats and related species[2–4]. With the advancement of sequencing technologies, several goat genome assemblies, for example CHIR_1.0, have been released, followed by updates to CHIR_2.0[5] (GenBank accession no. GCA_000317765.2), Saanen_v1[6] (GCA_015443085.1), and ARS1.2[7] (GCF_001704415.2). Nevertheless, in addition to unplaced contigs, the assemblies have unfilled gaps (e.g., 169 gaps in the most improved reference Saanen_v1

of a Saanen goat[6]) in the pseudochromosomes, many of which might cover the gene bodies and cause a loss of uncovered genes. Additionally, the complete sequences of many centromeres, telomeres, repetitive regions and Y chromosome have not been found in these published goat reference genomes, which limits our knowledge of their organization, structure, evolution, and function[8,9].

Advances in long-read sequencing technology, especially ultra-long ONT reads, have greatly facilitated the achievement of telomere-to-telomere (T2T) genome assemblies without any gaps, including those of human[10,11], Arabidopsis[12], watermelon[13], maize[14], soybean[15], rice[16], and chicken[17]. Complex genome regions of autosomes and chromosomes X and Y unlocked in the T2T reference genome of

human T2T-CHM13 have revealed quite a few rare variants and genes[18,19], providing insights into their maternal and paternal evolutionary patterns[20]. In goats, except for the metacentric Y chromosome, all the autosomes and the X chromosome are acrocentric, which impedes the complete assembly of centromeres and telomeres[6,7]. Acrocentric chromosomes are typically involved in the frequent genomic recombination[21]; thus, variations in the unresolved regions of acrocentric chromosomes are the key to understanding chromosomal evolution, genome rearrangement and selective signatures in goats and related species[22].

To date, T2T reference genomes of large vertebrates, in addition to those of humans, have been rarely reported with all complete chromosomes[11]. Here, we described the T2T gap-free genome assembly of the cashmere goat (T2T-goat1.0), which includes all autosomes and chromosomes X and Y, using ultralong ONT and PacBio HiFi reads together with Hi-C and Bionano data. We revealed the genomic organization, variations, and genes in previously unresolved regions, mostly for centromeres, telomeres, and the Y chromosome. We showed the advantages of using the T2T-goat1.0 genome as a reference for population genomics analyses, such as sequence alignment, variation calling, genetic structure and selective signal detection.

## Results

### T2T genome assembly

We selected a buck from the Inner Mongolia cashmere goat, a representative indigenous goat breed in China (Fig. 1a), for the T2T genome assembly. We generated a total of 328.44 Gb of ultralong ONT (114.8× coverage), 141.03 Gb of PacBio HiFi (49.3× coverage), 435.76 Gb of Hi-C

data (152.4× coverage), and 1188.9 Gb of Bionano optical mapping data (Supplementary Data 1). The initial goat genome assembly (GV1) was built based on PacBio HiFi data, and 97 contigs were scaffolded as the GV2 assembly with Bionano data (Supplementary Fig. 1). Using Hi-C data, the contigs/scaffolds were anchored and oriented onto 30 pseudochromosomes corresponding to 29 autosomes and the X chromosome of the GV3 assembly (Supplementary Fig. 2), with 145 gaps and an average of ~4.8 gaps per pseudochromosome (Supplementary Fig. 3). We named all the autosomes (CHI1 for *C. hircus* chromosome 1, and the other 28 chromosomes CHI2–CHI29) based on the sequence similarity to those of the NCBI goat genome reference ARS1 (GCF_001704415.1). Our assembly results showed that all the autosomes and the X chromosome are acrocentric (Supplementary Fig. 4a), causing great challenges in the complete chromosomal end assembly. For example, a tangle of centromeric regions among four chromosomes, CHI12, CHI13, CHI19 and CHI22, is shown in the assembly graph string (Fig. 1b), and our assembly resolved the centromeric regions for these four chromosomes, as shown for the completeness and genome features on CHI12 and CHI19 (Fig. 1c and Fig. 1d). Therefore, the gaps were enriched in centromeric regions (Supplementary Fig. 3a), with the longest gap (~4 Mb) located on the centromeric region of CHI14, which resulted from repetitive sequences. We employed ultralong ONT reads to fill these gaps via extension or local assembly, depending on the gap size and complexity. The filled gaps were confirmed for reliability based on the alignment of the HiFi and ONT long reads (Supplementary Fig. 3b), and the genome GV4 was created accordingly. To construct the telomeric regions accurately, HiFi reads containing telomeric repeat sequences were used to

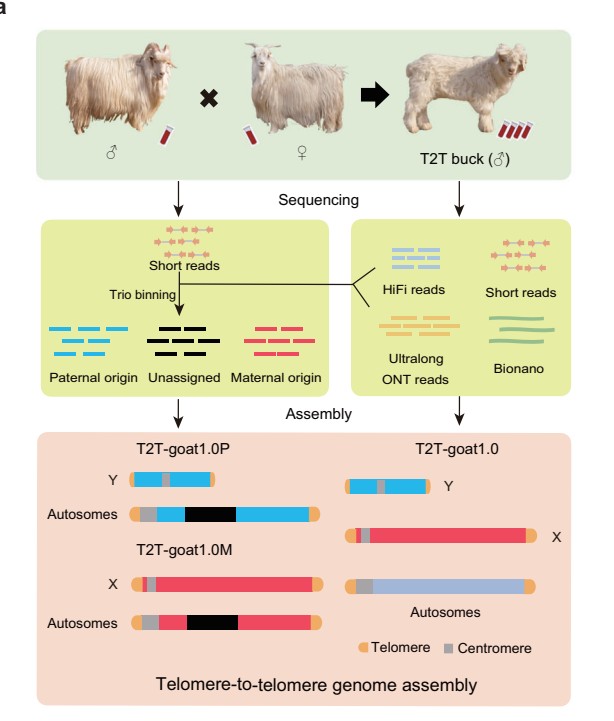

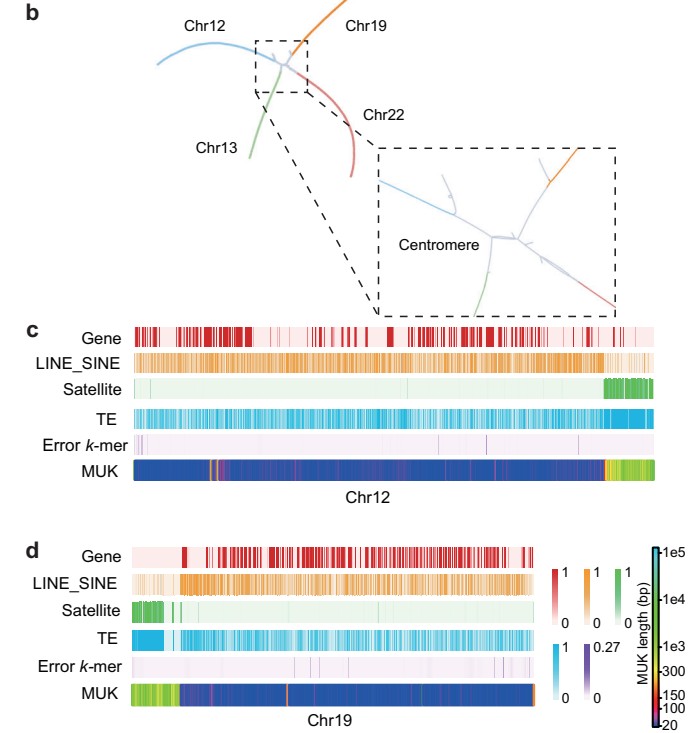

**Fig. 1 | Goat T2T genome assembly with 29 autosomes and chromosomes X and Y. a** Genome assembly strategy for T2T-goat1.0 and its haplotype genomes T2T-goat1.0P and T2T-goat1.0M of a buck. Trio-binning assemblies for the autosomes of T2T-goat1.0P and T2T-goat1.0M and Y chromosome were performed based on long reads of the buck and MGI short reads from its parents. **b** The assembly graph string shows a tangle among the four chromosomes, Chr12, Chr13, Chr19 and Chr22, due to the high similarities of centromeric sequences in gray. The centromeric regions in the assembly tangle are enlarged and shown in the right panel. **c, d** Genome features of Chr12 and Chr19 in T2T-goat1.0. The assembly graph tangle involving

Chr12, Chr13, Chr19 and Chr22 was resolved in T2T-goat1.0, and Chr12 and Chr19 are selected to exhibit the completeness and features across the whole chromosome. The following information is provided from top to bottom: the gene density (red), the density of LINEs and SINEs (orange), the satellite density (green), the TE density (blue), error *k*-mer (*k* = 21, purple), and the minimum unique *k*-mer (MUK) per 100 kb. The more MUK values indicate more repetitive sequences in a 100-kb window, and more yellow and green colors indicate the presence of the centromeric regions. All the features are shown in 10-kb windows, except for MUK.

**Table 1 | Summary of T2T-goat1.0, T2T-goat1.0P, T2T-goat1.0M, and ARS1 goat assemblies**

| Statistics | ARS1[a] | T2T-goat1.0 | T2T-goat1.0P | T2T-goat1.0M |
|---|---|---|---|---|
| Assembled bases (Gb) | 2.58 | 2.86 | 2.71 | 2.81 |
| Unplaced bases (Mb) | 340.66 | 0 | 0 | 0 |
| Number of gaps | 649 | 0 | 11 | 6 |
| Number of contigs | 30,378 | 31 | 41 | 36 |
| Contig N50 (Mb) | 26.24 | 100.79 | 103.46 | 109.45 |
| Protein coding genes | 20,542 | 20,953 | 19,720 | 20,530 |
| Percentage of segmental duplications (%) | 2.14 | 10.02 | 9.77 | 8.44 |
| Segmental duplication bases (Mb) | 55.25 | 286.7 | 264.78 | 236.67 |
| Number of segmental duplications | 4694 | 35,440 | 29,074 | 24,292 |
| Percentage of repeats (%) | 43.89 | 47.51 | 46.78 | 46.92 |
| Repeat bases (Mb) | 1133.22 | 1360.79 | 1267.73 | 1316.1 |
| Long interspersed nuclear elements (Mb) | 741.31 | 772.52 | 715.01 | 763.18 |
| Short interspersed nuclear elements (Mb) | 189.76 | 196.61 | 186.36 | 194.88 |
| Long terminal repeats (Mb) | 131.11 | 157.09 | 145.98 | 151.14 |
| Satellites (Mb) | 2.43 | 156.92 | 147.26 | 130.61 |
| DNA (Mb) | 68.08 | 73.5 | 69.3 | 72.58 |
| Simple repeats (Mb) | 1.61 | 2.46 | 2.18 | 2.22 |
| Low complexity (Mb) | 0.016 | 0.019 | 0.015 | 0.018 |

[a]The ARS1 genome files were downloaded from NCBI, and all the genomic features were calculated based on the same method as that for T2T-goat1.0.

independently assemble and replace the sequence errors at the ends of all chromosomes. The genome GV5 was generated, by combining with telomeric regions and Y chromosome (Supplementary Data 2). All the chromosomes in GV5 were polished using our local pipeline based on the ONT and PacBio HiFi long reads and the NGS short reads.

Based on long reads from the assembled buck and short read sequence data from its parents, the haplotype assemblies were generated by using Hifiasm[14] assembler based on the trio mode, for T2T-goat1.0P of paternal origin with complete Y chromosome and T2T-goat1.0M of maternal origin with complete X chromosome (Table 1). Parent-specific k-mers based on short reads were used to pick ONT and HiFi reads of paternal and maternal origin, to fill 31 gaps for T2T-goat1.0P and 25 gaps for T2T-goat1.0M. Y chromosome was also independently assembled with the trio-binned paternal ONT reads, and improved for scaffolding, gap filling and correction, with the assistance of Y chromosomal contigs de novo assembled from HiFi reads. As a result, we generated a complete Y chromosome with two telomeres spanning 20.96 Mb. The final T2T genome assembly, T2T-goat1.0, covering all the 29 autosomes and X and Y chromosomes, was obtained with a total length of 2.86 Gb (Table 1). This size was slightly longer than the estimated genome size of 2.76 Gb obtained using the NGS short reads. After final polishing, the chromosomes X and Y from T2T-goat1.0 were combined with the haplotype-resolved autosomes to obtain the paternal and maternal genome assemblies T2T-goat1.0P (2.71 Gb with N50 103.46 Mb) and T2T-goat1.0M (2.81 Gb with N50 109.45 Mb) respectively.

**Validation of the T2T goat genome assemblies**
Multiple strategies were further used to assess the integrity and accuracy of T2T-goat1.0. The genome-wide average consensus quality value (QV) was estimated to be 54.18, indicating a base accuracy >99.999% and an increase from 32.57 in the assembly GV5 (Supplementary Fig. 5). The homozygous single-base mutations due to the assembly errors were estimated based on alignment of short reads, and their rates decreased from 0.007888% in GV5 to 0.000174% in T2T-goat1.0. A completeness of 95.71% was obtained, and estimates of QV ranged from 50.07 to 59.31 across all the chromosomes, with 49.10 for the Y chromosome (Supplementary Data 3). The primary data, including HiFi and ONT long reads and short reads, were all mapped to

T2T-goat1.0, and the read coverage was found to be evenly distributed across nearly all the chromosomes with a few regions of abnormal read coverage for potential issues (Supplementary Fig. 6). Overall, 100% of the HiFi and 99.01% of the ultralong ONT reads could be aligned to T2T-goat1.0, and the unaligned long reads were found with highly repetitive sequences of two nucleotides [e.g., (AT)n], or with failed sequencing due to overloaded errors. In total, 99.92% of the short reads could be aligned to T2T-goat1.0, and the remaining unaligned short reads were confirmed to be bacterial contaminants or sequencing errors. The alignments of the Bionano optical maps also supported the accurate assembly of T2T-goat1.0 (Supplementary Fig. 7), and a few differences between Bionano optical maps and T2T-goat1.0 mostly fell into the centromeric regions.

Meanwhile, T2T-goat1.0P and T2T-goat1.0M achieved the T2T level assembly, with the whole genome QV 53.06 and 52.37 respectively. The challenges for assembly of acrocentric centromeres and lack of binned long reads still left 11 gaps in the centromeric regions of 9 chromosomes in T2T-goat1.0P and 6 gaps in the centromeric regions of 6 chromosomes in T2T-goat1.0M (Supplementary Data 4). Except for the centromeric regions, the genomes have relatively uniform HiFi and ONT coverage. We estimated the switching errors and obtained 0.036%, 0.016%, and 0.011% for the three T2T assemblies T2T-goat1.0, T2T-goat1.0P, and T2T-goat1.0M respectively. Furthermore, by comparing autosomes between the two haplotypes, we detected 5,287,863 single nucleotide variants (SNVs), 726,939 small insertions and deletions (<50 bp), and 22,689 SVs (≥ 50 bp). Considering the completeness and quality, T2T-goat1.0 represented a better assembly (Supplementary Fig. 4a) for the downstream analysis compared to the two haplotypes. T2T-goat1.0 also represented the longest gap-free chromosome-level genome in all the available goat genome assemblies, only two of which included incomplete Y chromosomes. The recent goat assemblies (e.g., GCF_001704415.1, GCA_015443085.1 and GCA_026652205.1) harbored 136 to 649 gaps, 231.45 to 287.20 Mb less in length than the T2T-goat1.0 assembled here.

**Assembly improvement of T2T-goat1.0**
A high degree of collinearity was observed between T2T-goat1.0 and the NCBI goat genome reference ARS1, indicating the high quality and reliability of T2T-goat1.0 (Fig. 2a). The assembly T2T-goat1.0 closed all

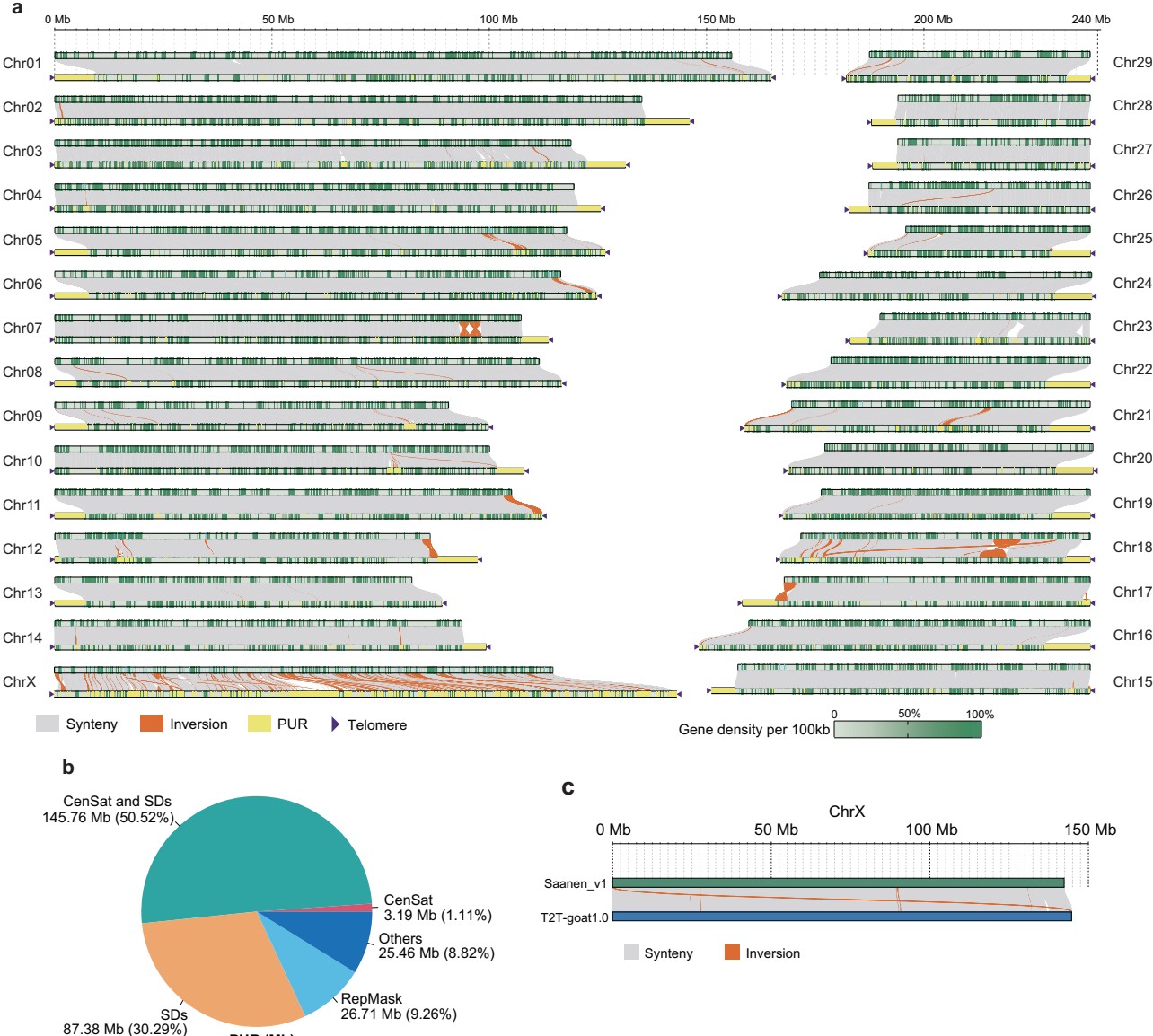

**Fig. 2 | Synteny and improvement of the T2T-goat1.0. a** Syntenic and nonsyntenic regions between ARS1 (top) without telomeres and T2T-goat1.0 (bottom) with telomeres that are indicated by dark purple triangles. The collinearity between the two genome assemblies is shown as gray lines or blocks, and the inversions are shown in orange. The yellow bars represent the previously unresolved regions (PURs) in T2T-goat1.0. The gene density in 100-kb windows is shown as dark green bars. **b** The proportions of various repetitive elements in PURs. CenSat, satellite sequences in the centromeric region; SDs, segmental duplications; RepMask, repeats by RepeatMasker. **c** Syntenic and nonsyntenic regions of the X chromosome (ChrX) between Saanen_v1.0 (NCBI accession no. GCA_015443085.1) and T2T-goat1.0. Synteny and inversion are shown in gray and orange respectively.

the 649 gaps in ARS1 and resolved the telomeric regions. Previously unresolved regions (PURs) were identified by comparing all the chromosomes of T2T-goat1.0 and ARS1[11] and inferred to the misassembled or newly assembled regions, for a total size of 288.5 Mb. More than 30 Mb of PURs were found on the X chromosome. These PURs consisted mostly of centromeric satellites and segmental duplications (SDs) which were discovered according to the previous method[23] and accounted for 81.92% (236.33 Mb) (Fig. 2b). Additionally, we detected an overall size of 157.38 Mb for the newly assembled regions (NARs) as a subset of PURs that are not included in all the chromosomes and unplaced contigs of ARS1, and found that most NARs were in centromeric regions. In contrast to the genome-wide average QV of 40.93 and 32.83 for chromosomes and all the sequences (including chromosomes and unplaced contigs) in ARS1 respectively, the QV estimate was greatly enhanced for T2T-goat1.0, for which the genome-wide

average QV was 54.19 and the maximum average QV of each single chromosome was 59.31 for CHI10. To assess the increase of unique sequences in T2T-goat1.0, we calculated the minimum unique $k$-mer (MUK) length, which was defined as a $k$-mer presenting the minimum distance in a window of 100 kb needed to identify a unique sequence in the genome[18], and longer MUK length (up to 100 kb) was obtained if more repetitive sequences found in a 100-kb window. We observed a substantial increase in the number of unique sequences in T2T-goat1.0 compared to that in ARS1 (Supplementary Fig. 8). Longer MUKs in 100-kb windows across the chromosomes were observed in the centromeric regions due to enriched repetitive sequences (Fig. 1c, d).

T2T-goat1.0 corrected abundant structural errors in ARS1 (Fig. 2a), such as, an inversion on the q arm of CHI18 in ARS1 compared to T2T-goat1.0 (CHI18: 19 Mb–26 Mb). We examined the alignments of the long read sequence data from ARS1 goat individual (NCBI

BioSample ID SAMN03863711) against the inversion regions of the two genome assemblies. We found that the two junction sites of this inversion in ARS1 were not covered by any long reads sequenced from both ARS1 and T2T-goat1.0, but the failed alignment of these reads at the junction sites is rescued by the corresponding region of T2T-goat1.0 (Supplementary Fig. 9). The assembly quality and reliability of this region were supported by the Bionano evidence. The accurate assembly of the X chromosome is further supported by read alignments (Supplementary Fig. 6) and Bionano maps (Supplementary Fig. 7). T2T-goat1.0 filled numerous gaps and corrected multiple structural errors in the highly fragmented sequences on the X chromosome of ARS1 (Fig. 2a). These structural errors on the X chromosome in ARS1 were not found in Saanen_v1 (CM027079.1), and the evaluation of collinearity between T2T-goat1.0 and Saanen_v1 demonstrated only 8 inversions (Fig. 2c). In addition, our T2T-goat1.0 had 4.62 Mb and 31.68 Mb of PURs added to the X chromosomes of the Saanen_v1 and ARS1 assemblies, respectively.

We investigated the reference bias of read alignment for population genetic analyses. Short read data from previous sequencing efforts focused on global populations of domestic and wild goats was retrieved from public databases (Supplementary Data 5). Compared to ARS1, T2T-goat1.0 enabled confident mapping of short reads in terms of more reads mapped and lower mismatch and error rates (Supplementary Fig. 10a). T2T-goat1.0 improved the number of mapped reads substantially, and >30.91% of the goat samples had >1% more short reads aligned (19.04% with >5% more reads and 1.98% with >10% more reads) than to ARS1. We observed an increased number of MQ0 reads in T2T-goat1.0, which could be attributed to the fact that most of the newly added regions are composed of repetitive sequences, and short reads are not advantageous in aligning to repetitive sequence positions due to their short lengths. Short reads from the Middle East and East Asia populations showed significantly lower numbers of mismatches and fewer errors, benefiting from their relatively close genetic relationships with the Inner Mongolia cashmere goat[2], the goat population with its T2T genome assembled here (Supplementary Fig. 10). Moreover, we sequenced the genomes of five goats from five representative populations to produce long reads (Supplementary Data 1). Like for short reads, more aligned reads and lower mismatch and error rates were observed in the alignment of the long reads against T2T-goat1.0 than against ARS1 (Supplementary Fig. 10b and Supplementary Data 6).

## Genome annotation

Annotation of repeat content based on a combination of prediction and evidence-based tools revealed 47.51% repetitive elements, including 26.97% long interspersed nuclear elements (LINEs), 6.87% short interspersed nuclear elements (SINEs), 5.48% long terminal repeats (LTRs) and 5.48% satellites (Fig. 1c, d, Supplementary Fig. 11, and Table 1). Enrichment of satellites was present in the centromeric region, with rare satellite repeats found on the Y chromosome, although LINEs and SINEs were widely distributed on each chromosome (Supplementary Fig. 11 and Supplementary Fig. 12). The repetitive sequences of centromeric satellites, SDs, and repeats identified by RepeatMasker software dominated the PURs, with a total length of 263.04 Mb (91.18%) (Fig. 2b). Overall, the repetitive sequences in PURs made up 12.91% of the entire genome's repetitive sequences and occupied 94.92% of all the satellite sequences. LTRs were found to be distributed across all chromosomes, preferably with peaks around centromeric regions enriched with satellites (Supplementary Fig. 12).

A combined strategy of de novo, transcriptome-based and homolog-based prediction was used to perform gene annotation. A total of 20,953 high-confidence protein-coding genes were obtained (Table 1) with the average gene length of 46.97 kb, compared to 44.53 kb in ARS1, and 97.90% of the Benchmarking Universal Single-Copy Orthologs (BUSCO)[24] were annotated. Among these protein-coding genes, 1126 were newly anchored genes in PURs, and 446 were newly assembled genes (NAGs) in NARs that were not included in ARS1. RNA-seq of 14 goat tissues (Supplementary Data 1) supported the expression of these NAGs, which were diversely expressed in tissues of the liver, spleen, rumen, etc. (Supplementary Fig. 13). A tangle of the assembly string graph located far from the centromeric end was found to be associated with 45S rDNA repetitive sequences, comprising of 18S, 5.8S, and 28S on the distal regions of q arm of five chromosomes, CHI2, CHI3, CHI4, CHI5, and CHI28 (Supplementary Fig. 4a), as the reference point for distal and proximal regions is the centromere. The locations of 45S rDNA were further confirmed by fluorescence in situ hybridization (FISH) signals on five pairs of chromosomes (Supplementary Fig. 4b). The 5S rDNA was only found on the proximal regions of CHI28, opposite to the 45S rDNA end, as shown in both the assembly and FISH (Supplementary Fig. 4b). The results revealed a pattern of telomere (T) - rDNA (R) - centromere (C) - telomere (T) (TR-CT), with centromere and rDNA located on opposite chromosomal ends (Supplementary Fig. 4c).

We calculated the sequence length (>1 kb) and alignment identity (>90% identity) to determine the SDs of T2T-goat1.0. We identified a total of 286.70 Mb nonredundant SDs in T2T-goat1.0, with 73.28% (210.12 Mb) located in the centromeric region, while only 55.25 Mb was found on the chromosomes of ARS1. T2T-goat1.0 showed a remarkable increase in the number and length (additional sequences of 212.18 Mb) of interchromosomal SD pairs compared to those of ARS1. Out of a total of 62,776 SDs, we confirmed the presence and reliability of 59,503 (94.79%) SDs that are covered by at least one ONT read. Next, we examined the remaining 3273 SDs without ONT read covering for the reliability based on the coverage, and we found that only 155 SDs accounting for 0.24% of the total number of SDs showed the ambiguity due to less than half of the average coverage across the whole genome, with 151 ones located in the centromeric regions (Supplementary Data 7). Overall, 275 genes in 118 families overlapped the regions with SDs, indicating a critical contribution of SDs to the expanded gene families. Furthermore, we observed significantly greater gene copy numbers in T2T-goat1.0 than in ARS1 and the goat assembly ASM2665220v1 (GCA_026652205.1), which could be attributed to the successful assembly of repetitive sequences and genome annotation improvements in T2T-goat1.0 (Supplementary Data 8). For example, in the gene family associated with scavenger receptor activity (OG0000004), 15 gene copies were annotated on CHI5 in T2T-goat1.0 instead of only 8 in ARS1, and the assembly of the corresponding ~3 Mb region was further confirmed for its continuity with evidence of long read alignment and collinearity between T2T-goat1.0P and T2T-goat1.0M (Supplementary Fig. 14).

## Chromosomal locations of centromeres and telomeres

We focused on deciphering the centromeric structure and repetitive structure in the unique acrocentric chromosomes (autosomes and X chromosome) and metacentric Y chromosome. First, we determined the centromeric regions based on the enriched regions of the mapped ChIP-seq reads with the histone antibody Phospho-CENP-A (Ser7) and the enriched methylated cytosine sites based on the HiFi and ONT data (Fig. 3a). Furthermore, the centromeric regions on the acrocentric autosomes were confirmed by a combination of ChIP-seq peaks, DNA hypermethylation, and enriched satellite DNAs (Supplementary Fig. 15a). To place centromeres on the leftmost ends of acrocentric autosomes, we transformed T2T-goat1.0 into T2T-goat2.0 which is also available together with the haplotype genomes T2T-goat2.0P and T2T-goat2.0M in NCBI for convenient usage. The lengths of the centromeric regions ranged from 0.6 Mb to 8.51 Mb (Fig. 3b), with a short centromere (0.6 Mb) on the X chromosome. Repeat annotation of centromeric regions revealed the dominance of satellite DNAs, and to identify high order repeats (HORs), 21 centromeric satellite sequences were identified using the SRF software[25]. Pairwise comparisons

classified the 21 repetitive sequences into three categories of centromeric satellite units, i.e., SatI (816 bp), SatII (702 bp), and SatIII (22 bp). Additionally, an invasion of other repeats, including LINE and SINE, was detected in the centromeric regions; notably, a LTR (4852 bp) was distributed across nearly all the centromeres, with an accumulative length of 1.36 Mb.

By employing a sequence query based on the telomeric repeat (TTAGGG at the 5′ end and CCCTAA at the 3′ end), we uncovered 62 telomeres in the assembly spanning 0.71 Mb. This led to the creation of 31 telomere-to-telomere chromosomes. The length of the identified telomeres ranged from 2993 to 26,638 nucleotides across all chromosomes, with the longest telomere located at the q arm of CHI3 (Supplementary Fig. 16).

## Centromeric satellites

Satellite compositional bias was detected across the chromosomes (Fig. 3b). The most abundant centromeric satellite repeat, SatI, was found in the centromeric regions of all autosomes and the X chromosome (Fig. 3b) and had an accumulative length of 118.19 Mb, covering approximately 67.81% of their centromeric regions. SatI was previously identified to be a constant length of 816 bp (X57335.1) and could be mapped well to the centromeric regions of T2T-goat1.0. SatII accounted for 15.21% of the total satellite sequences in T2T-goat1.0, with a total length of 26.50 Mb. Notably, the genomic locations of SatII were always associated with ChIP-seq peaks in the centromeric regions. For example, SatII was found to cooccur with ChIP-seq peaks on CHI22, CHI28, and the X chromosome. However, a specific link between SatII and ChIP-seq was not implied because ChIP-seq peaks could also be found in centromeric regions containing SatI, for example, on CHI20 and CHI21. We observed a high consistency on the occurrence of SatII across all the chromosomes except the Y chromosome in both FISH and the genome assembly (Fig. 3c). In contrast to the most abundant SatI and SatII, SatIII was not typically present in the core centromeric region but was present within the centromeric near-end regions, suggesting the evolutionary history of centromeric regions. SatIII contains two major variants differing in two nucleotides, which showed the intensified FISH signals of probe binding on different chromosomes (Fig. 3c and Supplementary Fig. 15b). We confirmed the conservation of centromeric satellite sequences between sheep and goats since the ovine centromeric SatI (KM272303.1, 816 bp[26]) and SatII (U24092.1, 440 bp[27]) sequences could be mapped well to the T2T-goat1.0 due to their high similarity.

To further understand the centromeric evolution on each chromosome, we generated pairwise sequence identity heatmaps within and around centromeric regions, which included distinct classes of HORs related to SatI, SatII, and SatIII. Compared to those of SatI, higher identity values for SatII and SatIII were observed, indicating closer similarities and fewer variations in the repeat sequences possibly associated with their recent origins (Supplementary Fig. 15a). This observation suggested centromeric reorganization of HORs, which was in accordance with the results of sequence complexity analysis (Entropy, Fig. 3a). For example, we found three major evolutionary layers corresponding to SatI, SatII and SatIII in the sequence identity heatmap of centromeric region of CHI1, and minor footprints of SatI and SatII around the SatIII layer (Fig. 3a).

## Y-chromosome assembly and structure

We used paternal-specific ONT ultralong reads (2.26 Gb, >100× coverage) to assemble the Y chromosome. The complete Y chromosome (T2T-CHIY1.0), with a length of 20.96 Mb, presented an overall QV of up to 49.10 after polishing, and the mapped HiFi and ONT reads showed evenly distributed coverage (Supplementary Fig. 6). The two available Y chromosome assemblies of Saanen_v1[6] (CM027080.1) and ARS1-based scaffolded contigs[28] (GCA_018357025.1) are only 9.6 Mb and 11.5 Mb in length, respectively, and were used to evaluate T2T-

CHIY1.0 quality. According to the collinearity plots, the pseudoautosomal regions (PARs) of ~7 Mb that are short regions of homology between the p arms of chromosomes X and Y were not assembled in these two incomplete Y chromosome assemblies (Supplementary Fig. 17). The SD-enriched region on the T2T-CHIY1.0 coordinates of 13–19 Mb (Fig. 4) resulted in poor collinearity, referring to low-identity organization with multiple contigs in the Y chromosome of ARS1 and abundant inversions in the Y chromosome of Saanen_v1 (Supplementary Fig. 17a and Supplementary Fig. 17b). We found 24 gaps in this region with poor-collinearity on Y chromosome of Saanen_v1, which could be ascribed to the presence of gene arrays as shown in T2T-CHIY1.0.

T2T-CHIY1.0 unlocked the PURs of 7.68 Mb compared to the Saanen_v1 Y chromosome, which mostly fell into the PARs shared by chromosomes X and Y (Supplementary Fig. 17c). By annotating the repetitive elements on the Y chromosome, we found that the repetitive content reached 53.79%, while the repetitive content on autosomes was only 45%. The most abundant repetitive sequences were LINEs (7.42 Mb in length), which accounted for 35.5% of the entire repetitive sequence, followed by the LTR elements (1.82 Mb, 8.70%) and SINE elements (1.42 Mb, 6.79%). Furthermore, LINE and SINE elements are more densely distributed in the PARs of T2T-CHIY1.0.

We annotated 134 protein-coding genes and 68 pseudogenes that were diversely distributed across the whole Y chromosome, of which 54 genes were newly found in the PURs of T2T-CHIY1.0 (Fig. 4). The annotation of 79 genes revealed two or more copies in the ampliconic regions on T2T-CHIY1.0 (Supplementary Data 9), including 27 gene copies of testis-specific protein Y-encoded (TSPY) and 14 copies of heat shock transcription factor Y-linked (HSFY) (Fig. 4), and the gene copies of TSPY and HSFY were validated by using ddPCR (Supplementary Data 10). Moreover, multiple copies of TSPY and HSFY were detected in ARS1. The TSPY genes on the Y chromosome were expressed only in the testis and were clustered in the region with enriched SDs, which was confirmed and supported by the sequence identity heatmap (Fig. 4). We constructed a phylogenetic tree of TSPY genes using human TSPY genes as the outgroup (Supplementary Fig. 18). The results revealed that the goat TSPY family could be divided into two clades. Similar to TSPY, the repeat array of HSFY was located in the SD-enriched region (Fig. 4). Furthermore, we annotated single-copy genes, including Y-chromosome-specific genes such as SRY, UTY, DDX3Y, and USP9Y.

Based on the HiFi and ONT data, hypermethylation in blood was identified on the PARs of T2T-CHIY1.0, and does not indicate the centromeric location on T2T-CHIY1.0. This suggested that the genes in this region were preferentially not expressed in blood (Fig. 4) due to transcriptional inhibition caused by hypermethylation of the genes' bodies and promoters. As centromeric repeats (SatI, SatII, and SatIII) and ChIP-seq signals based on Phospho-CENP-A (Ser7) antibody were not found on T2T-CHIY1.0, we extended our search for repeat sequences by reference to repeat sequences found on the sheep Y chromosome in an assembly of a sheep genome. A repeat sequence of 1474 bp was inferred as the putative centromere-specific repeat unit, CenY, since it was highly homologous to another CenY (2516 bp) in the centromeric region on the Y chromosome of T2T-sheep1.0[29] (T2T genome assembly for a ram of the Hu sheep breed, NCBI BioProject accession number PRJNA1033229). CenY was enriched in the middle of the Y chromosome, which was in accordance with the karyotype results[30] and indicated by a single sharp signal in the sequence identity heatmap, although no significant ChIP-seq peaks were found in T2T-CHIY1.0. CenY was not present in any of the autosomes or the X chromosome as shown in sequence search of T2T-goat1.0 and FISH results (Fig. 3c). We tested the performance of T2T-CHIY1.0 as a reference to call variants for a population of 102 bucks (Supplementary Data 5), which yielded a mapping rate as high as 99.95%. The bucks and does showed the significant differences on the mapped short reads

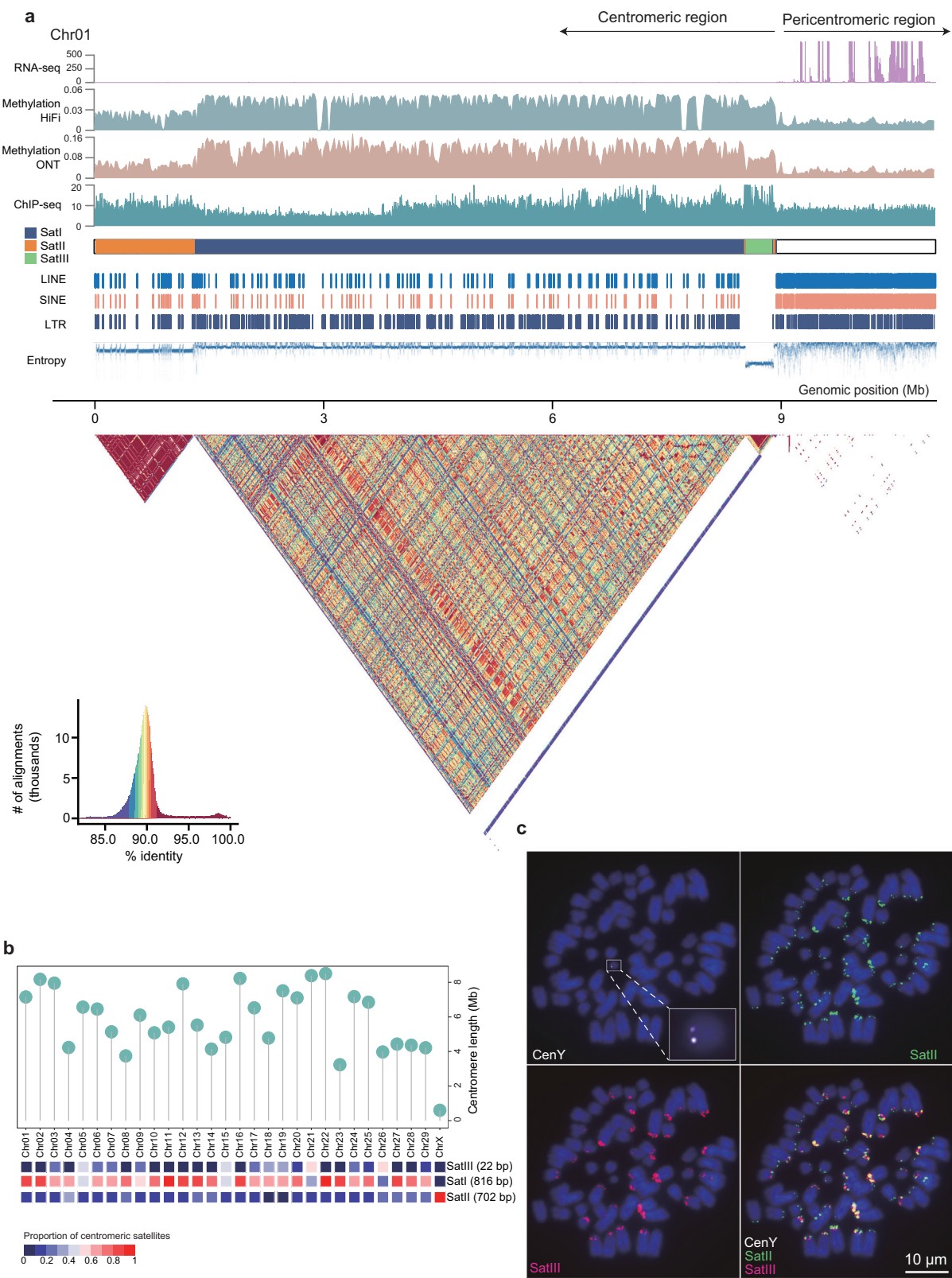

**b**

**c**

onto the male-specific region of the Y chromosome (MSY), with the average depth of aligned reads for buck individuals vs. only a few misaligned reads from doe individuals.

**Structural variants based on long reads**

To assess the advance of T2T-goat1.0 as a reference for long-read alignment and calling of large-scale structural variations (SVs), we used

the PacBio Sequel II platform to sequence the genomes of five female goats of five representative breeds with diverse origins (Supplementary Data 1 and Supplementary Fig. 19a). We focused on autosomes and chromosome X in female goats for SV calling due to no Y chromosome included in ARS1. After quality control, we aligned the long reads from these five individuals to both T2T-goat1.0 and ARS1 and called SVs for comparison between the two assemblies. We identified a total of

**Fig. 3 | Genomic structure of centromeric regions. a** Schematic representation showing the sequence compositions in the centromeric region of chromosome 1 (Chr01). From top to bottom: RNA-seq, expression (read counts in 1-kb windows) of genes in the pericentromeric region as indicated by an arrow on the top; Methylation by HiFi and ONT in 20-kb windows; ChIP-seq for CENP-A protein enrichment based on Phospho-CENP-A (Ser7) antibodies in 5-kb windows; Centromeric satellites of SatI (blue), SatII (orange) and SatIII (green); the occurrence of LINE, SINE, and LTR; and Entropy that was calculated for sequence complexity with low values for centromeric regions. The pairwise 10-kb sequence identity (%) heatmap in

centromeric region is shown below, with color key in the bottom left corner. The boundary between centromeric and pericentromeric regions is close to the coordinate 9 Mb on Chr01. **b** Length distribution of centromeric regions (top) and proportions of three centromeric satellite units (bottom) on all the autosomes and the X chromosome. **c**, FISH results for probes of CenY (white), SatII (green) and SatIII (red). CenY is shown uniquely on the Y chromosome, with experimental replicates ($n = 5$). The enlarged image of CenY probe (white) and its unique binding to the Y chromosome is shown in a small box in white line in the top left panel.

63,417 SVs in the PacBio long reads of the five individuals (27,163 – 29,561) using T2T-goat1.0 as a reference (Supplementary Data 11). The number of SVs increased approximately 1.25–8.89% compared with that of ARS1. The increase in SVs was attributed mostly to deletions and insertions rather than the inversions and translocations. Compared with that of ARS1, the use of T2T-goat1.0 as a reference genome led to the identification of slightly more deletions than insertions. The lengths of the deletions and insertions exhibited a distribution peak of 1–2 kb (Supplementary Fig. 19b and Supplementary Fig. 19c), which is longer than that of 200–300 bp in the human T2T-CHM13 genome[18]. The improvement in the detection of deletions and insertions could be attributed to the accurate assembly of repetitive sequences in T2T-goat1.0. We discovered a substantial increase in deletions and insertions within the satellite sequences, with 359 by ARS1 versus more than 9000 by T2T-goat1.0, compared to the other repetitive sequences, such as LINEs, SINEs, or LTRs (Supplementary Fig. 19d and Supplementary Fig. 19e). The improvement in SV detection resided not only in the quantity of deletions and insertions on satellites but also in the length frequency shift, with peak lengths of 751–1000 bp in T2T-goat1.0, compared to 50–100 bp in ARS1.

The accurate assembly and completeness of the PURs in T2T-goat1.0 enabled us to resolve complex genomic regions and enhance SV calling (Supplementary Fig. 20). Among the total SVs, up to 4711 SVs (7.42%) were identified to cluster densely in the PURs, particularly in the centromeric regions, consisting of 3021 deletions and 1690 insertions. We found that 18,804 SVs fell into the gene bodies, including 507 SVs overlapping exons. We further examined the 57 SVs that uniquely occurred in T2T-goat1.0 but were not in the other five goats, and found that *FGFBP1*[31,32], *KAP9-2*[33] and *COL1A1*[34] are known to be associated with hair and cashmere development (Supplementary Data 12). A total of 5224–6584 SVs were uniquely identified in the five female goats, and overlapped exons in 40–49 genes (Supplementary Data 13 and Supplementary Fig. 19f), which are frequently involved in olfactory receptors, interferon, immune system, and cashmere development. For example, mucin (MUC) family genes on CHI29, including *MUC2*, *MUC5B* and *MUC6*, experienced independent deletions in the four goats except for Tibetan goat. Two insertions in *TNNI1* gene were found in Tibetan goat, as *TNNI1* is involved in the calcium-sensitive regulation of skeletal muscle contraction, and associated with oxygen level and aerobic exercise[35].

**Variants based on short reads**
To investigate the impact of T2T-goat1.0 on short-read variant calling, we collected NGS data from 516 caprine samples (~15x on average), which consisted of 68 wild and 448 domesticated goats from five major geographic regions worldwide (Supplementary Data 5 and Fig. 5a). Clean short reads of the genomes were aligned to both T2T-goat1.0 and ARS1 for single nucleotide polymorphism (SNP) calling. After population-wide SNP calling and filtering for quality control, we obtained a total of 25,397,794 high-quality SNPs across the samples against T2T-goat1.0, whereas 24,238,138 SNPs were yielded when mapping to ARS1. T2T-goat1.0 added various high-quality SNPs compared to those of ARS1, ranging from 6405 to 67,614 on each chromosome. For example, the number of SNPs on the X

chromosome increased from 861,416 to 977,085 (Supplementary Fig. 21a), which could have resulted from the base-level errors in ARS1. Subsequently, we identified a total of 545,026 SNPs within the PURs (Supplementary Fig. 21b), which accounted for 2.1% of the total SNPs called using T2T-goat1.0 as a reference. We observed the most SNPs in wild goats, and compared to that with ARS1 as a reference, we identified an increase of total SNPs in all populations, including homozygous and heterozygous SNPs, except for East Asia (Supplementary Fig. 21c).

We identified a total of 32,419 SVs based on short reads, including 23,466 deletions, 2284 duplications, 1552 inversions, and 5117 translocations, among which 870 SVs were found in the PURs. Overall, 65.84% of these SVs were in intergenic regions, and the others within or near genic regions included 4.89% in exonic regions, 27.56% in intronic regions, and 1.58% upstream or downstream of genes.

**Domestication selection with new variants**
SNPs by T2T-goat1.0 were reliable for elucidating phylogenetic relationships within the genus Capra, with similar results obtained based on ARS1. Wild and domestic goats showed different linkage disequilibrium (LD) decay rates (Supplementary Fig. 22), and the analysis of neighbor-joining (NJ) tree, principal component analysis (PCA) and ADMIXTURE distinguished five subgroups in domesticated goats based on the geographical origin, namely, Europe, Africa, East Asia, South Asia, and the Middle East (Supplementary Fig. 22).

In a comparison of domesticated and wild goats, we evaluated the performance of T2T-goat1.0 to determine the domestication selection. With T2T-goat1.0 as the reference genome, the genetic differentiation statistic $F_{ST}$ values in 50-kb windows were calculated based on SNPs as an indicator of domestication signals between the genomes of bezoars (wild goats, *Capra aegagrus*) and the whole domestic goat population (Fig. 5b). Selective sweeps harbored 2631 genomic regions, covering 829 genes (Supplementary Data 14). The selective regions associated with domestication spanned 49.35 Mb in T2T-goat1.0, with 39.42 Mb found in common with ARS1 (Fig. 5c and Supplementary Fig. 23a). Among them, the signals bore previously reported genes[2], including *STIM1*, *RRM1*, *MUC6*, *IFNAR1*, *FREM3*, and *CD84*, which underwent strong selection during the domestication process of goats. We narrowed our search to 406 selective signals located in the PURs, spanning 5.93 Mb, and identified 56 genes resulting from our complete assembly of T2T-goat1.0. For example, a selective signal of the top 1% on CHI9 ($F_{ST} = 0.67$) encompassed the tandem genes of *NKG2D*. It serves as an activating receptor and regulator of immune cell responsiveness that can be induced in response to infection by diverse types of pathogens[36]. The presence of selective signal was supported by overlapping signals by the π ratio statistic (Fig. 5d). The 13 SNP loci associated with this selective signal showed significant allelic frequency differences between bezoars and domesticated goat populations (Supplementary Fig. 24a). Accurate assembly around this selective signal resulted in 18 tandem *NKG2D* genes on CHI9 in T2T-goat1.0, while the collinearity analysis revealed a loss of large fragments (Fig. 5e) and only 10 *NKG2D* genes located on CHI9 and three other unplaced contigs (Fig. 5f) in ARS1. The tandem *NKG2D* genes on CHI9 were in accordance with the SDs in this region, and exhibited

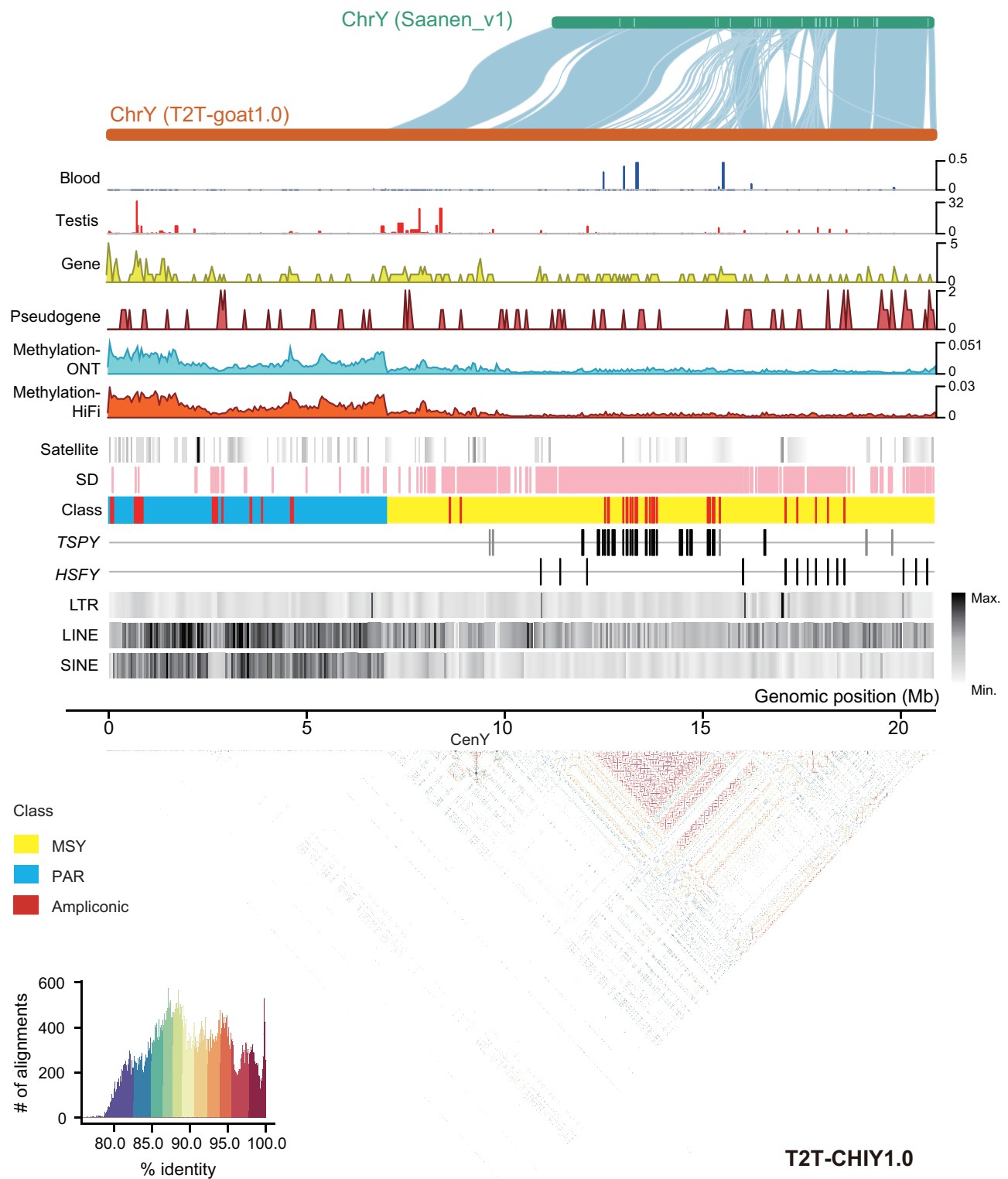

**Fig. 4 | Genomic structure of the Y chromosome. a** Genomic structure and features of the Y chromosome T2T-CHIY1.0. From top to bottom: collinearity of Y chromosomes between Saanen_v1.0 and T2T-goat1.0; Gene expression in blood and testis tissues; Protein-coding gene density; Pseudogene density; Methylation (5mC) levels estimated with ONT and HiFi reads; Satellite; Segmental duplication (SD); Class, the pseudoautosomal region (PAR) (blue), male-specific region of the Y chromosome (MSY, yellow), and ampliconic (red) regions on T2T-CHIY1.0; *TSPY* genes highlighted for Clade I in black and Clade II in gray; *HSFY* genes; LTR density; LINE density; and SINE density. The density of methylation, Satellite, LINE, SINE, and LTR is shown in 50-kb windows. The pairwise 10-kb sequence identity (%) heatmap across T2T-CHIY1.0 is shown below, with color key in the bottom left corner.

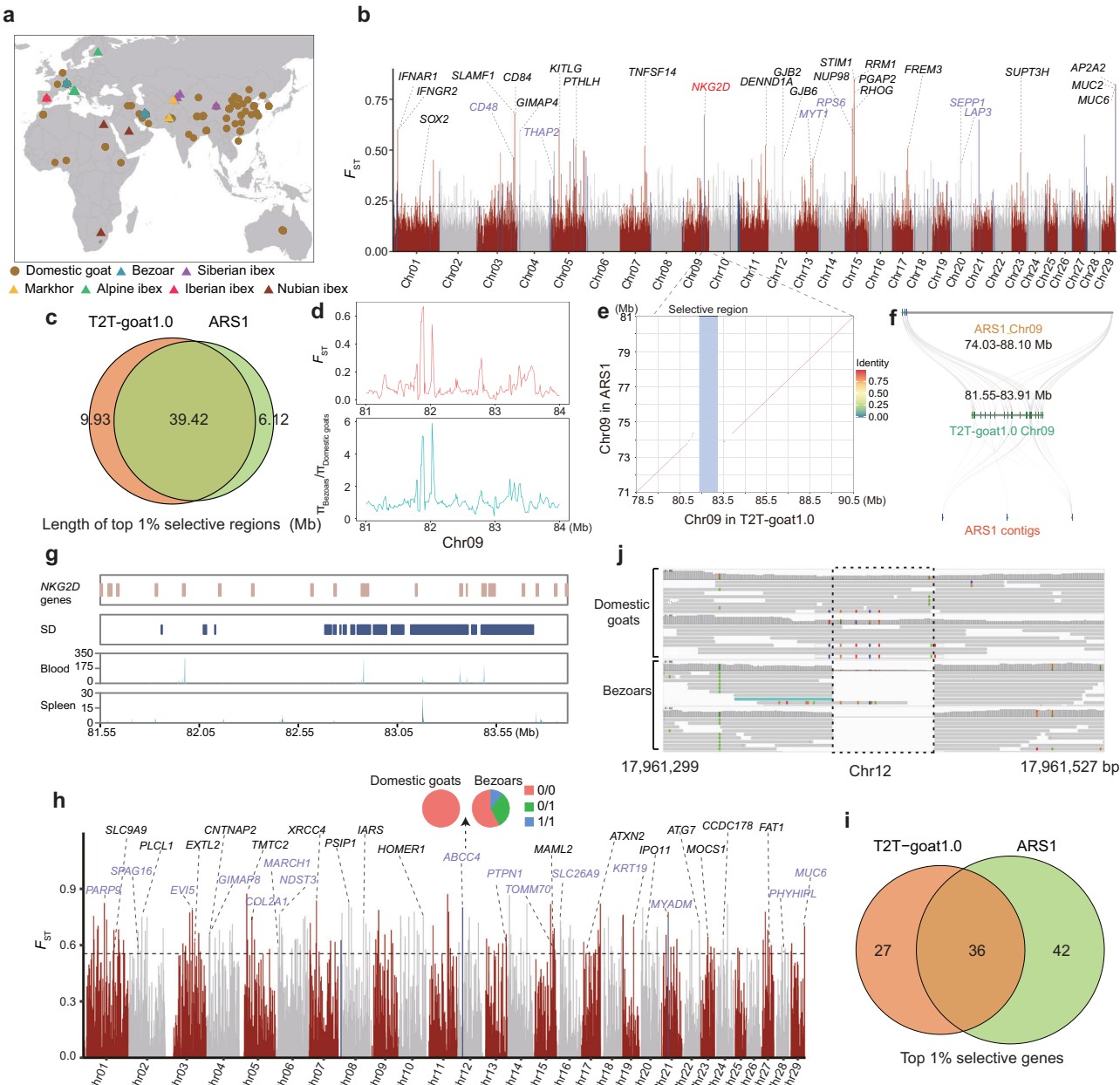

**Fig. 5 | Selection signatures for domestication. a** Geographic distributions of domestic and wild (including bezoars) goats all over the world whose whole gen- ome sequencing datasets are used in this study. **b** Genome-wide selective signals based on SNPs and $F_{ST}$ between bezoars and the domestic goats. A horizontal dash line is used to show the top 1% selective signals. The selective signals in the PURs are highlighted in blue bars. The gene symbols are shown in black for the ones iden- tified by both T2T-goat1.0 and ARS1, and the ones only by T2T-goat1.0 in purple. *NKG2D* in red uniquely identified by T2T-goat1.0 is selected to be demonstrated as followed. **c** Venn plot of selective regions based on SNPs and top 1% $F_{ST}$ values between T2T-goat1.0 and ARS1 as references. **d** Selective region with tandem *NKG2D* genes on chromosome 9 (Chr09) based on the $F_{ST}$ and π ratio of bezoars and domestic goats. **e** Syntenic plotting of the region with *NKG2D* genes on Chr09

between T2T-goat1.0 and ARS1. The selective region highlighted in blue is exactly located inside the region where tandem *NKG2D* genes are not assembled well in ARS1. **f** Collinearity of the region with tandem *NKG2D* genes on Chr09 between T2T- goat1.0 and ARS1. The tandem *NKG2D* genes in T2T-goat1.0 were found to corre- spond to the homologous genes not only on Chr09 of ARS1, but also on the three unplaced contigs. **g** Tandem *NKG2D* genes are in accordance with SDs and show their expressions in blood and spleen. **h** Selective signals based on SVs and $F_{ST}$ between bezoars and the domestic goats. The plot is made based on the same method to that of Fig. 5b. **i** Venn plot of selective genes that overlapped with selective regions based on top 1% $F_{ST}$ values of SVs identified by T2T-goat1.0 and ARS1. **j** A deletion was confirmed within *ABCC4* (IMCG12g00097) of bezoars in IGV.

various expressions according to RNA-seq in the spleen and blood (Fig. 5g).

We further investigated SVs found in domestication-associated genomic regions in bezoars and domesticated goat populations (Supplementary Data 5). The top 1% of $F_{ST}$ values identified a total of 202 SVs as candidate variants, 63 of which overlapped with the gene bodies (Fig. 5h) and shared 36 genes with those obtained with ARS1 as

the reference genome (Fig. 5i, Supplementary Fig. 23b, and Supple- mentary Data 15). These 63 genes were associated with phenotypic characteristics, for example, immune activity (*MUC6*, *GIMAP8*, *CNTNAP2*, *KCND3*, and *NBEAL1*), stress resistance (*ABCC4*, *PSIP1*, and *TOMM70*), and sperm flagellar function (*SPAG16*), and were classified into three groups. The first group, which included 36 genes, was detected by both T2T-goat1.0 and ARS1; the second, which included 23

genes, was detected in non-PURs of T2T-goat1.0 but was not found in ARS1; and the third, which included 4 genes, was detected only in the PURs of T2T-goat1.0. A 62-bp deletion was found in the *MUC6* gene of bezoars on CHI29, which was in accordance with the deletion allele in bezoars compared to goats, as previously reported[2]. Additionally, a deletion (54 bp) was found within *ABCC4* (gene ID: IMCG12g00097) of bezoars on PURs of CHI12 and was confirmed with the IGV[37] tool (v2.13) (Fig. 5j), whose allele frequencies showed a strong selective signal in domestic goats compared to bezoars. Further examination revealed that the region was annotated with 14 tandem *ABCC4* genes but was not completely assembled in ARS1. A duplication (137,204 bp) was found in the myeloid-associated differentiation marker (*MYADM*, IMCG21g00085) of bezoars on the PURs of CHI21 (Supplementary Fig. 24b), which involved 17 *MYADM* in neighboring regions, and *MYADM* was shown to be a host factor essential for parechovirus entry and infection[38].

## Selective signatures underlying the genetic improvement of cashmere traits

To decipher the genetic basis of the cashmere trait, we conducted a selection analysis of the cross-population composite likelihood ratio (XP-CLR) between the cashmere and noncashmere goat populations. A total of 46 individuals in eight typical northern cashmere populations (four from Tibetan plateau, three from Northern China, and one from India) were selected as the cashmere group in a comparison with 27 non-cashmere goat breeds (Supplementary Data 5), in

selective sweep analysis, as they held divergent genetic basis. We focused on the cashmere goats for cashmere trait selection, rather than the 19 breeds of northern Chinese goats in a previous study[22]. Using the top 1% of the XP-CLR score values, we identified a total of 2,692 selective signals, spanning 109.22 Mb and covering 1,440 genes (Fig. 6a and Supplementary Data 16). In addition to the previously reported cashmere-related *FGF5* on CHI6 and *EDA2R* on the X chromosome previously reported[22], we have identified selective signals associated with cashmere traits, such as the *KRT* genes on both CHI5 and CHI19, and the *FGF* genes on both CHI5 and CHI20. We also observed selective genes associated with coat color, such as *TYRP1* on CHI8, *KITLG* on CHI5 and *SLC24A4* on CHI21. Compared to those in the ARS1 reference genome (Supplementary Fig. 25a and Supplementary Fig. 25b), we identified 74.36 Mb selective regions uniquely in T2T-goat1.0 (Fig. 6b). Furthermore, we discovered 274 selective signals located in the PURs, covering 72 genes. A strong signal in the PURs of CHI12 (CHI12: 14.88 Mb – 16.58 Mb), which was supported by the π ratio estimate between the non-cashmere and cashmere goat populations (Fig. 6c), referred to the selection of *ABCC4* genes. The collinearity results indicated incomplete assembly of this region in ARS1 (Fig. 6d), where 14 tandemly repeated *ABCC4* genes enriched with SDs were assembled in T2T-goat1.0 (Fig. 6e). We also revealed significant differences in allele frequencies between cashmere and noncashmere populations in the intronic region of one *ABCC4* gene (IMCG12g00092, Supplementary Fig. 26a). The *ABCC4* gene is responsible for transporting a range of endogenous molecules and

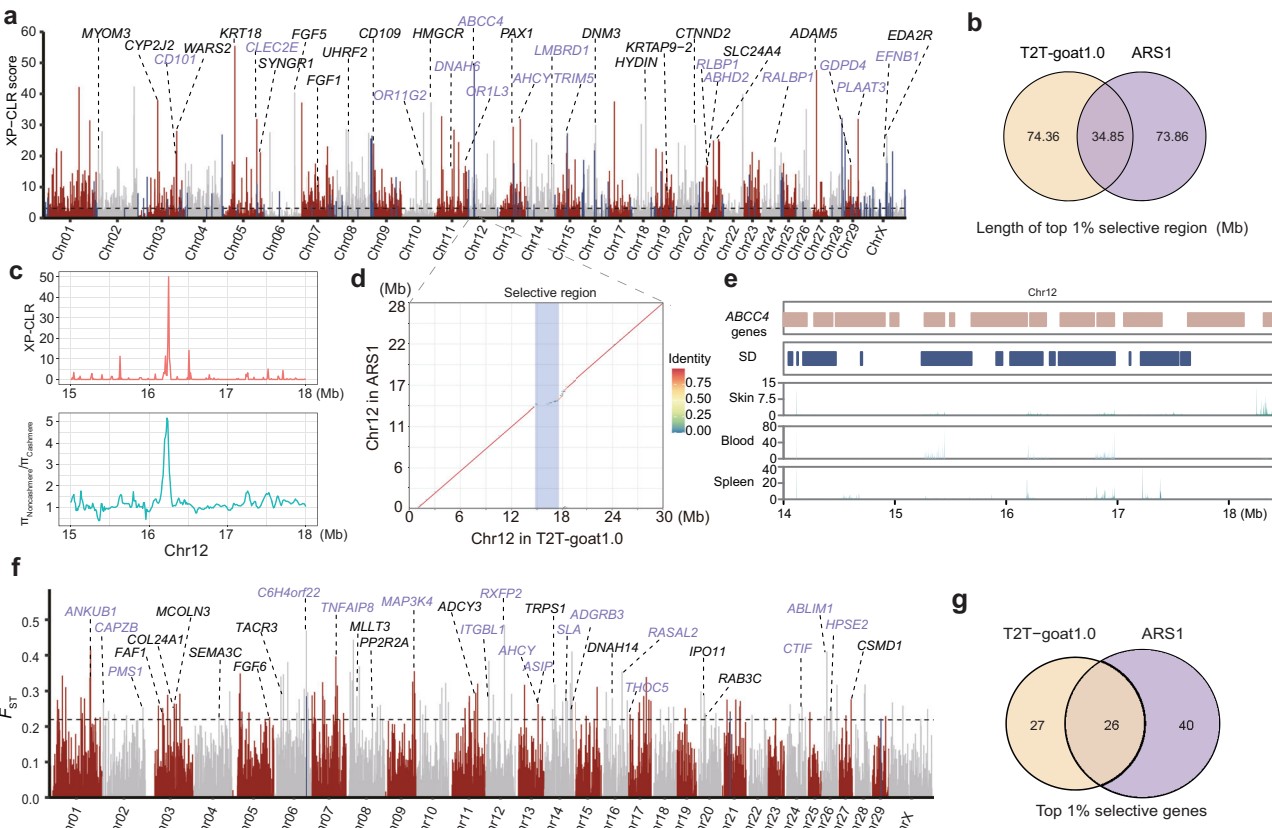

**Fig. 6 | Selection signatures for cashmere traits. a** Selective signals based on SNPs and top 1% XP-CLR scores for cashmere trait. The plot is made based on the same method to that of Fig. 5b. **b** Venn plot of selective regions based on SNPs and top 1% XP-CLR scores for the cashmere trait between T2T-goat1.0 and ARS1 as references. **c** Selective region with tandem *ABCC4* genes on Chr12 based on the XP-CLR score and π ratio of non-cashmere and cashmere goats. **d** Collinearity of the region with tandem *ABCC4* genes on Chr12 between T2T-goat1.0 and ARS1. The selective region

highlighted in blue bar is exactly located inside the region that was not assembled correctly in ARS1. **e** *ABCC4* genes are in accordance with SDs and showed their expression in skin, blood, and spleen tissues. **f** Selective signals based on SVs and top 1% $F_{ST}$ values between cashmere and non-cashmere goats. The plot is made based on the same method to that of Fig. 5b. **g** Venn plot of selective genes based on SVs and top 1% $F_{ST}$ scores for the cashmere trait between T2T-goat1.0 and ARS1 as references.

cellular detoxification and is deeply involved in keratinocyte differentiation and stratum corneum keratinization[39].

In addition to SNPs, 173 selective SVs associated with the cashmere trait were identified based on the top 1% $F_{ST}$ values between the cashmere and noncashmere populations (Fig. 6f and Supplementary Data 17). We found that 53 genes related to these SVs were involved in multiple pathways, including those related to hair growth (*FGF6*, *ASIP* and *MCOLN3*), horn formation (*RXFP2*), immune system (*TNFAIP8*, *CSMD1* and *TACR3*), and olfactory system (*OR2T12*). The 26 genes were identified by both the two genome assemblies (Fig. 6g and Supplementary Fig. 25c), and 27 genes were found uniquely by T2T-goat1.0. For example, the agouti-signaling protein (*ASIP*) gene is involved in the regulation of melanogenesis and hair pigment traits in terms of mutation and copy number[40,41], and was found to be associated with a duplication of 162,498 bp on CHI13 in cashmere goats, which also revealed a differentiation in copy number between cashmere and noncashmere goats (Supplementary Fig. 26b). The top 12 $F_{ST}$ values of SVs were selected to be double checked in the SVs based on long reads and confirmed for their presence (Supplementary Data 18). Taken together, our selection tests revealed quite a few genes and variants associated with cashmere formation during the selection of cashmere traits, and demonstrated the ability of T2T-goat1.0 to detect selective signals and candidate genes associated with phenotypic traits in goats.

## Discussion

To date, T2T-goat1.0 is one of the first mammal genomes to achieve complete, seamless assembly following the human T2T assembly[11]. With the advancement of T2T-goat1.0, a black box corresponding to highly repetitive regions was uncovered, especially for centromeric regions. T2T-goat1.0 revealed three types of centromeric satellite units in goats, with lengths of 816 bp, 702 bp and 22 bp; these findings are supported by the previous studies on centromeric satellite units of 816 bp[26] and 22 bp[42]. Satellite sequences of the Bovidae family are associated with chromosomal evolution[43], demonstrating the phylogenetic conservation; however, in mammals, the centromeric satellite repeat unit varies in length and harbors conserved bases in the evolutionarily conserved domain[44]. This phylogenetic conservation could be confirmed by the good alignment of sheep SatI and SatII against T2T-goat1.0 and the high similarity of SatIII with 83% of the homologs between sheep and goats[42]. Without the isolation and enrichment of satellite DNA by the methods of density gradient centrifugation or restriction enzyme digestion[43], the T2T genome assembly provided a global view of satellite sequences across each chromosome, and even revealed two variants within SatIII of 22 bp (Fig. 3b and Supplementary Fig. 15b). The conservation of centromeric satellites could be further investigated on acrocentric chromosomes in the T2T genome assemblies of Bovidae species, for example, cattle, sheep, and deer.

Due to the high repetitiveness and homologous regions with the X chromosome (i.e., PARs), the complete assembly of the Y chromosome is extremely challenging. Since the trio-binning method with parental sequencing data was reported for haplotype-resolved assembly[45], Y-chromosome assemblies turned to combining long reads and parental reads. The trio-binned reads enabled the T2T assembly of Y chromosomes, for example, Y-chromosome assemblies of human[19,46] and apes[47]. T2T-goat1.0 successfully resolved the PARs and uncovered 27 gene copies of *TSPY* and 14 copies of *HSFY*. An increase in the number of copies of the testis-specific *TSPY* gene was also reported on the Y chromosomes of human (45 copies)[19] and cattle (94 copies)[48]. This dosage effect addresses spermatogenesis and is related to the regulation of spermatogenic efficiency[19]. The copy number of *TSPY* reportedly varies significantly among various populations[48]. *HSFY* is believed to be expressed predominantly in the testis and to regulate spermatogenesis by mediating the expression of heat shock proteins (HSPs)[49]. The number of *HSFY* copies varies among the species, with only 2 copies in human[19], 8 copies in feline[50], 6 copies in sheep[51], and

197 copies in cattle[49]. No other multicopy protein-coding gene families were identified on the Y chromosome in T2T-goat1.0. The Y chromosome of goats features a unique centromeric repeat, CenY, which was not found on the other chromosomes with centromeric SatI-SatIII. A unique centromeric repeat unit on the Y chromosome was also found in the other animals. For example, a 1747-bp repeat unit was found in the Mongolian gerbil[52].

The T2T-goat1.0 assembly enables more capacity to search for additional selective signatures. First, our assembly revealed 486 NAGs that were not found in ARS1. Similarly, T2T genome assemblies added 227 NAGs to the previous human genome reference CHM13[10], and 246 NAGs in maize[14], 314 NAGs in rice[53], and 316 NAGs in soybean. Another key point for an addition of selective signatures is that the T2T-goat1.0 significantly improved repetitive sequences in the goat genome assembly, which facilitated the accurate assembly of tandemly repeated genes under selection signals, such as *NKG2D* and *ABCC4*. These regions are valuable for selection in terms of both copy number variations and mutations in one of the tandem genes. The strategy of multiple gene copies might be associated with the selection of copy number variations, such as *KIT*[3] and *ASIP*[54] for coat color in goats. Remarkable contraction of the *ABCC4* genes was found in domestic goats compared to bezoars[55], and it was believed that the *ABCC4* genes were involved in the immune system. We confirmed the selection of *ABCC4* (IMCG12g00097) gene during domestication, which showed a deletion in bezoars. Furthermore, we found that another *ABCC4* gene (IMCG12g00092) was under selection for the cashmere trait, which proved the occurrence of multiple selections in the region with tandemly duplicated *ABCC4* genes, which function in different tissues. Together with other ABC transporters, *ABCC4* genes might be potentially important regulators of epithelial biology in human hair follicles[56]. These genes are under selection for cashmere traits in goats and deserve further investigation. *FGF5* on CHI6 and *EDA2R* on CHIX were previously shown to be associated with cashmere traits[22], suggesting the reliability of our selection results (Fig. 6a). Additionally, we identified several selective genes that are significantly associated with cashmere traits; for example, multiple *KRT* genes under the strongest selection signal on CHI5 were found (Fig. 6a). Keratin proteins, as a super gene family, are regarded as markers of the secondary hair follicle cycle in cashmere goats[57]. Multiple copies of *KRT* genes were shown to cluster on CHI5, which had the strongest selective signals. We showed strong evidence that these regions consisted of multiple copies of genes under selection in goat genomes, including those with *ABCC4*, *KRT*, *NKG2D*, and *MYADM*. Thus, T2T-goat1.0 enabled the accurate discovery of these regions and their variants.

In conclusion, the T2T-goat1.0 assembly enabled a comprehensive evaluation of complex regions with repetitive sequences, improved the alignment of short and long reads, increased the variant calling rate, and identified additional novel selective signatures previously unreported. Population genetic analyses benefited from not only more SNPs in PURs for selective signals but also accurately assembled regions with tandemly duplicated genes.

## Methods

### Sample collection and sequencing

A 4-month-old buck of the Inner Mongolia cashmere goat from the National Preservation Farm for Arbas Cashmere Goat (Ordos, Inner Mongolia, China) was selected for DNA sequencing, RNA-seq and Iso-seq in the T2T genome assembly and subsequent analyses (Fig. 1a). The parents with complete breeding records were also sampled for the assistance of Y-chromosome and haplotype-resolved assembly (Fig. 1a). Additionally, 14 tissues (Supplementary Data 1) from Inner Mongolia cashmere goats on the same farm were collected for RNA-seq and gene annotation.

Five representative goat breeds with specialized traits, i.e., Liaoning cashmere goats with high wool production, Zhongwei goats

with white fur and good leather, Jining gray goats with high fertility, Boer goats with high-quality meat, and Tibetan goats with good adaptation to plateaus, were selected for genome sequencing to produce the PacBio long reads (Supplementary Methods), which were used to call SVs (Supplementary Data 1 and Supplementary Fig. 19a).

Blood from the 4-month-old buck and its parents was collected in 10-ml BD Vacutainer blood collection tubes (Cat# 368589, Becton Dickinson, NY, USA). The tubes were placed on dry ice, transferred to the laboratory, and stored at −80 °C until DNA extraction. High-molecular-weight genomic DNA was extracted based on the CTAB method[58], and purified with a Blood & Cell Culture DNA Kit (Cat# 13343, Qiagen, Beijing, China) following the manufacturer's protocols. Library construction and sequencing for PacBio (20-kb libraries, Pacific Biosciences, CA, USA), ultralong ONT (200-kb libraries, Oxford Nanopore Technologies, Oxford, UK), MGI (MGI Tech, Shenzhen, China), Hi-C and Bionano optical maps were all performed at the Grandomics Ltd. (Wuhan, China), and are described in the Supplementary Methods.

All animal procedures were performed in accordance with the guidelines for animal experiments approved by the China Agricultural University Institutional Animal Care and Use Committee (CAU20160628-2).

## Bulk RNA-seq and Iso-seq
Total RNA was extracted from blood using TRIzol reagent in an RNAprep Pure Tissue Kit (Cat# 4992236, TIANGEN Biotech, Beijing, China). After the determination of RNA purity and concentration with Nanodrop (Thermo Fisher Scientific, Waltham, USA) and Qubit (Thermo Fisher Scientific, Waltham, USA), high-quality RNA samples (RIN > 8, OD260/OD280 = 1.8–2.2, OD260/OD230 > 2.0) were used for cDNA synthesis in bulk RNA-seq and Iso-seq. Sequencing libraries for Iso-Seq were constructed using the SMRTbell Template Prep Kit 2.0 (Pacific Biosciences, CA, USA), and sequenced on the PacBio Sequel II platform using the Sequel Binding Kit 2.0 (Pacific Biosciences). For RNA-seq, fragmentation and cDNA synthesis were performed using the MGIEasy RNA Library Prep Kit V3.1 (Cat# 1000005276, MGI), and circularization of double-stranded cDNAs was achieved using the splint oligos in the MGIEasy Circularization Kit (Cat# 1000005260, MGI). Single-stranded circular DNA (ssCir DNA) was used as the final library, and the qualified libraries were sequenced on the DNBSEQ-T7RS platform (MGI). The other 14 tissues (Supplementary Data 1) were also subjected to RNA extraction and Illumina sequencing (details in the Supplementary Methods).

## Fluorescence in situ hybridization (FISH)
The sequences of 45S rDNA, 1474-bp CenY, 702-bp SatII, and 22-bp Sat III were synthesized as probes and labeled with Dig-dUTP or Bio-dUTP (Roche Diagnostics, Basel, Switzerland) using the Biotin and Dig Nick Translation Mix (Roche, Mannheim, Germany). Chromosome preparations were made from fibroblast cell culture as described previously[59]. Slides with cell suspensions at metaphase were hybridized with hybridization mix containing probes and washed for imaging. The hybridization signals were detected based on Alexa Fluor 488 streptavidin (Thermo Fisher Scientific, Waltham, MA, USA) for biotin-labeled probes and rhodamine-conjugated anti-digoxigenin (Roche Diagnostics, Basel, Switzerland) for digoxigenin-labeled probes. Chromosomes were counterstained with DAPI in a mounting medium (Vector Laboratories, Odessa, Florida, USA). Chromosomes in FISH were observed in an Olympus BX63 fluorescence microscope equipped with an Olympus DP80 CCD camera (Olympus, Tokyo, Japan), and images were acquired using cellSens Dimension 1.9 software (Olympus, Tokyo, Japan).

## Draft assembly
For the 4-month-old buck, we generated 141.03 Gb of PacBio HiFi (49.3× coverage) with an N50 of 18.95 kb (Supplementary Data 1), which was used to construct the draft assembly (GV1) using Hifiasm[14] (v0.16.1-r375) with default parameters. The nonnuclear genome sequences were removed based on the alignment results against the NCBI nucleotide nonredundant database using BLASTN[60] (v2.10.0) with the parameter of "-evalue 1e-5". All contigs were aligned to the goat reference genome assembly ARS1 (GCF_001704415.1) to evaluate the accuracy of GV1 using Minimap2[61] (v2.26) before further processing.

## Scaffolding with Bionano optical maps
Next, we generated optical maps for scaffolding. Clean Bionano data (molecular length >150 kb and minSites >9) of 1209.75 Gb (~ 362.66× coverage) was used to perform a de novo assembly, and generate a consensus map file (.cmap) using the BioNano Solve package (v3.5.1, https://bionano.com/software-downloads/). To generate more continuous consensus genome maps, hybrid scaffolding was conducted between the genome contigs (GV1) and the above genome maps using the "HybridScaffold" module of the Bionano Solve package. The hybrid scaffolds were aligned to the original in silico maps from sequence scaffolds using RefAligner (https://bionanogenomics.com/support/software-downloads/), and sequences from GV1 were split at breakpoints (i.e., the conflict positions). The scaffold sequences were produced from the XMAP-formatted alignment and outputted as FASTA files.

Breakpoints generated by the Bionano tools were inspected manually by the Bionano Access software (v1.7; https://bionano.com/software-downloads/), and further verified by the ONT and HiFi data. In summary, ONT and HiFi reads were aligned to the GV1 assembly using Minimap2 with the parameter of "--secondary=no", and were visualized with the IGV tool (v2.13) to confirm the exact breakpoints. When the ONT and HiFi read alignments supported the original sequences [> 5 reads uniquely aligned and mapping quality (MAPQ) score > 60] where the breakpoints occurred using Bionano tools, the gaps generated by Bionano were abandoned, and the contigs were manually returned. The resulting scaffolded genome (GV2) was subsequently generated to construct the pseudo-chromosome-level genome assembly as follows.

## Pseudochromosome construction
After trimming using fastp (v0.23.1)[62], high-quality Hi-C reads were aligned to the scaffolded genome assembly (GV2) using Bowtie2[63] (v2.3.2) with the parameters "-end-to-end --very-sensitive -L 30". Valid read pairs were generated to identify interactions using HiC-Pro[64] (v2.8.1). All the sequences were clustered, ordered and oriented onto the 31 pseudochromosomes (29 autosomes and X and Y chromosomes) using the program LACHESIS[65], with the following parameters: "CLUSTER_MIN_RE_SITES = 100, CLUSTER_MAX_LINK_DENSITY = 2.5, CLUSTER_NONINFORMATIVE_RATIO = 1.4, ORDER_MIN_N_RES_IN_TRUNK = 60, ORDER_MIN_N_RES_IN_SHREDS = 60". The Y chromosome was abandoned due to failed assembly of the homologous region with the X chromosome, and independent assembly was adopted for the Y chromosome with the assistance of the parent's genome sequences. Placement and orientation errors with obvious discrete chromatin interaction patterns were manually adjusted to assemble the chromosome-level assembly (GV3) of 30 pseudochromosomes (29 autosomes and the X chromosome).

## Gap filling
To fill the gaps, ultralong ONT reads with a length of >100 kb were mapped to all the 30 pseudochromosomes of GV3 using Minimap2 with the option of "-x map-ont". The ultralong ONT reads that were precisely aligned to the ends of the neighbor contigs connecting the beginning site and ending site of a gap (identity ≥ 95%, coverage ≥ 90%, and QV ≥ 20) were selected as anchor sequences. Short gaps were easily filled by extending the anchor sequences. To fill large gaps, we selected three kinds of ultralong ONT reads based on the read

alignment results to conduct local assembly. In addition to the first kind of anchor sequences, we retrieved the second kind of ONT reads, which were roughly aligned to the neighboring regions of the gaps (identity <95% and coverage <90%). The reads without any hits against the pseudochromosomes of GV3 were selected as the third kind, as some of them might have originated from one specific gap. The above three kinds of reads were aligned with each other using Minimap2 (v2.23) to produce a library for the overlapping of the pairwise ONT reads. Gaps are mostly located in highly repetitive sequences, such as centromeres, and can result in overlapping errors with a high frequency of $k$-mers. To reduce the false positive overlap caused by repetitive sequences, $k$-mer ($k = 23$) hash tables were calculated based on the MGI short reads using Meryl (v1.4.1)[66], and rare $k$-mers or low-frequency $k$-mers were determined when the depth was less than the average depth. Subsequently, with the first kind of reads as the starting and ending points, a string graph was built for one specific gap using the Nextgraph module in NextDenovo[67] (v2.5.2), based on the overlapping ONT reads and their rare $k$-mers. The graph with the longest accumulated length of rare $k$-mers was selected to assemble the sequences to fill the gaps. All ONT reads were aligned to the gap-filled genome to check the coverage and depth of gap regions in IGV (v2.13), and the gap region sequences were adjusted manually with multiple iterations. We subsequently generated the gap-free genome assembly GV4, which covered all the unplaced contigs in GV3.

## Y-chromosome assembly

The Y chromosome was assembled using a trio-binning strategy and hash tables of $k$-mers from the parents' short reads. In brief, *21-mer* libraries were constructed from MGI short reads (~120× coverage) for the buck individual for the T2T genome assembly and its parents using the Jellyfish[68] (v2.3.0). Furthermore, the paternal and maternal unique *21-mers* were identified if a *21-mer* was found only in either the father or mother, respectively, and had a sequence depth of >3 and <120 (the average depth across the whole genome) in the T2T buck individual. The ultralong ONT reads were chosen to determine paternal and maternal origination with more paternal and maternal *21-mers* respectively. The paternal ONT reads sequenced from the autosomes were subsequently removed out to retrieve those potentially from the X and Y chromosomes, which were used to construct the assembly graph of the Y chromosome using the Nextgraph module in NextDenovo (v2.5.2). The Y chromosome assembly was further improved by filling gaps and correcting possible assembly errors according to the Y contigs by the trio-binning model of Hifiasm (v0.16.1–r375) based on HiFi reads and *31-mers* from the T2T buck parents. The Y chromosome was combined with all the autosomes and the X chromosome to obtain the genome assembly GV5.

## Telomere assembly

We pooled three types of HiFi reads for independent telomere assembly, namely, those that contained >10 telomere-specific repeats (i.e., AACCCT or AGGGTT), those that were not mapped to the genome assembly GV5, and those that were partially aligned to the 1-Mb chromosomal ends. Telomere assembly was performed using HiFiasm (v0.16.1-r375) with the default parameters, and the contigs with telomeric sequences were aligned to GV5 using Minimap2 to determine their potential chromosomal locations. Telomeres were placed at the 1-Mb ends of each chromosome based on the overlap between contigs and chromosomal ends using RagTag[69] (v2.1.0, Supplementary Methods). The updated chromosomal ends were manually checked for complete alignments of the HiFi and ONT long reads.

## Genome polishing

To polish the genome assembly GV5, we developed a pipeline[70] (https://github.com/Wuhui2024/CAU-T2T-Goat) that involves long and short reads (Supplementary Fig. 5 and Supplementary Methods). First, HiFi reads were mapped to GV5 using Minimap2, and the low-quality regions were then determined based on the HiFi read alignment (those with a MAPQ score ≤1, clipped reads at the both ends, and <3 HiFi aligned reads), while the remaining regions were defined as high-quality regions. The low-quality regions were polished for the first round based on HiFi long reads using Nextpolish2[71] (v0.1.0), the second round based on ONT long reads and the third round based on MGI short reads using Nextpolish[72] (v1.4.1). Furthermore, the low- and high-quality regions were both polished based on HiFi long reads using Nextpolish2, before being merged as a whole genome. To further improve the QV, the misassembled centromeric regions preferably with abnormal coverage and high $k$-mer error rates, were manually examined and adjusted for the local assembly. The last round of polishing was performed with HiFi reads using Nextpolish2, to generate the final seamless genome assembly T2T-goat1.0.

## Haplotype genome assembly

The initial assembly for the autosomes of the haplotype-resolved genomes (T2T-goat1.0P and T2T-goat1.0M) was performed by using Hifiasm (v0.16.1-r375) based on the trio mode. The scaffolding, gap filling and polishing for T2T-goat1.0P and T2T-goat1.0M were performed with the similar method for T2T-goat1.0. The ONT and HiFi reads were binned for paternal and maternal origin based on the $k$-mer ($k = 21$) generated from parental short reads, which is the same to the trio-binning strategy as shown for Y chromosome assembly. The binned long reads were used to fill gaps and polish the haplotype genomes by using Nextpolish2 (v0.1.0), rather than the NGS data, to avoid the introduction of the other haplotype sequences.

## Validation of the T2T genome assembly

First, for read coverage analysis, MGI short reads and long reads of ONT and HiFi were aligned to T2T-goat1.0 using BWA (v0.7.17) and Minimap2, respectively. Genome coverage was subsequently evaluated with a window size of 100 kb using Bamdst (a BAM depth statistics tool; https://github.com/shiquan/bamdst), and plotted using the KaryoPloteR[73] package (v1.8.4). For collinearity analysis, the two chromosomes were compared using Minimap2, and genome synteny was visualized using paf2doplot (https://github.com/moold/paf2dotplot) and GenomeSyn[74]. The completeness of the T2T-goat1.0, T2T-goat1.0P, and T2T-goat1.0M genomes was evaluated using BUSCO[24] (v4.0.5) based on the mammalia_odb10 database. To evaluate the consensus accuracy, a $k$-mer library ($k = 21$) was built from the MGI short reads using Meryl (v1.4.1), and quality scores per base (i.e., QV) and switch errors were calculated for the whole genome assembly using Merqury[66] (v1.3). The PURs were determined based on the alignments between T2T-goat1.0 and ARS1 (Supplementary Methods), according to the previous method[11].

## Annotation of repetitive sequences

The methods of de novo prediction and homolog-based searching were used to annotate the repetitive sequences including tandem repeats (TRs), transposons, satellites, etc. Four programs, such as GMATA[75] (v2.2), Tandem Repeats Finder[76] (TRF, v4.09.1), MITE-Hunter[77] (v1.0) and RepeatModeler2[78] (v2.0.4), were applied for de novo prediction. The de novo repeat libraries were merged into a repetitive sequence database, which was used for homolog-based searching via RepeatMasker[79] (v4.1.4), with default parameters.

Segmental duplications (SDs) were detected using BISER[80] (v1.4), and low-quality SDs were filtered when they did not meet the following requirements[23]: 1) Sequence identity >90%, 2) ≤50% gapped sequence in the alignment, 3) Aligned sequence of ≥1 kb, and 4) ≤70% satellite sequence annotated by RepeatMasker. Determination of SDs in PURs was further described in Supplementary Methods.

## Protein-coding gene annotation

Three strategies were used to implement the gene annotation in the T2T-goat1.0 assembly, namely, the de novo, homology-based, and transcriptome -based approaches. For the transcriptome-based approach, a total of 93 RNA-seq datasets from 40 goat tissues[5,81–98], including 50 ones downloaded from NCBI and 43 newly sequenced ones in this study, were collected (Supplementary Data 1 and Supplementary Data 19), to assemble 145,492 transcripts and obtain 19,437 nonredundant transcripts with Stringtie[99] (v1.3.4 d). Full-length transcripts from Iso-seq were aligned to T2T-goat1.0 using Minimap2, and redundancies were removed using the "collapse_isoforms_by_sam.py" command of IsoSeq3 (v3.8.2, https://github.com/PacificBiosciences/IsoSeq). Based on the combined nonredundant transcripts from RNA-seq and Iso-seq, gene models were predicted and determined using PASA[100] software (v2.5.2). For homology-based prediction, homologous proteins from seven genome assemblies of four closely related species (goat, sheep, human, and mouse) were downloaded from NCBI (Supplementary Data 20), and gene models were determined after alignment to T2T-goat1.0 using GeMoMa[101] software. For the de novo prediction approach, genes were predicted based on the training set of the above genes in T2T-goat1.0 with repeats masked by RepeatMasker using AUGUSTUS[102] (v3.3.1). We integrated all the above predictions of the gene models using EvidenceModeler[103] (v1.1.1) with default parameters, and the genes with potential transposable elements (TEs) were removed using TransposonPSI (v1.0.0, https://github.com/NBISweden/TransposonPSI). The presence of >700 known genes in previous publications was manually examined to confirm the completeness of the gene annotations in T2T-goat1.0. The gene annotations were manually corrected in a modified version of IGV, IGV-GSAman (https://gitee.com/CJchen/IGV-sRNA), based on the transcript evidence.

## Functional annotation of protein-coding genes

To annotate the protein-coding genes, protein sequences were aligned to three databases, Swiss-Prot (http://web.expasy.org/docs/swiss-prot_guideline.html), Kyoto Encyclopedia of Genes and Genomes (KEGG, https://www.genome.jp/kegg/), and the NCBI nonredundant (NR) protein database (ftp:/ftp.ncbi.nih.gov/blast/db/), using BLASTP[60] (v2.10.0) with the parameters: "-evalue 1e-5 -max_target_seqs 1". The proteins were also aligned to the Pfam database (https://pfam-legacy.xfam.org/) using InterProScan[104] (v5.58), and GO terms (http://geneontology.org) were retrieved.

## Centromere identification

A chromatin immunoprecipitation (ChIP) assay was performed to determine the locations of centromeres based on the binding properties of histone-specific antibodies. In brief, 10 ml of fresh blood from the same buck used for the T2T assembly was cross-linked using 1% formaldehyde for 15 minutes at room temperature. The reaction was then quenched by adding glycine at a final concentration of 200 mmol/L. Chromatin was sonicated to obtain soluble sheared chromatin DNA with an average length of 200-500 bp. Afterward, immunoprecipitation was conducted with Phospho-CENP-A (Ser7) antibody (Cat# 2187, Cell Signaling Technology, Beverly, MA, USA). Immunoprecipitated DNA was used to construct sequencing libraries with the NEXTflex ChIP-Seq Library Prep Kit for Illumina sequencing (Cat# NOVA-5143-01; Bioo Scientific, Austin, TX, USA). The final libraries were sequenced on a HiSeq Xten platform with paired-end mode (Illumina, San Diego, CA, USA). After trimming using fastp[62] (v0.23.1), the clean reads were aligned to T2T-goat1.0 using Bowtie2 (v2.4.2) with the parameters "--very-sensitive --no-mixed --no-discordant -k 10". The read depth for ChIP enrichment in 5-kb windows was calculated using BEDTools (v2.30.0) and plotted using the KaryoPloteR package (v1.8.4).

The linguistic sequence complexity was calculated with the program NeSSie[105] in a window size of 10 kb. The locations of the centromeres were subsequently determined based on the lower sequence complexity due to the enrichment of repetitive sequences[52]. Sequence identity heatmaps for centromeric repeats were generated by StainedGlass[106] (v0.5), with similar repeats colored according to percentage identity.

## Identification of repetitive sequences inside centromeres

The known 816-bp centromeric satellite DNA sequence (U25964.1) was subjected to BLAST searches against T2T-goat1.0, together with ChIP-Seq signals, to preliminarily anchor the centromeric regions. A k-mer library was generated based on the above deduced centromere regions using KMC[107] (v3.1.1) with the parameters "-fm -k151 -ci20 -cs100000". Based on the centromeric k-mers and their frequencies, centromeric repeat units were identified using the program SRF[25] and were clustered into three groups of minimal satellite repeat units (SatI, SatII and SatIII) based on the sequence identity. The sequence similarity of the goat Y chromosome centromeric repeat unit (CenY, 1474 bp) with the sheep CenY (2516 bp) was determined (Supplementary Methods) using BLASTN (v2.10.0). These centromeric satellite repeats were aligned back to T2T-goat1.0 to estimate their abundance using BLASTN (v2.10.0). The final centromeric regions were determined based on the distribution of these four types of repeat units.

## Methylation determination by long reads

DNA methylation was calculated based on the HiFi and ONT long reads. HiFi reads were aligned to T2T-goat1.0 using pbmm2 (v1.13.0) (https://github.com/PacificBiosciences/pbmm2) with the default parameters. The bam file was processed using the "aligned_bam_to_cpg_scores" command of pb-CpG-tools software (v1.0.0) (https://github.com/PacificBiosciences/pb-CpG-tools). The site probabilities of 5-methylcytosine (5mC) methylation were determined and viewed in the IGV program (v2.13). The methylated base (5mC) was also generated from the fast5 format of ultralong ONT reads using Nanopolish[108] (v0.14.0). Methylation coverage was assessed in a window size of 20 kb using BEDTools[109] (v2.31.0).

## Structural variation identification

To obtain a high-quality collection of structural variants (SVs), we utilized three tools for the SV calling, namely, pbsv (v2.9.0, https://github.com/PacificBiosciences/pbsv), cuteSV[110] (v2.0.2), and Sniffles (v2.0.7, https://github.com/fritzsedlazeck/Sniffles). We first aligned the HiFi reads to T2T-goat1.0 and ARS1 to generate bam files using Minimap2 and SAMtools[111] (v1.16). With the program pbsv, we investigated SV signatures based on bam files and called SVs based on PacBio reads for all the five goat samples using the "discover" and "call" modules of pbsv. For the program cuteSV, SVs were called with the parameters of "--max_cluster_bias_INS 1000 --diff_ratio_merging_INS 0.9 --max_cluster_bias_DEL 1000 --diff_ratio_merging_DEL 0.5 --genotype". For the program Sniffles, we performed SV calling with default parameters. The SVs identified by the three programs were merged using the "merge" command of SURVIVOR[112] (v1.0.7), with a merging bin of 1000 bp and an SV length >50 bp.

## SNP and SV calling based on short reads

The whole-genome sequence data of caprine genomes, which included 448 domesticated and 68 wild goat individuals, were downloaded from public databases[2–4,113–131], including NextGen consortium projects (PRJEB3134, PRJEB3135, PRJEB3136, and PRJEB3140) (Supplementary Data 5). First, we performed quality control for the raw sequences using FastQC (v0.11.9; https://www.bioinformatics.babraham.ac.uk/projects/fastqc/) and Trimommatic[132] (v0.39). Clean short reads were aligned to T2T-goat1.0 and ARS1 using BWA[133] mem (v0.7.17-r1188) with the parameters "-k 23 -M". The bam files were sorted using SAMtools[111] (v1.16) and were subsequently used to call SNPs via GATK[134] (v4.3) as

described previously[135]. The raw SNPs were filtered using the VariantFiltration module of GATK with the following parameters: "QD < 2.0||QUAL < 30.0||SOR > 3.0||FS > 60.0||MQ < 40.0 || MQRankSum <−12.5 || ReadPosRankSum <−8.0". After removing those with more than 10% missing genotypes and a MAF (minor allele frequency) <0.05 using VCFtools[111] (v0.1.16), high-quality biallelic SNPs were retained for subsequent analysis.

Three tools were used to call SVs based on short reads, namely, Delly[136] (v0.8.7), Manta[137] (v1.6.0) and Smoove (v0.2.8, https://github.com/brentp/smoove), and the SVs identified by the three programs were merged for all the goat individuals using SURVIVOR (v1.0.7), with a merging bin of 1000 bp and an SV length of >50 bp.

## Population genetic analyses

A phylogenetic tree was constructed using the high-quality autosomal biallelic SNPs and the neighbor-joining (NJ) method in the program PHYLIP[138] (v3.697) and visualized by iTOL[139] (v6.8.1). LD pruning was performed for independent SNPs using PLINK[140] (v2.00a3.7) with the following parameters: "--indep-pairwise 50 5 0.2". PCA was performed on the independent SNPs with the "smartpca" command of the EIGENSOFT[141] package (v8.0.0). Additionally, population genetic structure was inferred based on the independent SNPs using ADMIXTURE[142] (v1.3.0). The LD decay ($r^2$) was measured with a maximum distance of 300 kb using PopLDdecay[143] (v3.42).

## Selective sweeps for domestication and cashmere trait improvement

To identify selective signals based on both SNPs and SVs, $F_{ST}$ values were calculated with 50-kb windows and a 10-kb step between the whole genomes of bezoars and domestic goats using VCFtools (0.1.16). Additionally, we calculated XP-CLR scores using XP-CLR[144] (v1.1.2) with the parameters "-L 0.95 -P -M 600 --size 50000 --step 10000" and identified potential selective signatures associated with selection for the cashmere traits. Nucleotide diversity (π) was calculated within 50-kb windows using VCFtools (v0.1.16). The top 1% of $F_{ST}$ and XP-CLR values were considered as putative selective sweeps associated with domestication and cashmere traits, and the π ratio between the pairwise populations was used to confirm the selective sweeps.

## Reporting summary

Further information on research design is available in the Nature Portfolio Reporting Summary linked to this article.

## Data availability

The genome assemblies T2T-goat1.0, T2T-goat1.0P, and T2T-goat1.0M; and T2T-goat2.0, T2T-goat2.0P, and T2T-goat2.0M are available in NCBI under accession numbers GCA_040806595.1 [https://www.ncbi.nlm.nih.gov/datasets/genome/GCA_040806595.1/], GCA_041920685.1 [https://www.ncbi.nlm.nih.gov/datasets/genome/GCA_041920685.1/], GCA_041736475.1 [https://www.ncbi.nlm.nih.gov/datasets/genome/GCA_041736475.1/], GCA_041735815.1 [https://www.ncbi.nlm.nih.gov/datasets/genome/GCA_041735815.1/], GCA_041053135.1 [https://www.ncbi.nlm.nih.gov/datasets/genome/GCA_041053135.1/], and GCA_040970035.1 [https://www.ncbi.nlm.nih.gov/datasets/genome/GCA_040970035.1/], respectively. Raw sequencing data generated in this study, including PacBio HiFi data, ultralong ONT data, MGI data, Iso-seq data and ChIP-seq data, can be achieved from National Genomics Data Center (https://ngdc.cncb.ac.cn/) under BioProject number PRJCA022847, and NCBI under BioProject number PRJNA1062519. The details of data mentioned above and other publicly available data downloaded in this study are provided in Supplementary Data 1, 5 and 19. T2T-goat2.0 is the update of T2T-goat1.0, with all the centromeres placed on the leftmost ends of acrocentric autosomes. Source data are provided with this paper.

## Code availability

Custom scripts and codes used in this study are available at GitHub (https://github.com/Wuhui2024/CAU-T2T-Goat) and Zenodo (https://doi.org/10.5281/zenodo.13917328). Software and parameters used are stated with more details in the Supplementary Methods.

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

## Acknowledgements

We thank Shaoqing Liu at Inner Mongolia Yiwei White Cashmere Goat Co., Ltd. for helping sampling the T2T buck and the parents, and Yueqing Wang and Xueyan Feng (China Agricultural University) for the help on data analysis. This work was supported by grants from the National Key Research and Development Program-Key Projects (2021YFD1200900 to M.H.L. and 2022YFF1003402 to S.G.J.), the Biological Breeding-National Science and Technology Major Project (2023ZD0407106 to M.H.L.), the National Natural Science Foundation of China (nos. 32320103006 to M.H.L., 31825024 to M.H.L., and 31661143014 to M.H.L.), the Project of Northern Agriculture and Livestock Husbandry Technical Innovation Center, Chinese Academy of Agricultural Sciences (BFGJ2022002 to M.H.L.), the Strategic Priority Research Program of Chinese Academy of Sciences (No. XDA24030205 to M.H.L.), and the Second Tibetan Plateau Scientific Expedition and Research Program (STEP; no. 2019QZKK0501 to M.H.L.). We thank all synonymous reviewers for their valuable comments on the peer review of this work. We also thank the High-performance Computing Platform of China Agricultural University.

## Author contributions

M.-H.L. conceived the project. M.-H.L and S.-G.J. supervised the study. H.W. and L.-Y.L. performed genome assembly and data analysis. Y.-H.Z. conducted the experiments on FISH and ChIP-seq, and was involved in generation of the ONT and PacBio data and data analysis. L.-M.Z. performed the telomere analysis, and participated in the discussion of results. J.-H.H. was involved in interpretation and plotting of the results. Z.-H.L., C.-Y.Z., Z.-X.W., and Y.-C.W. performed the RNA-seq sample collection and transcriptome analysis. Z.-H.L. provided part help and financial support for the study. For PacBio sequencing to call SVs, H.-H.E., Y.Y., H.-D.D., and Z.-Q.Z. collected the blood samples of Zhongwei goat, W.-L.B., D.H., and X.-T. D. collected the blood samples of Liaoning cashmere goat, Y.-L.R., X.-J.W., L.-Y.L. and D.-X.M. collected samples of Jining gray goat, R.D. and L.-Y.L. collected samples of Tibetan goat, and H.-L.C., L.-Y.L. and D.-X.M. collected samples of Boer goat. S.-G.J., H.W., and M.-H.L. wrote the manuscript.

## Competing interests

The authors declare no competing interests.

## Additional information

[1]Frontiers Science Center for Molecular Design Breeding (MOE); State Key Laboratory of Animal Biotech Breeding; College of Animal Science and Technology, China Agricultural University, Beijing 100193, China. [2]Northern Agriculture and Animal Husbandry Technical Innovation Center, Chinese Academy of Agricultural Sciences, Hohhot, China. [3]College of Animal Science, Inner Mongolia Agricultural University, Hohhot, China. [4]State Key Laboratory of Herbage Improvement and Grassland Agro-ecosystems, College of Pastoral Agriculture Science and Technology, Lanzhou University, Lanzhou, China. [5]Institute of Animal Science, NingXia Academy of Agriculture and Forestry Sciences, Yinchuan, China. [6]College of Animal Science and Veterinary Medicine, Shenyang Agricultural University, Shenyang, China. [7]Modern Agricultural Production Base Construction Engineering Center of Liaoning Province, Liaoyang, China. [8]Liaoning Province Liaoning Cashmere Goat Original Breeding Farm Co., Ltd., Liaoyang, China. [9]Shandong Binzhou Academy of Animal Science and Veterinary Medicine, Binzhou, China. [10]Gannan Institute of Animal Husbandry Science, Hezuo, China. [11]Beijing Lvyeqingchuan Zoo Co., Ltd., Beijing, China. [12]Zhongwei Goat Breeding Center of Ningxia Province, Zhongwei, China. [13]Jiaxiang Animal Husbandry and Veterinary Development Center, Jining, China. [14]College of Grassland Science and Technology, China Agricultural University, Beijing, China. [15]These authors contributed equally: Hui Wu, Ling-Yun Luo, Ya-Hui Zhang, Chong-Yan Zhang. ✉e-mail: shangang.jia@cau.edu.cn; liuzh7799@163.com; menghua.li@cau.edu.cn

