## [Transparent Peer Review file · Nature Communications]

Telomere-to-telomere genome assembly of a male goat reveals variants associated with cashmere traits

Corresponding Author: Professor Meng-Hua Li

Version 0:

Reviewer comments:

Reviewer #1

(Remarks to the Author)

This manuscript from Wu and colleagues is a well-written account of the generation and use of a telomere-to-telomere (T2T) assembly of a male cashmere goat's genome sequence.

This assembly represents the most complete sequence of a goat genome to date. Thus, this genome assembly will be an invaluable resource for the goat genetics and genomics research community and more widely.

The assembly and underlying sequence data have been lodged in the Chinese National Genomics Data Center. However, the completeness of these data cannot be verified as the data have not yet been released.

The use of this T2T goat genome sequence would be greater if the assembly and sequence data were also lodged in the NCBI/EBI/DDBJ databases.

The laboratory and data analysis methodologies are appropriate.
The conclusions are supported by the data and figures provided.

The biggest question concerns the failure to generate haplotype resolved assemblies of the autosomes. There is no explanation as to why such haplotype resolved assemblies were not generated - hifiasm is capable of generating such assemblies. It appears, therefore that the T2T-goat1.0 may comprise pseudohaploid assemblies of each autosome. This issue is not addressed in the manuscript.

No explanation is provided for the assignment of the assembled sequences to named autosomes.

There are a few minor issues to address:

Supplementary Figure 2. The gaps were not 'generated' by the scaffolding. Rather the gaps were either bridged by the scaffolding.

A table with the number of contigs and scaffolds through the progress of the assembly process would be useful perhaps with a figure showing the strategy from GV1 to GV5.

Figure 1b - why not orient all the chromosomes with the centromere to the left.

Line 107-108. Whilst figure 1b shows that the assembled chromosomes are all acrocentric (except chr Y), the figure does not show that assembling the short arm and telomeres of the acrocentric chromosomes is challenging.

Supplementary figure 3. Whilst the green colored centromeres for most chromosomes can be easily seen as distinct from the chromosome arms, for several chromosomes such as Chr20, ChrX, ChrY it is difficult to see the boundary.

Line 112. The CHInn nomenclature for the chromosomes has been used before line 112. The expansion of the abbreviation should be provided at the first use.

Lines 145-146 and Supplementary Figure 4. The color scheme for supplementary figure 4 is not explained in the figure legend. Although there is overall good agreement between the assembled sequence and the optical maps, the few differences are not explained.

Line 155. In classical genetics synteny describes the co-localization of genetic loci on the same chromosome within an individual or species. Through misuse the term has now come to be used as a shorthand for conservation of synteny between species. Perhaps it would be better to start this sentence with "A high degree of co-linearity was observed between...."

Lines 174 onwards. What is the evidence that the inversions relative to ARS1 are assembly errors in ARS1 rather than structural variant differences between the two individuals. It is not clear from the text whether the junction sites in the arsl assembly were checked with sequence data from the ARS1 individual.

Line 325. It was evidently possible to identify paternal-specific ONT reads from which to assemble ChrY. Thus, it should have been possible to identify paternal and maternal long read data (both PacBio and ONT) that could have been used to assemble haplotype resolved chromosomes (autosomes as well as X and Y).

Line 377. Fig 3d should be fig 3c

Reviewer #2

(Remarks to the Author)

The author has constructed a high-quality, gap-free goat genome utilizing the latest sequencing technologies. Remarkably, while filling complex regions of the autosomes, a more complete sequence of chrX and chrY was also obtained. It is evident that this genome contributes significantly to molecular studies of goats. However, there are several areas in the article that require further clarification:

1. Generally, genome assemblies generated using HiFi reads require minimal polishing, as excessive polishing may introduce potential errors. Please utilize second-generation reads from this individual to identify homozygous SNPs through alignment and assess the single-base error rate of the assembly based on their percentage. If possible, evaluate the effect of multiple polishing rounds on genome improvement.

2. In Line 160, a significant proportion of the PURs region consists of SD regions. Please provide a detailed description of the criteria used for determination. Additionally, SD regions may also potentially be the assembly errors due to heterozygosity. Have measures been taken to distinguish and exclude them?

3. In Line 257, the SD (more accurately described as random duplication) shown in supplementary figure 11 is also worthy of validation to confirm its continuity as a genuine sequence rather than heterozygous sequence.

4. In supplementary figure 14, the chrX of T2Tgoat1.0 appears shorter than that of Saanen_v1, and there is a substantial difference in chrX length among the three assembly versions. Is this a genuine difference, or is it merely a representation error?

5. In Line 353, the discussion regarding the testis-specific protein Y-encoded (TSPY) and heat shock transcription factor Y-linked (HSFY) genes is intriguing. Considering the significant differences in their numbers compared to the ARS1 assembly and the human genome, further validation is necessary to confirm their authenticity.

Reviewer #3

(Remarks to the Author)

The authors report construction of a gapless T2T assembly for a goat and applied this as a reference for detection of structural variants and SNP, by mapping HiFi data generated for one animal from each of five other sheep breeds. Selective sweeps were detected using either the T2T-goat1.0 assembly or the existing RefSeq assembly ARS1. A population genetic analysis was performed with public short read data from 495 goats among 11 populations to identify regions associated with cashmere trait.

In general the authors provide convincing evidence that the assembly represents the complete genome of a goat, one of the first mammals and particularly livestock species to reach this goal. This is notable as the goat was also the first mammalian assembly (the ARS1 reference) to reach "gold" status in 2016 based on a combination of long read, short read, and chromatin conformation data.

The authors note extensive regions of additional genome (primarily heterochromatin) along with additional genes not included in the ARS1 reference. The author's approach was unusual for T2T assembly in that they used "merged" assembly as a first step rather than constructing the parent-specific kmer database and performing haplotype-resolved contig assembly. As such, the final haploid assembly does not represent either parental haplotype in its entirety. The authors do not provide hamming error rate or switching error rate so the extent to which this might lead to erroneous expansion or contraction is difficult to assess, and the depiction of the assembly graph in Supplementary Figure 3 is not colored by parental kmers. The authors did perform some appropriate quality measures, although the demonstration of this in the Figures is difficult to assess due to their resolution (a common difficulty in presenting complete genomes). For example, mapping the ONT reads to the assembly is a good step, but the resolution of Figure 1b renders the demonstration rather

meaningless. Nothing less than a drop or increase in coverage spanning many kilobases to megabases would be visible at this level of resolution to detect any issues. In addition, Figure 1b alternates the acrocentric chromosomes to have the beginning at the start or end of the chromosome—this may reflect error in the ARS1 assembly but seems like a good place to fix this. Generally, acrocentric chromosomes are presented with the short arm/telocentric arm as the proximal (leftmost or top) of the representation or sequence.

The method for assembly is well documented and it is likely that if the primary data is available it could be replicated. The manuscript indicates public release of the data through the National Genomics Data Center in China, but I was unable to access it. Unfortunately, this is not an uncommon occurrence when attempting access from the U.S. The manuscript has very extensive characterization of the Y chromosome in particular, along with centromere analysis of autosomes and sex chromosomes, plus the association analyses. The total data is impressive but hard to get through when so much is packed into a single manuscript, making it very long and complicated. For contrast, the human T2T description had these features in separate companion manuscripts that could each delve into specific aspects.

The selective sweep analysis is a reasonable first approximation based on selective sweep analysis of cashmere and non-cashmere animals, although it could have been confounded by other breed effects since the study was not conducted in a population segregating the trait. Rather, the study uses animals of cashmere breed(s) compared to non-cashmere goat populations.

Figures : The detailed and extensive research includes 7 primary figures summing to 43 panels and 23 supplementary figures summing to 42 panels, for a total of 85 panels. This density of information made it quite slow to adequately review, in addition to causing difficulty in interpretation of some figures due to the resolution. As mentioned above, Figure 1b provides too low resolution to be of value and could be moved to supplementary or better described to note a region of a given minimum length showing coverage below a threshold (e.g. “the coverage across all chromosomes did not fall below X% of average for any region exceeding Y bases”). Figure 1c has very low information content as the assembly size by year of release does not seem to have a lot of bearing on the conclusions presented; this panel could be deleted. Figure 2a and 2b show redundant information and 2a has additional useful illustration so suggest 2b could be deleted. The example of support for correcting ARS1 in panel 2d could be moved to a supplemental figure as it simply illustrates one of the many differences. The resolution in figure 2d is too low to be able to interpret much from it and could be removed, in addition panel 2f demonstrates collinearity with another improved X chromosome making the alignment in 2e less important. The tiny blob plots of figure 2g could easily be put into larger supplementary figure and line 194 is sufficiently descriptive of the result without the visual. Panel 2h could be fully described by quoting statistics (average and range) for the five breeds in the long read mapping effort, the panel is not illuminating and also will end up very small print at the scale shown in the full-page figure.

The legend of Figure 3 nor the text indicates what chromosome is being shown in panel 3a, but the ChIP-seq is a bit hard to interpret with possibly the region between 4 and 9 Mb being enriched (but not more than the signal of 0-1.4 Mb?). The “methylation dip” associated with the centromere would appear to be at about 3 and 8 Mb suggesting the centromere should lie in between, but the ONT data does not show much agreement with this. The text is unclear on the author’s interpretations of these results, as line 308-9 states that a dip is observed coincident with ChIP-seq signals, but the figure doesn’t seem to show that. Perhaps annotation of the figure as to where exactly the authors think the centromere being illustrated is supposed to be?

Figure 4b and 4c could be moved to supplemental since they are only used to illustrate improvement of the assembly over two earlier assemblies and not to make any important points about the goat Y chromosome. Similarly, Figure 5a, 5b, and 5c show the improvement when using a complete genome over the previous assemblies which can be easily capture by a few statistics in one table and move the charts to supplemental. Figure 5d is very difficult to interpret since the resolution is too low for meaningful analysis. It is not clear that Figure 5e is necessary for the point being made in the text, that the keratin filament pathway was the GO term with most significance and number of genes enriched, so this could also be supplemental figure or removed.

Line 121 I suggest that “short read sequence data” is better than “next-generation sequencing data”

Line 163 It is not clear what utility there may be in using PUR and NAR as acronyms for what appear to be identical features? Or at least it is not clear what the difference is from the text. Are NARs a subset of PURs? If so what basis is there for the subset?

Line 167 What does “maximum QV” refer to? Is that the maximum average QV value for a chromosome?

Line 168 The description of MUK is inadequate—the authors define it as “a k-mer present only once in the genome”, but how is that a minimum unique k-mer? What meaning does a sliding 100kb window of kmer have exactly? If it is “unique” k-mer, then there shouldn’t be large regions of genome as depicted (at very low resolution) in the centromeres?

Line 174 Delete “Additionally”

Line 191 Delete “Furthermore”

Line 192 More informative would be “Short read data from previous sequencing efforts focused on global populations of domestic and wild goats was retrieved from public databases (Supplementary Table 3).” This avoids any impression that the present study generated this data.

Line 195 “improved the number of mapped reads significantly” may mean “substantially”, unless there was an actual significance test performed in which case a p-value should be provided.

Line 213 There is no mention of why chromosomes 12 and 19 are illustrated, are they meant to be representative in some way or are they special cases that some feature should be pointed out?

Line 228 “Among these genes” is vague, does this refer to all protein-coding genes or BUSCO genes?

Line 234 Why would one think that newly assembled genes would be enriched in pathways? If you are thinking that a general principle is that genes lying in repetitive regions or segmental duplications share pathways, is there a reference to

support this?

Line 236 delete "Additionally"

Line 237 add reference to the supplementary figure with tangle in string graph

Line 241 (and other lines) It is not established jargon to use "left" and "right" in relation to position on chromosomes, generally the usage is "proximal" or "distal" and the reference point is the telomere of the q arm (or telomere at the centromeric end for acrocentric) as "proximal"

Line 244 I think you mean "located on opposite chromosomal ends", not that they can be on either one end or the other.

Line 313 CNEP-A should be CENP-A

Line 370 "extended our search for repeat sequences on the sheep Y chromosome." I think this needs some context, e.g. "extended our search for repeat sequences by reference to repeat sequences found on the sheep Y chromosome in an assembly of a sheep genome." Also, a reference and accession for this T2T-sheep1.0 assembly should be provided.

Line 374 The CenY symbol and tight "peak" in Figure 4a does not appear to be at a border of a hypermethylated region? Why is there no track for CenP-A for Y chromosome, it presumably uses CenP-A just like the other chromosomes?

Line 380 a difference of 0.11% higher mapping rate seems pretty small, are these numbers correct? It would seem the ARS1 would be relatively worse?

Line 385 why use female goats? This comes right after making a major point about the improvement of the Y chromosome. A brief explanation would be pertinent.

Line 388 If reporting the average of five data sets, a range would be useful

Line 416 Should this be "covered" exons or "overlapped" exons? (it could be "covered" is correct, just wanted to check that is accurate)

Line 427 The supplementary table lists references for data used in the manuscript, but these should be included in the reference list not just in a supplementary table.

Line 445 This analysis doesn't seem to have much meaning in my view. It is mainly a function of the distribution of identified goat QTL rather than a reflection of the likelihood that PUR contain tightly linked SNP that would explain more variation than previously identified SNP.

Line 449 remove "Additionally". Also, how did these overlap with long read SVs?

Lines 455-467 It is not clear why it is surprising or important that SNP identified with this reference can be used in the same way as the SNP identified with the old reference? I believe this recapitulates previously published relationships and doesn't add anything new, at least nothing the authors remark upon. Thus, it would be more briefly summarized without the need for Figure 6c or 6d. This contrasts with the next section about selective sweep definition being influenced by the new assembly which is new.

Line 457 NJ tree wasn't "used to divide" rather "A neighbor-joining (NJ) tree distinguished two major branches corresponding to wild and domestic goats".

Line 464 I think it would be clearer to state "LD decay rate in wild goats was slower relative to domestic goats, with the East Asian..."

Line 474 "commonly found in ARS1" should be "found in common with ARS1"

Line 475 "previously reported genes" should have references

Line 482 why is this remarkable? Suggest just removing that word and stating that "the presence of selective signal was supported by overlapping signals by the π ratio statistic.

Line 492 "investigated the domestication of SVs", how does one domesticate an SV? Maybe this should be "We further investigated SVs found in domestication-associated genomic regions in bezoars and domesticated goat populations."

Line 496 The list of genes is far shorter than 63, so maybe should say "including" or "for example" for the genes that are selected to be presented. Otherwise, why are these particular genes shown?

Line 500 not clear why one would fail to detect Fst when mapping to the new assembly vs ARS1 for "group 2" genes?

Apparently the genes are present in ARS1 so why would using that as reference fail to identify them?

Line 517 mildly odd that the number of selective signals is nearly identical to those for domestication, a bit surprising I guess (no change needed, just noting this)

Line 524 It is not clear why one would find so much genomic region in selective sweep only when using ARS1 as reference, if the hypothesis is that regions missing in that assembly are a primary source of novel selection detection. If the analysis is this sensitive to reference genome, one wonders how reliable any of the identified regions might be? The results suggest that selective sweep analysis may not be completely substituting for a GWAS in a population segregating the trait, even if previously identified genes fall in some of those regions.

Line 531 allele frequency differences between cashmere/non-cashmere populations can be breed effects not necessarily indicative of association with the trait, as population stratification is inherent in the approach taken.

Line 554 neither the cattle nor gerbil genomes referenced are complete seamless mammal genomes. The gerbil assembly has complete centromeres but was not purported to be T2T. There are some non-human primate assemblies that are T2T status in the NCBI database but so far no publications that I am aware of. I believe that a mouse T2T assembly has been published but didn't have time to search for the reference. Might be safer to say that this assembly is one of the first mammal genomes to achieve complete, seamless assembly following the first human T2T assembly and use the Science reference only. For example, there are plenty of assemblies of cattle published that are as good as the one mentioned in reference 45, also many other species have assemblies of similar quality.

Line 569 I don't understand this statement since none of the three references report T2T genome assemblies.

Line 571 It isn't clear why the authors originally used a merged assembly method when parental sequence was available. In general of late, the data types reportedly used would likely have assembled the Y chromosome just using the trio approach in hifiasm that they employed in merged mode instead of trio mode; certainly likely would have resolved the PAR regions of X and Y chromosome.

Line 581 I think this may be a misstatement, it is unlikely that there aren't multicopy protein-coding gene families elsewhere in the genome, perhaps this was meant to say "found on the Y chromosome"? Certainly the immune gene clusters would count as multicopy protein coding gene families for example.

Methods :

Line 643 "CTAB method" needs reference.

Line 644 "manufacturer's"

Line 991-993 The BioProject indicates "will be available on 2026-01-14"

Line 997 the link for software at github returns "page not found"

Version 1:

Reviewer comments:

Reviewer #2

(Remarks to the Author)

The explanations are relatively complete, and the entire process is clearer and more credible. It is foreseeable that this genome assembly will become an important reference sequence in goat molecular research. Additionally, I apologize for mistakenly writing "tandem duplication" as "random duplication" in the previous round of review, which might have caused confusion. However, you provided a convincing response. Overall, this is a rigorous and meaningful work.

Reviewer #3

(Remarks to the Author)

The authors have responded appropriately to all reviewer comments in my view. The few rebuttals appear to represent differences of opinion not problems with the manuscript, e.g. the value of selective sweep versus GWAS analysis. The manuscript appears better focused and more readable with the reduction in figures although there are still 30 figure panels in the main body.

RESPONSE TO REVIEWERS' COMMENTS

Reviewer #1 (Remarks to the Author):

This manuscript from Wu and colleagues is a well-written account of the generation and use of a telomere-to-telomere (T2T) assembly of a male cashmere goat's genome sequence. This assembly represents the most complete sequence of a goat genome to date. Thus, this genome assembly will be an invaluable resource for the goat genetics and genomics research community and more widely.

[1.1] The assembly and underlying sequence data have been lodged in the Chinese National Genomics Data Center. However, the completeness of these data cannot be verified as the data have not yet been released. The use of this T2T goat genome sequence would be greater if the assembly and sequence data were also lodged in the NCBI/EBI/DDBJ databases.

– In addition to the Chinese National Genomics Data Center (CNGDC), we have also uploaded the genome assemblies (the merged T2T goat genome T2T-goat1.0 and the two haplotype genome assemblies T2T-goat1.0P and T2T-goat1.0M) and raw data to NCBI (<https://www.ncbi.nlm.nih.gov>) with the BioProject number PRJNA1062519. Following your suggestion below (please see the comment [1.6]), we ordered the acrocentric chromosomes with all the centromeres on the leftmost end, and generated and deposited the version 2.0, i.e., T2T-goat2.0, in the public database NCBI. Also, we deposited the haplotype-resolved maternal and paternal assemblies in NCBI. A total of six genome assemblies and raw sequencing data generated in this study could be accessed from both the CNGDC and NCBI databases before the submission of the revision, and the accessions are included in Data Availability in the revision, as shown as follows.

“The genome assemblies (T2T-goat1.0, T2T-goat1.0P, and T2T-goat1.0M; and T2T-goat2.0, T2T-goat2.0P, and T2T-goat2.0M) and raw sequencing data generated in this study, including PacBio HiFi data, ultralong ONT data, MGI data, Iso-seq data and ChIP-seq data, can be achieved from National Genomics Data Center (<https://ngdc.cncb.ac.cn/>) with BioProject number PRJCA022847, and NCBI with accession number PRJNA1062519. T2T-goat2.0 is the update of T2T-goat1.0, with all the centromeres placed on the leftmost ends of acrocentric autosomes.”

[1.2] The laboratory and data analysis methodologies are appropriate. The conclusions are supported by the data and figures provided. The biggest question concerns the failure to generate haplotype resolved assemblies of the autosomes. There is no explanation as to why such haplotype resolved assemblies were not generated - hifiasm is capable of generating such assemblies. It appears, therefore that the T2T-goat1.0 may comprise pseudohaploid assemblies of each autosome. This issue is not addressed in the manuscript.

– Good comment! We generated the haplotypic assemblies based on sequences of the parents and ONT/HiFi reads using the Hifiasm trio mode. We repeated the pipelines of

T2T-goat1.0 by implementing scaffolding, gap-filling and polishing for these two haplotypic assemblies. Finally, we generated T2T-goat1.0P (paternal origin) and T2T-goat1.0M (maternal origin), whose average QV reach to 53.06 and 52.37, respectively. We added the description and summary tables and figures in Methods and Results of the revision (See lines 128-143, 166-177, and 846-854). We also released these haplotypic assemblies and the summary statistics of them in the public databases as shown in the response above.

– The method for haplotype genome assemblies (lines 846-854):

“Haplotype genome assembly

The initial assembly for the autosomes of the haplotype-resolved genomes (T2T-goat1.0P and T2T-goat1.0M) was performed by using Hifiasm (v0.16.1 r375) based on the trio mode. The scaffolding, gap filling and polishing for T2T-goat1.0P and T2T-goat1.0M were performed with the similar method for T2T-goat1.0. The ONT and HiFi reads were binned for paternal and maternal origin based on the k -mer ($k = 21$) generated from parental short reads, which is the same to the trio-binning strategy as shown for Y chromosome assembly. The binned long reads were used to fill gaps and polish the haplotype genomes by using Nextpolish2 (v0.1.0), rather than the NGS data, to avoid the introduction of the other haplotype sequences.”

– The result description for haplotype genome assembly (lines 128-143):

“Based on long reads from the assembled buck and short read sequence data from its parents, the haplotype assemblies were generated by using Hifiasm¹⁴ assembler based on the trio mode, for T2T-goat1.0P of paternal origin with complete Y chromosome and T2T-goat1.0M of maternal origin with complete X chromosome respectively (Table 1). Parent-specific k -mers based on short reads were used to pick ONT and HiFi reads of paternal and maternal origin, to fill 31 gaps for T2T-goat1.0P and 25 gaps for T2T-goat1.0M. Y chromosome was also independently assembled with the trio-binned paternal ONT reads, and improved for scaffolding, gap filling and correction, with the assistance of Y chromosomal contigs *de novo* assembled from HiFi reads. As a result, we generated a complete Y chromosome with two telomeres spanning 20.96 Mb. The final T2T genome assembly, T2T-goat1.0, covering all the 29 autosomes and X and Y chromosomes, was obtained with a total length of 2.86 Gb (Table 1). This size was slightly longer than the estimated genome size of 2.76 Gb obtained using the NGS short reads. After final polishing, the chromosomes X and Y from T2T-goat1.0 were combined with the haplotype-resolved autosomes to obtain the paternal and maternal genome assemblies T2T-goat1.0P (2.71 Gb with N50 103.46 Mb) and T2T-goat1.0M (2.81 Gb with N50 109.45 Mb) respectively.”

– Lines 166-177:

“Meanwhile, T2T-goat1.0P and T2T-goat1.0M achieved the T2T level assembly, with the whole genome QV 53.06 and 52.37 respectively. The challenges for assembly of acrocentric centromeres and lack of binned long reads still left 11 gaps in the centromeric regions of 9 chromosomes in T2T-goat1.0P and 6 gaps in the centromeric

regions of 6 chromosomes in *T2T-goat1.0M* (Supplemental Table 4). Except for the centromeric regions, the genomes have relatively uniform HiFi and ONT coverage. We estimated the switching errors and obtained 0.036%, 0.016%, and 0.011% for the three T2T assemblies *T2T-goat1.0*, *T2T-goat1.0P*, and *T2T-goat1.0M* respectively. Furthermore, by comparing autosomes between the two haplotypes, we detected 5,287,863 single nucleotide variants (SNVs), 726,939 small insertions and deletions (< 50 bp), and 22689 SVs (\geq 50 bp). Considering the completeness and quality, *T2T-goat1.0* represented a better assembly (Supplementary Fig. 4a) for the downstream analysis compared to the two haplotypes.”

[1.3] No explanation is provided for the assignment of the assembled sequences to named autosomes.

– *We ordered and placed the contigs into the pseudo-chromosomes at the step of Hi-C analysis. Also, we named the autosomes based on their correspondence with those of ARS1. We modified the description of Hi-C for chromosomal assignment Please see below and also lines 104-109.*

“Using Hi-C data, the contigs/scaffolds were anchored and oriented onto 30 pseudochromosomes corresponding to 29 autosomes and the X chromosome of the GV3 assembly (Supplementary Fig. 2), with 145 gaps and an average of ~4.8 gaps per pseudochromosome (Supplementary Fig. 3). We named all the autosomes (CHI1 for *C. hircus* chromosome 1, and the other 28 chromosomes CHI2–CHI29) based on the sequence similarity to those of the NCBI goat genome reference *ARS1* (GCF_001704415.1).”

There are a few minor issues to address:

[1.4] Supplementary Figure 2. The gaps were not 'generated' by the scaffolding. Rather the gaps were either bridged by the scaffolding.

– *Corrected. Please see below and lines 1513-1514 in the legend to Supplementary Figure 2.*

“The green stars indicate the gaps bridged by the Hi-C scaffolding.”

[1.5] A table with the number of contigs and scaffolds through the progress of the assembly process would be useful perhaps with a figure showing the strategy from GV1 to GV5.

– *To get the final genome assembly T2T-goat1.0, we built the assemblies of GV1 to GV5. We have added Supplementary Table 2 in the revision showing a summary of contigs and scaffolds for these GV1 ~ GV5, and added a new figure Supplementary Fig. 1 to present the assembly strategy with a flowchart. Please see Supplementary Table 2 and Supplementary Fig. 3 (see below also) in the revised version.*

Supplementary Fig. 1 Genome assembly strategy for *T2T-goat1.0*. GV1~GV5 assemblies were created for *T2T-goat1.0*. GV1 is the initial assembly based on HiFi reads using Hifiasm software. Bionano scaffolding was performed to generate GV2, and Hi-C placed the contigs/scaffold onto the pseudochromosomes for GV3. Gaps were filled using ONT ultralong reads for GV4. Telomeres and chromosome Y independently assembled were added to GV5, which was polished for high-quality of *T2T-goat1.0*.

[1.6] Figure 1b - why not orient all the chromosomes with the centromere to the left.
 – *In order to be consistent with ARS1 and make a comparison between the two assemblies, we kept the same chromosome orientations in T2T-goat1.0 as ARS1. Thus, the centromeres were randomly located either in the left or right ends of all acrocentric autosomes. The centromere positions of all the chromosomes are shown in Supplementary Fig. 4 according to their chromosomal orientations.*

Following your suggestion, we provided an updated version of the goat T2T genome assembly, namely T2T-goat2.0. T2T-goat2.0 placed all the centromeres on the leftmost ends of all the chromosomes, based on the chromosomal sequences of T2T-goat1.0. Both the six assemblies of the two versions are available in the CNGDC (accession PRJCA022847) and NCBI databases (accession PRJNA1062519), as shown in the response [1.1].

[1.7] Line 107-108. Whilst figure 1b shows that the assembled chromosomes are all acrocentric (except chr Y), the figure does not show that assembling the short arm and telomeres of the acrocentric chromosomes is challenging.

– *Good comment again! The assembly of acrocentric chromosomes is challenging, because centromeric regions contained similar or identical centromeric repeats, which are mainly dominated by satellites including SatI, SatII, and SatIII. The similarity of*

sequences in the centromeric region of acrocentric chromosomes had failed the complete assembly in the previous assemblies. As a result, all the previous goat assemblies did not show the complete centromeric regions. As suggested, we added the graph assembly string with a tangle of centromeric regions among four chromosomes (CHI12, CHI13, CHI19 and CHI22) in the updated Fig. 1b in the revision, to show the assembling challenges regarding to assembling centromeric regions in acrocentric chromosomes. Please see below and lines 109-115.

“Our assembly results showed that all the autosomes and the X chromosome were acrocentric (Supplementary Fig. 4), causing great challenges in the complete chromosomal end assembly. For example, a tangle of centromeric regions among four chromosomes, CHI12, CHI13, CHI19 and CHI22, was shown in the assembly graph string (Fig. 1b), and our assembly resolved the centromeric regions for these four chromosomes, including CHI12 and CHI19 (Fig. 1c and Fig. 1d).”

Fig. 1 Goat T2T genome assembly with 29 autosomes and chromosomes X and Y.

a, Genome assembly strategy for *T2T-goat1.0* and its haplotype genomes *T2T-goat1.0P* and *T2T-goat1.0M* of a buck. Trio-binning assemblies for the autosomes of *T2T-goat1.0P* and *T2T-goat1.0M* and Y chromosome was performed based on long reads of the buck and MGI short reads from its parents. **b**, The assembly graph string shows a tangle among the four chromosomes, Chr12 (blue line), Chr13 (green line), Chr19 (red line) and Chr22 (brown line), due to the high similarities of centromeric sequences in gray. The centromeric regions in the assembly tangle are enlarged and shown in the right panel. **c, d**, Genome features of Chr12 and Chr19 in *T2T-goat1.0*. The assembly graph tangle involving Chr12, Chr13, Chr19 and Chr22 was resolved in *T2T-goat1.0*, and Chr12 and Chr19 are selected to exhibit the completeness and features across the whole chromosome. The following information is provided from top to bottom: the gene density (red), the density of LINEs and SINEs (orange), the

satellite density (green), the TE density (blue), error k -mer ($k=21$, purple), and the minimum unique k -mer (MUK) per 100 kb. The more MUK values indicate more repetitive sequences in a 100-kb window, and more yellow and green colors indicate the presence of the centromeric regions. All the features are shown in 10-kb windows, except for MUK.

[1.8] Supplementary figure 3. Whilst the green colored centromeres for most chromosomes can be easily seen as distinct from the chromosome arms, for several chromosomes such as Chr20, ChrX, ChrY it is difficult to see the boundary.

– We replotted the assembly graph string in Supplementary Fig. 3a, which is updated as Supplementary Fig. 4a in the revision, and colored the centromeric regions in black lines. Now the boundaries on all the chromosomes could be viewed clearly, as shown below.

Supplementary Fig. 4a Assembly graph strings for the chromosomes. The chromosomes are colored in red for maternal *k*-mers, in blue for paternal *k*-mers, and in gray for no parental specific *k*-mers. The centromeric regions of the four chromosomes, Chr12, Chr13, Chr19 and Chr22, are entangled, and further enlarged and shown with centromeric regions highlighted in Fig. 1b. Another tangle related to 45S rDNA repetitive sequences (yellow lines in bottom left box) involve five chromosomes, Chr02, Chr03, Chr04, Chr05, and Chr28. The pseudoautosomal region (PAR) is shared between ChrX (two fragments in red) and ChrY (in blue), which are in the bottom right box.

[1.9] Line 112. The CHInn nomenclature for the chromosomes has been used before line 112. The expansion of the abbreviation should be provided at the first use.

– *CHInn refers to C. hircus chromosome number. We added the explanation of CHInn at the location where it is firstly used. Please see below and lines 107-109.*

“We named all the autosomes (CHI1 for *C. hircus* chromosome 1, and the other 28 chromosomes CHI2–CHI29) based on the sequence similarity to those of the NCBI goat genome reference *ARS1* (GCF_001704415.1).”

[1.10] Lines 145-146 and Supplementary Figure 4. The color scheme for supplementary figure 4 is not explained in the figure legend. Although there is overall good agreement between the assembled sequence and the optical maps, the few differences are not explained.

– *We showed the alignment of Bionano optical maps against in-silico maps of 31 chromosomes in T2T-goat1.0 in Supplementary Fig. 4 in the original submission (Supplementary Fig. 7 in the revision). As commented here, it showed that the Bionano alignments are overall good agreement between T2T-goat1.0 and Bionano optical maps. However, some differences are observed, especially for the green and yellow regions at the ends of the chromosomes, i.e., the centromeric regions. We modified the description as followed below. Please see also lines 161-164.*

“The alignments of the Bionano optical maps also supported the accurate assembly of *T2T-goat1.0* (Supplementary Fig. 7), and a few differences between Bionano optical maps and *T2T-goat1.0* mostly fell into the centromeric regions.”

– *We added more details and updated the legend of Supplementary Fig. 7 for explaining the differences in green and yellow colors. Please see below and also lines 1536-1548.*

“Supplementary Fig. 7 Bionano alignments against *in-silico* maps of 31 chromosomes in *T2T-goat1.0*. The top bars represent the *T2T-goat1.0* chromosomes, and Bionano optical maps were aligned below. Blue color indicates the regions with consistencies and co-linearity is represented by grey lines between the optical maps of *T2T-goat1.0* chromosomes and Bionano hybrid assembly. Meanwhile, yellow regions represent breaks in collinearity between *T2T-goat1.0* chromosomes and Bionano hybrid

assembly, while green regions without enzyme labeling sites are the stretches of unique sequences in *T2T-goat1.0* without co-linearity against Bionano maps below. Light blue regions in Bionano maps for hybrid assembly below indicate the stretches of unique sequences without enzyme labeling sites. The centromeric regions are enriched with repeats, and lack of enzyme labeling sites for Bionano. Therefore, green and yellow regions are found mostly in the ends of the acrocentric chromosomes, which indicate the locations of centromeres. So are green and yellow regions in the pseudoautosomal region (PAR) of chromosome Y.”

[1.11] Line 155. In classical genetics synteny describes the co-localization of genetic loci on the same chromosome within a an individual or species. Through misuse the term has now come to be used as a shorthand for conservation of synteny between species. Perhaps it would be better to start this sentence with "A high degree of co-linearity was observed between..."

– *We have changed to “A high degree of collinearity was observed between...”. Please see also line 184.*

[1.12] Lines 174 onwards. What is the evidence that the inversions relative to ARS1 are assembly errors in ARS1 rather than structural variant differences between the two individuals. It is not clear from the text whether the junction sites in the *ars1* assembly were checked with sequence data from the ARS1 individual.

– *The inversion relative to ARS1 is indeed the assembly errors in ARS1, rather than structural variations between the two assemblies. We have downloaded the long reads sequenced for the ARS1 individual (NCBI accession no. PRJNA340281) and aligned them to the ARS1 reference genome. We found that these two junction sites of the inversion are not covered by the ARS1 individual’s own reads, but the corresponding sites of T2T-goat1.0 could be mapped well by the long reads of both ARS1 and T2T-goat1.0, thereby proving that the inversions are not structural variations but assembly errors in ARS1. According to your comments, we have modified the description in the revision, and moved this figure with modifications to the supplementary materials as Supplementary Fig. 9 with the updated the figure legend. Please below and lines 208-214 and lines 1557-1563.*

– *The legend in Supplementary Fig. 9 in the revision is updated as follows. Please see also lines 1565-1571.*

Supplementary Fig. 9 An inversion error on chromosome 18 of *ARS1*. The two junction sites of an inversion on chromosome 18 of *ARS1* could not be covered by the PacBio reads sequenced for the *T2T-goat1.0* and *ARS1* individual (NCBI accession no. PRJNA340281), with the evidences of the clipped reads for alignments to *ARS1* visualized with IGV. The corresponding sites in *T2T-goat1.0* were assembled correctly with even coverage of long-read from both *ARS1* and *T2T-goat1.0*, and the accurate assembly of this region is further supported by Bionano alignment.

– The result description is updated as follows (lines 208-214 in the revision):

“*T2T-goat1.0* corrected abundant structural errors in *ARS1* (Fig. 2a), such as, an inversion on the q arm of CHI18 (CHI18: 19 Mb–26 Mb) in *ARS1* compared to *T2T-goat1.0*. We examined the alignments of the long read sequence data from *ARS1* goat individual (NCBI BioSample ID SAMN03863711) against the inversion regions of the two genome assemblies. We found that the two junction sites of this inversion in *ARS1* were not covered by any long reads sequenced from both *ARS1* and *T2T-goat1.0*, but the failed alignment of these reads at the junction sites is rescued by the corresponding region of *T2T-goat1.0* (Supplementary Fig. 9).”

[1.13] Line 325. It was evidently possible to identify paternal-specific ONT reads from which to assemble ChrY. Thus, it should have been possible to identify paternal and maternal long read data (both PacBio and ONT) that could have been used to assemble haplotype resolved chromosomes (autosomes as well as X and Y).

– *Following your comments, we added haplotype-resolved assemblies T2T-goat1.0P and T2T-goat1.0M, in addition to the merged one T2T-goat1.0. We performed haplotype assembly by using HiFiasm based on the trio mode. We repeated the whole assembly steps for them, including scaffolding, gap filling, polishing, and genome annotation for T2T-goat1.0P and T2T-goat1.0M. We identified maternal- and paternal-specific ONT and HiFi long reads based on the short reads from the parents and their k-mers, to implement gap filling and polishing. In the revision, we released the three genome assemblies, including merged one T2T-goat1.0, maternal one T2T-goat1.0M and paternal one T2T-goat1.0P, in both CNGDC (accession PRJCA022847) and NCBI databases (accession PRJNA1062519). We added the approach in the Method section and descriptions in Results. Please see the responses [1.1] and [1.2] and lines 128-143, 166-177, and 846-854.*

[1.14] Line 377. Fig 3d should be fig 3c

– *Corrected.*

Reviewer #2 (Remarks to the Author):

The author has constructed a high-quality, gap-free goat genome utilizing the latest sequencing technologies. Remarkably, while filling complex regions of the autosomes, a more complete sequence of chrX and chrY was also obtained. It is evident that this genome contributes significantly to molecular studies of goats. However, there are several areas in the article that require further clarification:

[2.1] Generally, genome assemblies generated using HiFi reads require minimal polishing, as excessive polishing may introduce potential errors. Please utilize second-generation reads from this individual to identify homozygous SNPs through alignment and assess the single-base error rate of the assembly based on their percentage. If possible, evaluate the effect of multiple polishing rounds on genome improvement.

– *Good comment indeed! In fact, our polishing process that is developed using Nextpolish² and Nextpolish2³ (<https://github.com/Wuhui2024/CAU-T2T-Goat>) contain the two major steps. The first one only focused on the low-quality regions and involved 5 rounds of polishing using HiFi (2 rounds), ONT (1 rounds), and MGI (2 rounds) reads. After polishing the low-quality regions, the second step worked on the whole genome with only one round of HiFi polishing. Consequently, the dominant high-quality genomic regions only experienced 2 rounds of HiFi polishing, and the low-quality regions received 6 rounds of polishing. In contrast, it is common with multiple rounds of polishing in the previous T2T genome assemblies, for example, five rounds involving ONT, HiFi and MGI reads for Chinese Han T2T genome⁴, and 10 rounds involving*

ONT (3 rounds), HiFi (3 rounds) and MGI (4 rounds) reads using Nextpolish for maize T2T genome¹.

To evaluate the effects of multiple rounds of polishing in our pipeline, we calculated the QV and single-base error following your comments. We found that the QVs were improved from 32.573 for GV5 to 54.1759 for the final assembly T2T-goat1.0, and significantly decreased homozygous SNPs (genotype 1-1) were obtained from 234,984 for GV5 to 4974 for T2T-goat1.0, with single-base error rates dropping from 0.007888% to 0.000174%. We added the description to address this improvement in the revision, as shown below. Please also see lines 146-151.

“Multiple strategies were further used to assess the integrity and accuracy of T2T-goat1.0. The genome-wide average consensus quality value (QV) was estimated to be 54.18, indicating a base accuracy >99.999% and an increase from 32.57 in the assembly GV5 (Supplementary Fig. 5). The homozygous single-base mutations due to the assembly errors were estimated based on alignment of short reads, and their rates decreased from 0.007888% in GV5 to 0.000174% in T2T-goat1.0.”

– To show the effect of multiple polishing rounds on genome improvement, we used Merqury tool to calculate the QVs and error rates for the key steps of our polishing. We added the results in the updated Supplementary Fig. 5, as shown below.

T2T genome polishing pipeline

1. Chen, J. et al. A complete telomere-to-telomere assembly of the maize genome. *Nat. Genet.* 55, 1221-1231 (2023).
2. Hu, J., Fan, J., Sun, Z. & Liu, S. NextPolish: a fast and efficient genome polishing tool for long-read assembly. *Bioinformatics* 36, 2253-2255 (2020).
3. Hu, J. et al. NextPolish2: A Repeat-aware Polishing Tool for Genomes Assembled Using HiFi Long Reads. *Genomics Proteomics Bioinformatics*. 22(1): qzad009 (2024).
4. Yang, C. et al. The complete and fully-phased diploid genome of a male Han Chinese. *Cell Res.* 33, 745–761 (2023).

[2.2] In Line 160, a significant proportion of the PURs region consists of SD regions. Please provide a detailed description of the criteria used for determination.

– *We added more descriptions related to SDs and overlapping SDs in PURs. Please see below and lines 187-192, 866-867 and 877-881.*

– *The SD in PURs in Results (lines 187-192):*

“Previously unresolved regions (PURs) were identified by comparing all the chromosomes of *T2T-goat1.0* and *ARSI*¹¹ and inferred to the misassembled or newly assembled regions, for a total size of 288.5 Mb. More than 30 Mb of PURs were found on the X chromosome. These PURs consisted mostly of centromeric satellites and segmental duplications (SDs) which were discovered according to the previous method²³ and accounted for 81.92% (236.33 Mb) (Fig. 2b).”

– *PUR determination in Methods (lines 868-869):*

“The PURs were determined based on the alignments between *T2T-goat1.0* and *ARSI* (Supplementary Methods), according to the previous method⁵.”

– *The description of identifying SDs in Methods is shown as follows. Please also see lines 879-883.*

“Segmental duplications (SDs) were detected using BISER⁷ (v1.4), and low-quality SDs were filtered when they did not meet the following requirements⁶: 1) Sequence identity >90%, 2) $\leq 50\%$ gapped sequence in the alignment, 3) Aligned sequence of ≥ 1 kb, and 4) $\leq 70\%$ satellite sequence annotated by RepeatMasker. Determination of SDs in PURs was further described in Supplementary Methods.”

– *The detailed method to overlap SDs and PURs has been added in Supplementary Methods, as shown below.*

Segmental duplications (SDs) and other repeats in PURs

SDs and other repeats in PURs were discovered by overlapping the bed files and PURs using BEDTools (v2.30.0).

merging all files of SDs

```
bedtools merge -i all.sd.bed > sd.bed
```

```

# overlapping SDs and PURs
bedtools intersect -a pur.bed -b sd.bed

#01 satelite_cen
#sort the centromeric satellite regions (final.Sat_centr.region)
sort -k1,1 -k2,2n unsorted.Sat_centr.region >Sat_centr.region
#sort PURs (pur.region)
sort -k1,1 -k2,2n unsorted.pur.region > pur.bed
# overlapping SDs and PURs for the file (sac_pur.bed)
bedtools intersect -a pur.bed -b Sat_centr.region >sac_pur.bed
# making bed file (nosac_pur.bed) for PURs without SDs
bedtools subtract -a pur.bed -b Sat_centr.region > nosac_pur.bed

#02 SDs in PURs
# merging all files of SDs
bedtools merge -i all.sd.bed > sd.bed
# overlapping SDs and PURs
bedtools intersect -a pur.bed -b sd.bed > sd_pur.bed
# bed regions with the overlapped SDs and centromeric satellite
regions
bedtools intersect -a sac_pur.bed -b sd_pur.bed >sd_sac_pur.bed

# only centromeric satellite regions without SDs
bedtools subtract -a sac_pur.bed -b sd_sac_pur.bed > sac_nosd.bed
# only SDs without centromeric satellite regions
bedtools subtract -a sd_pur.bed -b sd_sac_pur.bed > sd_nosac.bed

#03 rDNA in PURs
# retrieving PURs without centromeric satellite regions and SDs
bedtools subtract -a nosac_pur.bed -b sd_pur.bed >nosac_sd_pur.bed
# retrieving rDNAs in PURs without centromeric satellite regions and
SDs
bedtools intersect -a nosac_sd_pur.bed -b rDNA.bed >
rDNA_nosa_sd.bed

#04 Repeats by RepeatMasker
# merging all repeats by RepeatMasker
bedtools merge -i sort.repeat.bed >repeat.bed
# retrieving the PURs without centromeric satellites, SDs, and rDNA.
bedtools subtract -a nosac_sd_pur.bed -b

```

```
rDNA.bed >nosac_sd_rdna.pur.bed
# only repeats by RepeatMasker in PURs
bedtools intersect -a nosac_sd_rdna.pur.bed -b
repeat.bed >repeat_pur.bed
```

#05 others

```
bedtools subtract -a pur.bed -b Sat_centregion >tmp1
bedtools subtract -a tmp1 -b sd_pur.bed >tmp2
bedtools subtract -a tmp2 -b rDNA.bed >tmp3
bedtools subtract -a tmp3 -b repeat.bed >final_others_pur.bed
```

[2.3] Additionally, SD regions may also potentially be the assembly errors due to heterozygosity. Have measures been taken to distinguish and exclude them? In Line 257, the SD (more accurately described as **random duplication**) shown in supplementary figure 11 is also worthy of validation to confirm its continuity as a genuine sequence rather than heterozygous sequence.

– *For assembly errors, we used multiple methods to evaluate the genome quality and potential assembly issues, including read coverage, QV, collinearity, etc. As mentioned in the above response, our SDs were determined using the strict criteria. The SD regions showed the overall good assembly quality as shown in the main text. Totally, we identified 286.70 Mb nonredundant SDs in T2T-goat1.0, and 73.28% (210.12 Mb) of the SD regions are located in the centromeric region. We further estimated the coverage of the SD regions based on ONT long reads, and checked whether the reads' coverage be reduced due to the heterozygosity from parents' variations. Due to the high degree of repetitive sequences and the high similarity of centromeric regions, the coverage in the centromeric regions is not even as shown in the other regions. We confirmed the presence and reliability of SD based on being covered by at least one ONT read. Using this validation method, we found that out of 62,776 SDs, 59,503 (94.79%) were completely spanned by ONT reads. Next, the remaining 3,273 SDs without ONT read covering were further examined for the reliability based on the coverage. We found that only 155 SDs accounting for 0.24% of the total number of SDs showed the coverages with less than half of the average coverage across the whole genome, among which 336 are located in the centromeric regions. We placed these 155 ambiguous SDs without >1 ONT read covering and less than half of the average coverage into Supplementary Table 7 in the revision. Actually, the similar enrichment of SDs is also observed in the human⁶ acrocentric chromosomes 13, 14, 15, 21 and 22.*

– *We added the description in the text of the revision to address this issue. Please see below and lines 280-289.*

“We identified a total of 286.70 Mb nonredundant SDs in T2T-goat1.0, with 73.28% (210.12 Mb) located in the centromeric region, while only 55.25 Mb was found on the chromosomes of ARS1. T2T-goat1.0 showed a remarkable increase in the number and length (additional sequences of 212.18 Mb) of interchromosomal SD pairs compared

to those of *ARS1*. Out of a total of 62,776 SDs, we confirmed the presence and reliability of 59,503 (94.79%) SDs that are covered by at least one ONT read. Next, we examined the remaining 3,273 SDs without ONT read covering for the reliability based on the coverage. We found that only 155 SDs accounting for 0.24% of the total number of SDs showed the ambiguity due to less than half of the average coverage across the whole genome, with 151 ones located in the centromeric regions (Supplementary Table 7).”

– In addition, we examined the read coverage in IGV for the SD region with scavenger receptor activity (OG0000004) family in Supplementary Fig. 14, and found that the ~3 Mb region covering 15 gene copies were aligned well by both ONT and HiFi reads. We performed the co-linearity analysis on this ~3 Mb region between *T2T-goat1.0P* and *T2T-goat1.0M*, and the assembly of this region was further confirmed, without the concerns of errors and heterozygosity.

”For example, in the gene family associated with scavenger receptor activity (OG0000004), 15 gene copies were annotated on CHI5 in *T2T-goat1.0* instead of only 8 in *ARS1*, and the assembly of the corresponding ~3 Mb region was further confirmed for its continuity with evidence of long read alignment and collinearity between *T2T-goat1.0P* and *T2T-goat1.0M* (Supplementary Fig. 14).”

Supplementary Fig. 14 Assembly of the OG0000004 gene family genes in *T2T-goat1.0* compared to that in *ARS1*. a, Collinearity of the region with tandem genes

of the scavenger receptor activity (OG0000004) family is shown between *T2T-goat1.0* and *ARS1*. The 15 genes in this gene family were assembled in a tandem way on Chr05 of *T2T-goat1.0*, but only 8 genes on Chr05 and unplaced contigs of *ARS1*. **b**, The assembly of the region containing OG0000004 is examined for alignments of HiFi and ONT reads against *T2T-goat1.0*. **c**, The collinearity between *T2T-goat1.0P* and *T2T-goat1.0M* confirmed the accurate assembly of this region containing OG0000004 on Chr05, rather than the assembly errors due to the heterozygosity between the haplotype genome assemblies.

6. Vollger, M.R. et al. Segmental duplications and their variation in a complete human genome. *Science* 376, eabj6965 (2022).

[2.4] In supplementary figure 14, the chrX of T2Tgoat1.0 appears shorter than that of Saanen_v1, and there is a substantial difference in chrX length among the three assembly versions. Is this a genuine difference, or is it merely a representation error?

– *Sorry for the representation error on X chromosomal length in Supplementary Fig. 14. The length of X chromosome (144.69 Mb) in T2T-goat1.0 is longer than those in the other two genomes (142.35 Mb for Saanen_v1 and 115.94 Mb for ARS1). We replotted this figure, and updated the X chromosomal lengths of these three genome assemblies in Supplementary Fig. 16 in the revision.*

Supplementary Fig. 17c The syntenic regions are observed between chromosomes X (ChrX) and Y (ChrY). The chromosomes (or scaffolds) X and Y in *ARS1* and *Saanen_v1* both show the high collinearity with that in *T2T-goat1.0*. Lines for collinearity are shown in random colors between ChrY scaffold of *ARS1* and ChrX and ChrY of *T2T-goat1.0*.

[2.5] In Line 353, the discussion regarding the testis-specific protein Y-encoded (TSPY) and heat shock transcription factor Y-linked (HSFY) genes is intriguing. Considering the significant differences in their numbers compared to the *ARS1* assembly and the human genome, further validation is necessary to confirm their authenticity.

– *We annotated 134 protein-coding genes across the whole Y chromosome, including 27 gene copies of testis-specific protein Y-encoded (TSPY) and 14 copies of heat shock*

transcription factor Y-linked (HSFY). The multiple copies of TSPY and HSFY are commonly observed in the mammals' Y chromosomes. We added this in the Discussion section. Please see below and lines 614-621.

“An increase in the number of copies of the testis-specific *TSPY* gene was also reported on the Y chromosomes of human (45 copies)⁸ and cattle (94 copies)⁹. This dosage effect addresses spermatogenesis and is related to the regulation of spermatogenic efficiency⁸. The copy number of *TSPY* reportedly varies significantly among various populations⁹. *HSFY* is believed to be expressed predominantly in the testis and to regulate spermatogenesis by mediating the expression of heat shock proteins (HSPs)¹⁰. The number of *HSFY* copies varies among the species, with only 2 copies in human⁸, 8 copies in feline¹¹, 6 copies in sheep¹², and 197 copies in cattle¹⁰.”

– *In addition, we performed digital droplet PCR (ddPCR) to confirm the gene copies of these two genes (TSPY and HSFY), and added a new Supplementary Table 10. We added the description in the revision, and please see below and the lines 389-393.*

“The annotation revealed two or more copies of 74 genes in the ampliconic regions on *T2T-CHIY1.0* (Supplementary Table 9), including 27 gene copies of testis-specific protein Y-encoded (*TSPY*) and 14 copies of heat shock transcription factor Y-linked (*HSFY*) (Fig. 4), and the gene copies of *TSPY* and *HSFY* were validated by using ddPCR (Supplementary Table 10).”

Reviewer #3 (Remarks to the Author):

The authors report construction of a gapless T2T assembly for a goat and applied this as a reference for detection of structural variants and SNP, by mapping HiFi data generated for one animal from each of five other sheep breeds. Selective sweeps were detected using either the T2T-goat1.0 assembly or the existing RefSeq assembly ARS1. A population genetic analysis was performed with public short read data from 495 goats among 11 populations to identify regions associated with cashmere trait.

[3.1] In general the authors provide convincing evidence that the assembly represents the complete genome of a goat, one of the first mammals and particularly livestock species to reach this goal. This is notable as the goat was also the first mammalian assembly (the ARS1 reference) to reach “gold” status in 2016 based on a combination of long read, short read, and chromatin conformation data.

– *We appreciate all your valuable comments. Following your comments, we have made tremendous modifications to correct the potential errors or inappropriate points*

in the revision.

[3.2] The authors note extensive regions of additional genome (primarily heterochromatin) along with additional genes not included in the ARS1 reference. The author's approach was unusual for T2T assembly in that they used "merged" assembly as a first step rather than constructing the parent-specific kmer database and performing haplotype-resolved contig assembly. As such, the final haploid assembly does not represent either parental haplotype in its entirety.

– *We conducted the additional assemblies by using Hifiasm trio mode to generate the haplotypic assemblies T2T-goat1.0P (paternal origin) and T2T-goat1.0M (maternal origin) based on the sequencing of the parents and ONT/HiFi reads for T2T-goat1.0 individual. We repeated the pipelines of T2T-goat1.0, to do scaffolding, gap-filling, polishing for these two haplotypic assemblies. Finally, we generated T2T-goat1.0P (paternal origin) and T2T-goat1.0M (maternal origin), whose average QV reach to 53.06 and 52.37, respectively. We added the description and summary tables and figures in Methods and Results of the revision (See lines lines 128-143, 166-177, and 846-854). We also released these haplotypic assemblies as shown in the response [1.1] above and "Data Availability" in the revision.*

– *The method for haplotype genome assemblies (lines 846-854):*

"Haplotype genome assembly

The initial assembly for the autosomes of the haplotype-resolved genomes (*T2T-goat1.0P* and *T2T-goat1.0M*) was performed by using Hifiasm (v0.16.1 r375) based on the trio mode. The scaffolding, gap filling and polishing for *T2T-goat1.0P* and *T2T-goat1.0M* were performed with the similar method for *T2T-goat1.0*. The ONT and HiFi reads were binned for paternal and maternal origin based on the k -mer ($k = 21$) generated from parental short reads, which is the same to the trio-binning strategy as shown for Y chromosome assembly. The binned long reads were used to fill gaps and polish the haplotype genomes by using Nextpolish2 (v0.1.0), rather than the NGS data, to avoid the introduction of the other haplotype sequences."

– *The result description for haplotype genome assembly (lines 128-143):*

"Based on long reads from the assembled buck and short read sequence data from its parents, the haplotype assemblies were generated by using Hifiasm¹ assembler based on the trio mode, for *T2T-goat1.0P* of paternal origin with complete Y chromosome and *T2T-goat1.0M* of maternal origin with complete X chromosome respectively (Table 1). Parent-specific k -mers based on short reads were used to pick ONT and HiFi reads of paternal and maternal origin, to fill 31 gaps for *T2T-goat1.0P* and 25 gaps for *T2T-goat1.0M*. Y chromosome was also independently assembled with the trio-binned paternal ONT reads, and improved for scaffolding, gap filling and correction, with the assistance of Y chromosomal contigs *de novo* assembled from HiFi reads. As a result, we generated a complete Y chromosome with two telomeres spanning 20.96 Mb. The final T2T genome assembly, *T2T-goat1.0*, covering all the 29 autosomes and X and Y chromosomes, was obtained with a total length of 2.86 Gb (Table 1). This size was

slightly longer than the estimated genome size of 2.76 Gb obtained using the NGS short reads. After final polishing, the chromosomes X and Y from *T2T-goat1.0* were combined with the haplotype-resolved autosomes to obtain the paternal and maternal genome assemblies *T2T-goat1.0P* (2.71 Gb with N50 103.46 Mb) and *T2T-goat1.0M* (2.81 Gb with N50 109.45 Mb) respectively.”

– *Lines 166-177:*

“Meanwhile, *T2T-goat1.0P* and *T2T-goat1.0M* achieved the T2T level assembly, with the whole genome QV 53.06 and 52.37 respectively. The challenges for assembly of acrocentric centromeres and lack of binned long reads still left 11 gaps in the centromeric regions of 9 chromosomes in *T2T-goat1.0P* and 6 gaps in the centromeric regions of 6 chromosomes in *T2T-goat1.0M* (Supplemental Table 4). Except for the centromeric regions, the genomes have relatively uniform HiFi and ONT coverage. We estimated the switching errors and obtained 0.036%, 0.016%, and 0.011% for the three T2T assemblies *T2T-goat1.0*, *T2T-goat1.0P*, and *T2T-goat1.0M* respectively. Furthermore, by comparing autosomes between the two haplotypes, we detected 5,287,863 single nucleotide variants (SNVs), 726,939 small insertions and deletions (< 50 bp), and 22,689 SVs (\geq 50 bp). Considering the completeness and quality, *T2T-goat1.0* represented a better assembly (Supplementary Fig. 4a) for the downstream analysis compared to the two haplotypes.”

[3.3] The authors do not provide hamming error rate or switching error rate so the extent to which this might lead to erroneous expansion or contraction is difficult to assess, and the depiction of the assembly graph in Supplementary Figure 3 is not colored by parental kmers.

– *We used NGS reads to calculate switch error rate using Merqury software. We provided the switch error rates of 0.036%, 0.016%, and 0.011% for the three T2T assemblies T2T-goat1.0, T2T-goat1.0P, and T2T-goat1.0M respectively. We added this result together with the description of haplotype assemblies in the revision as shown in the response [3.2] above.*

– *We used the parental kmers to color the assembly graph, and added a new figure as Supplementary Figure 4a.*

Supplementary Fig. 4a Assembly graph strings for the chromosomes. The chromosomes are colored in red for maternal k -mers, in blue for paternal k -mers, and in gray for no parental specific k -mers. The centromeric regions of the four chromosomes, Chr12, Chr13, Chr19 and Chr22, are entangled, and further enlarged and shown with centromeric regions highlighted in Fig. 1b. Another tangle related to 45S rDNA repetitive sequences (yellow lines in bottom left box) involve five chromosomes, Chr02, Chr03, Chr04, Chr05, and Chr28. The pseudoautosomal region (PAR) is shared between ChrX (two fragments in red) and ChrY (in blue), which are in the bottom right box.

[3.4] The authors did perform some appropriate quality measures, although the demonstration of this in the Figures is difficult to assess due to their resolution (a common difficulty in presenting complete genomes). For example, mapping the ONT reads to the assembly is a good step, but the resolution of Figure 1b renders the demonstration rather meaningless. Nothing less than a drop or increase in coverage spanning many kilobases to megabases would be visible at this level of resolution to detect any issues.

– *To address the issue of resolution in Fig. 1b in original submission, we highlighted*

the low-coverage (a drop due to <0.5 folds of the average coverage) and high-coverage (an increase due to >2 folds of the average coverage) regions, to color the potential assembly issues. According to the reviewer's suggestion [3.8] below, we increased the resolution (100 kb non-overlapping windows), modified this figure, and moved it into Supplementary Fig. 6 in the revision, as shown below.

Supplementary Fig. 6 Coverage and issues of 31 chromosomes in T2T-goat1.0.

Coverages of NGS (purple), ONT (red), and HiFi (blue) reads are shown for T2T-goat1.0. The low-coverage (a drop due to <0.5 folds of the average coverage) and high-coverage (an increase due to >2 folds of the average coverage) regions are highlighted for the potential assembly issues. The centromeres are highlighted in red for the chromosomal bars (gray) in the bottom.

[3.5] In addition, Figure 1b alternates the acrocentric chromosomes to have the beginning at the start or end of the chromosome—this may reflect error in the ARS1 assembly but seems like a good place to fix this. Generally, acrocentric chromosomes are presented with the short arm/telocentric arm as the proximal (leftmost or top) of the representation or sequence.

– Good comment! To be consistent with the orientation of chromosomes in the current goat genome reference ARS1, we kept the identical chromosome orientations in T2T-goat1.0, which would be convenient to make a comparison between T2T-goat1.0 and ARS1. Accordingly, the centromeres were randomly located either in the left or right ends of all acrocentric autosomes. Meanwhile, we followed your suggestion and provided the updated version for goat T2T genome assembly, namely T2T-goat2.0. T2T-goat2.0 placed all the centromeres on the leftmost ends of all the chromosomes, based on the chromosomal sequences of T2T-goat1.0. T2T-goat1.0 and T2T-goat2.0 (together with four haplotype assemblies) are available in both CNGDC (BioProject accession PRJCA022847) and NCBI databases (accession PRJNA106251), as shown in the Data Availability and the response [1.1] above.

[3.6] The method for assembly is well documented and it is likely that if the primary data is available it could be replicated. The manuscript indicates public release of the data through the National Genomics Data Center in China, but I was unable to access it. Unfortunately, this is not an uncommon occurrence when attempting access from the U.S. The manuscript has very extensive characterization of the Y chromosome in particular, along with centromere analysis of autosomes and sex chromosomes, plus the association analyses. The total data is impressive but hard to get through when so much

is packed into a single manuscript, making it very long and complicated. For contrast, the human T2T description had these features in separate companion manuscripts that could each delve into specific aspects.

– We uploaded our T2T genome assemblies (T2T-goat1.0, T2T-goat1.0P, T2T-goat1.0M, T2T-goat2.0, T2T-goat2.0P, and T2T-goat2.0M) and raw data into both CNGDC (BioProject accession PRJCA022847) and NCBI (accession PRJNA106251) databases. We have released these data before the submission of the revision. We updated the Data Availability with all accessions as follows. Please also see lines 1051-1057.

“The genome assemblies (T2T-goat1.0, T2T-goat1.0P, and T2T-goat1.0M; and T2T-goat2.0, T2T-goat2.0P, and T2T-goat2.0M) and raw sequencing data generated in this study, including PacBio HiFi data, ultralong ONT data, MGI data, Iso-seq data and ChIP-seq data, can be achieved from National Genomics Data Center (<https://ngdc.cnbc.ac.cn/>) with BioProject number PRJCA022847, and NCBI with accession number PRJNA1062519. T2T-goat2.0 is the update of T2T-goat1.0, with all the centromeres placed on the leftmost ends of acrocentric autosomes.”

– Good comment! We reorganized the manuscript by focusing on the improvement of T2T-goat1.0 and new findings, including novel genes in PURs, Y chromosomal structure with previously unknown genes, centromeric features in goat, and populational identified selection signals with genes and variants in PURs. The genome assembly resources and results refreshed the goat ARS1 reference¹³ that was considered as the first mammalian assembly to reach “gold” status in 2016, according to the reviewer’s comment. T2T-goat1.0 provides insight into the novel features unknown in the previous goat genome assemblies. Up to date, T2T-goat1.0 is still one of the first mammalian T2T genome assemblies, under the hot topic of T2T assemblies, and we have omitted some parts in the human T2T companion papers, such as, epigenetic regulation and transcriptional patterns.

13. Bickhart, D.M. et al. Single-molecule sequencing and chromatin conformation capture enable de novo reference assembly of the domestic goat genome. *Nat. Genet.* 49, 643-650 (2017).

[3.7] The selective sweep analysis is a reasonable first approximation based on selective sweep analysis of cashmere and non-cashmere animals, although it could have been confounded by other breed effects since the study was not conducted in a population segregating the trait. Rather, the study uses animals of cashmere breed(s) compared to non-cashmere goat populations.

– We performed selective sweep analysis between cashmere and non-cashmere goat populations. Selective sweep is a method to search for the fixed or selected alleles that might be genetic hitchhiking, in a population against the other one. Combining multiple breeds were always used in selective sweep analysis. For example, domestication (worldwide domestic goat populations) and wild (24 bezoars) goats were compared for selective sweeps previously¹⁴. For local adaption and cashmere selection, selective

*sweep analysis was performed between northern (19 breeds) and southern (39 breeds) Chinese goats, as 11 of 19 northern goats are cashmere-producing, and southern breeds are not cashmere-producing¹⁵. Breed effects are unavoidable, and the selected signals are not necessary to be directly associated with the cashmere trait. However, the top two F_{ST} outliers in this previous study contained two genes, *FGF5* and *EDA2R*¹⁵, which are both identified in this study and directly related to the development of hair follicles and cashmere trait.*

*In this study, we selected eight typical cashmere populations (four from Tibetan plateau, three from Northern China, and one from India) as the cashmere group, in a comparison with non-cashmere goats, in selective sweep analysis related with cashmere trait selection, rather than the 19 breeds of northern Chinese goats in a previous study¹⁵. This strategy is expected to screen out the selection signals that are common in these eight cashmere goat breeds that produce cashmere wool, and reduce the selected signals related to breeds or breed effects, although they experienced their own breeding history and are expected to hold selection sites. Our results showed a successful identification of 2692 selective signals spanning 109.22 Mb and covering 1440 genes that are shared in all cashmere individuals. We found some typical genes that are known to be associated with cashmere trait, for example, *KRT* genes on both *CHI5* and *CHI19*, and the *FGF* genes on both *CHI5* and *CHI20*. Also, *TYRP1* on *CHI8*, *KITLG* on *CHI5* and *SLC24A4* on *CHI21* are associated with coat color. Of note, 274 selective signals located in the PURs, covering 72 genes, are valuable for a further investigation on their role in cashmere-trait-related functions. Actually, cashmere trait has not been measured in a non-cashmere breed, and cashmere trait is typically observed and specialized in cashmere goats. We added the description to address this analysis and its background, as followed.*

“A total of 46 individuals in eight typical northern cashmere populations (four from Tibetan plateau, three from Northern China, and one from India) were selected as the cashmere group in a comparison with 27 non-cashmere goat breeds (Supplementary Table 5), in selective sweep analysis, as they held divergent genetic basis. We focused on the cashmere goats for cashmere trait selection, rather than the 19 breeds of northern Chinese goats in a previous study¹⁵.”

14. Zheng, Z. et al. The origin of domestication genes in goats. *Sci. Adv.* 6, eaaz5216 (2020).

15. Cai, Y. et al. Ancient genomes reveal the evolutionary history and origin of cashmere-producing goats in China. *Mol. Biol. Evol.* 37, 2099-2109 (2020).

[3.8] Figures : The detailed and extensive research includes 7 primary figures summing to 43 panels and 23 supplementary figures summing to 42 panels, for a total of 85 panels. This density of information made it quite slow to adequately review, in addition to causing difficulty in interpretation of some figures due to the resolution. As mentioned above, Figure 1b provides too low resolution to be of value and could be moved to

supplementary or better described to note a region of a given minimum length showing coverage below a threshold (e.g. “the coverage across all chromosomes did not fall below X% of average for any region exceeding Y bases”). Figure 1c has very low information content as the assembly size by year of release does not seem to have a lot of bearing on the conclusions presented; this panel could be deleted. Figure 2a and 2b show redundant information and 2a has additional useful illustration so suggest 2b could be deleted. The example of support for correcting ARS1 in panel 2d could be moved to a supplemental figure as it simply illustrates one of the many differences. The resolution in figure 2d is too low to be able to interpret much from it and could be removed, in addition panel 2f demonstrates collinearity with another improved X chromosome making the alignment in 2e less important. The tiny blob plots of figure 2g could easily be put into larger supplementary figure and line 194 is sufficiently descriptive of the result without the visual. Panel 2h could be fully described by quoting statistics (average and range) for the five breeds in the long read mapping effort, the panel is not illuminating and also will end up very small print at the scale shown in the full-page figure.

– We have reorganized and modified Fig. 1 and Fig. 2 as suggested. We deleted the Fig. 1c and Fig. 2b in the original submission. Fig. 1b, Fig. 2d, 2e, and 2g/2h were moved to Supplementary Fig. 6, Supplementary Fig. 9, Supplementary Fig. 7, and Supplementary Fig. 10 respectively, in the revision. Fig. 2 is shown as followed in the revision.

Fig. 1 Goat T2T genome assembly with 29 autosomes and chromosomes X and Y.

a, Genome assembly strategy for *T2T-goat1.0* and its haplotype genomes *T2T-goat1.0P* and *T2T-goat1.0M* of a buck. Trio-binning assemblies for the autosomes of *T2T-goat1.0P* and *T2T-goat1.0M* and Y chromosome was performed based on long reads of the buck and MGI short reads from its parents. **b**, The assembly graph string shows a tangle among the four chromosomes, Chr12 (blue line), Chr13 (green line),

Chr19 (red line) and Chr22 (brown line), due to the high similarities of centromeric sequences in gray. The centromeric regions in the assembly tangle are enlarged and shown in the right panel. **c, d**, Genome features of Chr12 and Chr19 in *T2T-goat1.0*. The assembly graph tangle involving Chr12, Chr13, Chr19 and Chr22 was resolved in *T2T-goat1.0*, and Chr12 and Chr19 are selected to exhibit the completeness and features across the whole chromosome. The following information is provided from top to bottom: the gene density (red), the density of LINEs and SINEs (orange), the satellite density (green), the TE density (blue), error k -mer ($k=21$, purple), and the minimum unique k -mer (MUK) per 100 kb. The more MUK values indicate more repetitive sequences in a 100-kb window, and more yellow and green colors indicate the presence of the centromeric regions. All the features are shown in 10-kb windows, except for MUK.

Fig. 2 Synteny and improvement of the *T2T-goat1.0*. **a**, Syntenic and nonsyntenic regions between *ARS1* (top) without telomeres and *T2T-goat1.0* (bottom) with telomeres that are indicated by dark purple triangles. The collinearity between the two genome assemblies is shown as gray lines or blocks, and the inversions are shown in orange. The yellow bars represent the previously unresolved regions (PURs) in *T2T-goat1.0*. The gene density in 100-kb windows is shown as dark green bars. **b**, The

proportions of various repetitive elements in PURs. CenSat, satellite sequences in the centromeric region; SDs, segmental duplications; RepMask, repeats by RepeatMasker. c, Syntenic and nonsyntenic regions of the X chromosome (ChrX) between *Saanen_v1.0* (NCBI accession no. GCA_015443085.1) and *T2T-goat1.0*. Synteny and inversion are shown in gray and orange respectively.

[3.9] The legend of Figure 3 nor the text indicates what chromosome is being shown in panel 3a, but the ChIP-seq is a bit hard to interpret with possibly the region between 4 and 9 Mb being enriched (but not more than the signal of 0-1.4 Mb?). The “methylation dip” associated with the centromere would appear to be at about 3 and 8 Mb suggesting the centromere should lie in between, but the ONT data does not show much agreement with this. The text is unclear on the author’s interpretations of these results, as line 308-9 states that a dip is observed coincident with ChIP-seq signals, but the figure doesn’t seem to show that. Perhaps annotation of the figure as to where exactly the authors think the centromere being illustrated is supposed to be?

– *Fig. 3a infers to chromosome 1 (CH1), which was added into Fig. 3a in the revision. The ChIP-seq signal enrichment difference could be observed between centromeric and pericentromeric regions, which is split at ~9Mb of CH1, in Fig. 3a. However, the coverage depth of ChIP-seq were found lower on some locations of the centromeric regions, such as, ~1.5Mb to 4Mb on the whole CH1 centromeric region (0~9Mb). Also, the enrichment of methylation was observed in the whole centromeric regions of CH1. We added an arrow to indicate the boundary of centromeric and pericentromeric regions in Fig. 3a. Our results did not show the “methylation dip” and an apparent correlation with ChIP-seq signals, as mentioned by the reviewer in this comment. We deleted the section for the description on dip and correlation of ChIP-seq and methylation in the revision. Instead, we addressed the support of the centromeric regions from ChIP-seq and methylation results.*

[3.10] Figure 4b and 4c could be moved to supplemental since they are only used to illustrate improvement of the assembly over two earlier assemblies and not to make any important points about the goat Y chromosome. Similarly, Figure 5a, 5b, and 5c show the improvement when using a complete genome over the previous assemblies which can be easily capture by a few statistics in one table and move the charts to supplemental. Figure 5d is very difficult to interpret since the resolution is too low for meaningful analysis. It is not clear that Figure 5e is necessary for the point being made in the text, that the keratin filament pathway was the GO term with most significance and number of genes enriched, so this could also be supplemental figure or removed.

– *We reorganized Fig. 4 and Fig. 5 in the revision. Figure 4b and 4c have been moved to Supplementary Fig. 17. We listed the data in Fig. 5a in Supplementary Table 11. Fig. 5b and 5c are moved into Supplementary Fig. 19 while Fig. 5d is moved into Supplementary Fig. 20. Fig. 5e is removed in the revision. In the revision, there are a total of 6 figures, rather than 7 ones in the original submission, as Fig. 5 in the original submission was not kept.*

[3.11] Line 121 I suggest that “short read sequence data” is better than “next-generation sequencing data”

– *Revised as suggested.*

[3.12] Line 163 It is not clear what utility there may be in using PUR and NAR as acronyms for what appear to be identical features? Or at least it is not clear what the difference is from the text. Are NARs a subset of PURs? If so what basis is there for the subset?

– *PURs are the completely assembled region on the chromosome level in T2T-goat1.0, which are not assembled in the chromosomes of ARS1, while NARs are the newly assembled sequences in T2T-goat1.0, which are not included in chromosomes and unplaced contigs of ARS1. PUR is obtained by comparing the chromosomes of T2T-goat1.0 and ARS1. NAR is obtained by comparing all chromosomes of T2T-goat1.0 and all contigs of ARS1. NARs are a subset of PURs. NARs are the totally new sequences that are not shown in ARS1, and some of PURs are not found in chromosomes of ARS1, but present in the unplaced contigs of ARS1. The PURs are valuable to discover more variations or genes in the selective genomic regions that were previously not located on the chromosomes of ARS1, while the NARs are key to identifying new genes. We also updated the descriptions on PUR and NAR in the revision, as followed below. Please also see lines 187-189 and 192-194.*

“Previously unresolved regions (PURs) were identified by comparing all the chromosomes of *T2T-goat1.0* and *ARS1*⁵ and inferred to the misassembled or newly assembled regions, for a total size of 288.5 Mb.”

“Additionally, we detected an overall size of 157.38 Mb for the newly assembled regions (NARs) as a subset of PURs that are not included in all the chromosomes and unplaced contigs of *ARS1*, and found that most NARs were in centromeric regions.”

[3.13] Line 167 What does “maximum QV” refer to? Is that the maximum average QV value for a chromosome?

– *The Quality Value (QV) is calculated based on k-mers, comparing the consistency between second-generation reads and the genome to evaluate genome quality. It is calculated across the whole genome or the whole chromosome. The maximum quality value refers to the highest average QV value among all chromosomes. In T2T-goat1.0, the quality value of chromosome 10 is 59.31, which is the highest average quality value among all chromosomes. We modified the sentence in the revision. Please see below and lines 194-198.*

“In contrast to the genome-wide average QV of 40.93 and 32.83 for chromosomes and all the sequences (including chromosomes and unplaced contigs) in *ARS1* respectively, the QV estimate was greatly enhanced for *T2T-goat1.0*, for which the genome-wide average QV was 54.19 and the maximum average QV of each single chromosome was 59.31 for CHI10.”

[3.14] Line 168 The description of MUK is inadequate—the authors define it as “a k-mer present only once in the genome”, but how is that a minimum unique k-mer? What meaning does a sliding 100kb window of kmer have exactly? If it is “unique” k-mer, then there shouldn’t be large regions of genome as depicted (at very low resolution) in the centromeres?

– *Following your suggestion, we have made modifications for better description of MUK. We rewrote the description of MUK to show meaningful results of unique sequence increase in T2T-goat1.0. We followed the previous method¹⁶ and their scripts (https://github.com/msauria/T2T_MUK_Analysis) to calculate MUK. Please see below.*

– *The description is updated as followed below (also see lines 198-206):*

“To assess the increase of unique sequences in *T2T-goat1.0*, we calculated the minimum unique *k*-mer (MUK) length, which was defined as a *k*-mer presenting the minimum distance in a window of 100 kb needed to identify a unique sequence in the genome¹⁸, and longer MUK length (up to 100 kb) was obtained if more repetitive sequences found in a 100-kb window. We observed a substantial increase in the number of unique sequences in *T2T-goat1.0* compared to that in *ARSI* (Supplementary Fig. 8). Longer MUKs in 100-kb windows across the chromosomes were observed in the centromeric regions due to enriched repetitive sequences (Fig. 1c and Fig. 1d).”

– *Meanwhile, we added the details of calculating MUK in Supplementary Methods. Please see below and lines 199-205 in Supplementary Methods.*

“**Minimum unique *k*-mer (MUK) calculation**

MUKs were calculated in 100-kb windows for both *T2T-goat1.0* and *ARSI*, according to the previously published method¹⁶. T2T Minimum Unique *K*-mer Analysis pipeline (https://github.com/msauria/T2T_MUK_Analysis) was used with a minor modification to identify the longest common prefix (LCP) based on the sequences concatenated from the two genome assemblies. LCP values plus one represented the MUK at a window site (100-kb windows), and the windows with gaps in *ARSI* were determined for no MUK.”

– *The legend for Supplementary Fig. 8 was updated as followed.*

“**Supplementary Fig. 8 Length of minimum unique *k*-mers in *T2T-goat1.0* compared to *ARSI*.** Minimum unique *k*-mers (MUKs) were calculated in 100-kb windows for the chromosomes of both *T2T-goat1.0* and *ARSI*, according to T2T Minimum Unique *K*-mer Analysis pipeline (https://github.com/msauria/T2T_MUK_Analysis). The more MUK values indicate more repetitive sequences in a 100-kb window.”

18. Aganezov, S. et al. A complete reference genome improves analysis of human genetic variation. *Science* 376, eab13533 (2022).

[3.15] Line 174 Delete “Additionally”

– Deleted. Please see line 208.

[3.16] Line 191 Delete “Furthermore”

– Deleted. Please see line 225.

[3.17] Line 192 More informative would be “Short read data from previous sequencing efforts focused on global populations of domestic and wild goats was retrieved from public databases (Supplementary Table 3).” This avoids any impression that the present study generated this data.

– Revised as suggested. Please see lines 225-227.

[3.18] Line 195 “improved the number of mapped reads significantly” may mean “substantially”, unless there was an actual significance test performed in which case a p-value should be provided.

– Replaced with “substantially.” Please see line 230.

[3.19] Line 213 There is no mention of why chromosomes 12 and 19 are illustrated, are they meant to be representative in some way or are they special cases that some feature should be pointed out?

– As shown in Supplementary Fig. 4 and Fig. 1b in the revision, chromosomes 12, 13, 19 and 22 are entangled with due to the shared centromeric complex structures, making the assembly of the two chromosomes somewhat challenging. Therefore, we displayed the genomic features of chromosomes 12 and 19 in Fig. 1c and Fig. 1d in the revision, especially exhibiting the resolved centromeres. We added the explanation in the legend of Fig. 1, and also added the description to mention this point in the revision. Please see below and lines 112-115.

“For example, a tangle of centromeric regions among four chromosomes, CHI12, CHI13, CHI19 and CHI22, is shown in the assembly graph string (Fig. 1b), and our assembly resolved the centromeric regions for these four chromosomes, as shown for the completeness and genome features on CHI12 and CHI19 (Fig. 1c and Fig. 1d).”

– The legend of Fig. 1c and 1d is updated as shown below.

“**Fig. 1c, d**, Genome features of Chr12 and Chr19 in *T2T-goat1.0*. The assembly graph tangle involving Chr12, Chr13, Chr19 and Chr22 was resolved in *T2T-goat1.0*, and Chr12 and Chr19 are selected to exhibit the completeness and features across the whole chromosome. The following information is provided from top to bottom: the gene density (red), the density of LINEs and SINEs (orange), the satellite density (green), the TE density (blue), error k -mer ($k=21$, purple), and the minimum unique k -mer (MUK) per 100 kb. The more MUK values indicate more repetitive sequences in a 100-kb window, and more yellow and green colors indicate the presence of the centromeric regions. All the features are shown in 10-kb windows, except for MUK.”

[3.20] Line 228 “Among these genes” is vague, does this refer to all protein-coding genes or BUSCO genes?

– *For “Among these genes”, it refers to all protein-coding genes. We have modified this sentence to clarify it in the revision. Please see below and lines 263-264.*

“Among these protein-coding genes, 1126 were newly anchored genes in PURs, and 446 were newly assembled genes (NAGs) in NARs that were not included in *ARSI*.”

[3.21] Line 234 Why would one think that newly assembled genes would be enriched in pathways? If you are thinking that a general principle is that genes lying in repetitive regions or segmental duplications share pathways, is there a reference to support this?

– *The enrichment of pathways for newly assembled genes has been removed in the revision.*

[3.22] Line 236 delete “Additionally”

– *Deleted.*

[3.23] Line 237 add reference to the supplementary figure with tangle in string graph

– *We have added references to the supplementary figure with tangle in string graph, shown as followed. Please see below and lines 267-277.*

“A tangle of the assembly string graph located far from the centromeric end was found to be associated with 45S rDNA repetitive sequences, comprising of 18S, 5.8S, and 28S on the distal regions of q arm of five chromosomes, CHI2, CHI3, CHI4, CHI5, and CHI28 (Supplementary Fig. 4a), as the reference point for distal and proximal regions is the centromere. The locations of 45S rDNA were further confirmed by fluorescence in situ hybridization (FISH) signals on five pairs of chromosomes (Supplementary Fig. 4b). The 5S rDNA was only found on the proximal regions of CHI28, opposite to the 45S rDNA end, as shown in both the assembly and FISH (Supplementary Fig. 4b). The results revealed a pattern of telomere (T) - rDNA (R) – centromere (C) – telomere (T) (TR-CT), with centromere and rDNA located on opposite chromosomal ends (Supplementary Fig. 4c).”

[3.24] Line 241 (and other lines) It is not established jargon to use “left” and “right” in relation to position on chromosomes, generally the usage is “proximal” or “distal” and the reference point is the telomere of the q arm (or telomere at the centromeric end for acrocentric) as “proximal”

– *We have updated all descriptions regarding chromosomal positions in the entire text by using "proximal" or "distal" on q arm of acrocentric autosomes and chromosome X, and p and q arms of metacentric chromosome Y.*

It is known that each chromosome has a short arm designated as “p,” a long arm

identified by the letter “q”, and a narrowed region at which the two arms are joined (centromere). “Distal” indicates away or farthest from a particular point of reference, meaning the chromosome’s centromere (<https://rarediseases.org/rare-diseases/chromosome-4-monosomy-distal-4q/#>). Based on the definitions of p, q, distal and proximal terms, we followed the suggestion from the reviewer, and selected the centromere, i.e., the telomere end neighbor to the centromere for acrocentric autosomes, as reference point for description of distal and proximal regions. For example, the description related to rDNA locations is shown as follows. Please see the response [3.23] above and lines 267-277.

[3.25] Line 244 I think you mean “located on opposite chromosomal ends”, not that they can be on either one end or the other.

– Replaced with “located on opposite chromosomal ends”. Please see line 276.

[3.26] Line 313 CNEP-A should be CENP-A

– Corrected.

[3.27] Line 370 “extended our search for repeat sequences on the sheep Y chromosome.” I think this needs some context, e.g. “extended our search for repeat sequences by reference to repeat sequences found on the sheep Y chromosome in an assembly of a sheep genome.” Also, a reference and accession for this T2T-sheep1.0 assembly should be provided.

– Following your comments, we have made the correction. The centromeric repeat unit sequences of sheep Y chromosome (CenY for T2T-sheep1.0) have been included in Supplementary Method in this revision. Before submission of the revision, we have released all the genome assemblies and raw data for both sheep (T2T-sheep1.0) and goat (T2T-goat1.0), for reviewing convenience of reviewers and editors. The sheep T2T-sheep1.0 manuscript is under review and consideration in another Nature journal, and we added its reference (preprint version) in the revision. Please see below and lines 407-414.

“As centromeric repeats (SatI, SatII and SatIII) and ChIP-seq signals based on Phospho-CENP-A (Ser7) antibody were not found on T2T-CHIY1.0, we extended our search for repeat sequences by reference to repeat sequences found on the sheep Y chromosome in an assembly of a sheep genome. A new repeat sequence of 1474 bp was inferred as the putative centromere-specific repeat unit, CenY, since it was highly homologous to another CenY (2516 bp) in the centromeric region on the Y chromosome of T2T-sheep1.0¹⁷ (T2T genome assembly for a ram of the Hu sheep breed, NCBI BioProject accession number PRJNA1033229).”

– Sheep T2T genome assemblies (T2T-sheep1.0 and its haplotypes T2T-sheep1.0P and T2T-sheep1.0M) are available in NCBI under the BioProject accession number PRJNA1033229.

17. Luo, L.-Y. et al. Telomere-to-telomere sheep genome assembly reveals new variants associated with wool fineness trait. bioRxiv (2024).

[3.28] Line 374 The CenY symbol and tight “peak” in Figure 4a does not appear to be at a border of a hypermethylated region? Why is there no track for CenP-A for Y chromosome, it presumably uses CenP-A just like the other chromosomes?

– *The reason for no track of CENP-A on Y chromosome is no enrichment of ChIP-seq signals based on the antibody Phospho-CENP-A (Ser7) in this study. CenY as the putative centromeric repeats in goat Y chromosome was first found based on the similarity to another CenY in sheep Y chromosome (T2T-sheep1.0), and further supported by the evidence of enrichment in the middle of the Y chromosome, which was in accordance with the karyotype results¹⁸. However, we did not find apparent hypermethylation or ChIP-seq peaks around CenY. We used Phospho-CENP-A (Ser7) antibody to capture the DNA fragments from CENP-A enrichment for sequencing, and this antibody is different from other CENP-A antibodies. Phospho-CENP-A (Ser7) Antibody detects endogenous levels of CENP-A protein only when phosphorylated on Ser7. This antibody does not cross-react with other histone proteins, including Histone H3. And it works well on the autosomes of sheep and goat. However, we found that Phospho-CENP-A (Ser7) Antibody failed to bind the CENP-A of Y chromosome in both sheep and goat. We also repeated our ChIP-seq, and confirmed the failure of this antibody (Phospho-CENP-A (Ser7)) for Y chromosome, as it works well on autosomes and X chromosome.*

The pseudoautosomal regions (PARs) at the coordinate ~7 Mb with a high level of methylation, which is associated with the inhibited expression of genes in PARs, is very close to the putative CenY (~8 Mb) in goat, and the hypermethylation for CenY might be influenced and no sharp hypermethylation peak is found around CenY. In addition, “CenY was not present in any of the autosomes or the X chromosome as shown in sequence search of T2T-goat1.0 and FISH results (Fig.3c).” In summary, CenY is a putative centromeric repeat unit for goat Y chromosome.

We added the description of no ChIP-seq signals based on Phospho-CENP-A (Ser7) in the revision, as shown in the response [3.28] above. Please also see below and the lines 407-410.

“As centromeric repeats (SatI, SatII and SatIII) and ChIP-seq signals based on Phospho-CENP-A (Ser7) antibody were not found on T2T-CHIY1.0, we extended our search for repeat sequences by reference to repeat sequences found on the sheep Y chromosome in an assembly of a sheep genome.”

18. Iannuzzi, L., Di Meo, G.P. & Perucatti, A. G-and R-banded prometaphase karyotypes in goat (*Capra hircus* L.). Caryologia 49, 267-277 (1996).

[3.29] Line 380 a difference of 0.11% higher mapping rate seems pretty small, are these numbers correct? It would seem the ARS1 would be relatively worse?

– *In fact, T2T-goat1.0 increases the mapping rate of both short reads and long reads. We have already described the mapping results in the section of “Assembly improvement of T2T-goat1.0”, as shown below.*

“We investigated the reference bias of read alignment for population genetics analysis. Short read data from previous sequencing efforts focused on global populations of domestic and wild goats was retrieved from public databases (Supplementary Table 5). Compared to ARS1, T2T-goat1.0 enabled confident mapping of short reads in terms of more reads mapped and a lower mismatch and error rate (Supplementary Fig. 10). T2T-goat1.0 improved the number of mapped reads substantially, and >30.91% of the goat samples had >1% more short reads aligned (19.04% with >5% more reads and 1.98% with >10% more reads) than to ARS1. We observed an increased number of MQ0 reads in T2T-goat1.0, which could be attributed to the fact that most of the newly added regions are composed of repetitive sequences, and short reads are not advantageous in aligning to repetitive sequence positions due to their short lengths.”

“Moreover, we sequenced the genomes of five goats from five representative populations to produce long reads (Supplementary Table 1). Like for short reads, more aligned reads and lower mismatch and error rates were observed in the alignment of the long reads against T2T-goat1.0 than against ARS1 (Supplementary Fig. 10b and Supplementary Table 6).”

– *ARS1 shows relatively poorer alignment rates in male goat individuals due to the incomplete assembly of the Y chromosome. However, due to the relatively small size of Y chromosome and ARS1 containing Y contigs, the mapping rate increase is not that big, which is not going to show a significant result. Therefore, we removed this sentence in the revision. To determine the improvement of Y chromosome in T2T-goat1.0 for your reference, we compared the alignments of short reads from three male and three female individuals on the male specific region (MSY), and we found the substantial differences on mapped reads, coverage, and mean depth (see Rtable 1 below).*

Rtable 1 The mapping statistics of MSY region on Y chromosome of T2T-goat1.0. (This table is only for reviewing, and not included in the revision)

Sample	Sex	Number of mapped reads	Coverage	Mean depth
SRR8442978	Female (doe)	5142	1.36319	0.0514688
SRR8442979	Female (doe)	11140	2.07501	0.111814
SRR8442980	Female (doe)	13741	2.73372	0.132851
SRR8442981	Male (buck)	1252711	99.6105	13.5305
SRR8442987	Male (buck)	845812	99.3738	9.13446
SRR8442988	Male (buck)	858371	99.3957	9.26609

*~20X short reads were mapped to the whole genome, and the mean depth is around 10.

– *We added one sentence to describe this result in the revision, as followed below. Please also see lines 418-423.*

“We tested the performance of *T2T-CHIYI.0* as a reference to call variants for a population of 102 bucks (Supplementary Table 5), which yielded a mapping rate as high as 99.95%. The buck and doe showed the significant differences on the mapped short reads onto the male-specific region of the Y chromosome (MSY), with the average depth of aligned reads for buck individuals vs. only a few misaligned reads from doe individuals.”

[3.30] Line 385 why use female goats? This comes right after making a major point about the improvement of the Y chromosome. A brief explanation would be pertinent.

– *Y chromosome is not included in ARS1 genome assembly, and to make a comparison with ARS1, we chose 5 female goats to sequence long reads and call structural variations (SVs) with two genome assemblies T2T-goat1.0 and ARS1 as the references. We focused on autosomes and the X chromosome and showed the improved SV calling performances of T2T-goat1.0. We added the brief explanation to address this point in the revision. Please see below and lines 429-430.*

“We focused on autosomes and chromosome X in female goats for SV calling due to no Y chromosome included in *ARS1*.”

[3.31] Line 388 If reporting the average of five data sets, a range would be useful

– *We added a range for the five data sets. Please see below and lines 432-434.*

“We identified a total of 63,417 SVs in the PacBio long reads of the five individuals (27,163 – 29,561) using *T2T-goat1.0* as a reference (Supplementary Table 11).”

[3.32] Line 416 Should this be “covered” exons or “overlapped” exons? (it could be “covered” is correct, just wanted to check that is accurate)

– *Corrected to “overlapped” exons. Please see line 457.*

[3.33] Line 427 The supplementary table lists references for data used in the manuscript, but these should be included in the reference list not just in a supplementary table.

– *We have added the references in Supplementary Tables 5 and 19 into the reference list in Supplementary Methods.*

– *Please see the method “SNP and SV calling based on short reads” and lines 981-985 in the revision.*

“The whole-genome sequence data of caprine genomes, which included 448 domesticated and 68 wild goat individuals, were downloaded from public databases¹⁹⁻⁴⁰, including NextGen consortium projects (PRJEB3134, PRJEB3135, PRJEB3136, and PRJEB3140) (Supplementary Table 5).”

– *Please see the method “Protein-coding gene annotation” and lines 887-891 in the revision.*

“For the transcriptome-based approach, a total of 93 RNA-seq datasets from 40 goat tissues⁴¹⁻⁵⁹, including 50 ones downloaded from NCBI and 43 newly sequenced ones in this study, were collected (Supplementary Table 1 and Supplementary Table 19), to assemble 145,492 transcripts and obtain 19,437 nonredundant transcripts with Stringtie⁶⁰ (v1.3.4d).”

[3.34] Line 445 This analysis doesn’t seem to have much meaning in my view. It is mainly a function of the distribution of identified goat QTL rather than a reflection of the likelihood that PUR contain tightly linked SNP that would explain more variation than previously identified SNP.

– *Removed.*

[3.35] Line 449 remove “Additionally”. Also, how did these overlap with long read SVs?

– *“Additionally” removed.*

– *“We identified a total of 63,417 SVs in the PacBio long reads of the five individuals (27,163 – 29,561) using T2T-goat1.0 as a reference.” “We identified a total of 32,419 SVs based on short reads”, among which 5979 SVs (18.4%) are also called based on long reads. The two platforms based on short and long reads have different powers to call SVs, and were run in the different size of populations, which explained why only 18.4% SVs are shared. Our manuscript does not focus on the comparison of these two platforms for the performance of SV calling, but we made efforts to assess the reliabilities of SV called based on short reads. We selected top 15 SVs which are shown under selection of cashmere, and found that top 12 SVs based on F_{ST} values are supported by both short and long reads. We added an additional table as Supplementary Table 18 in the revision.*

[3.36] Lines 455-467 It is not clear why it is surprising or important that SNP identified with this reference can be used in the same way as the SNP identified with the old reference? I believe this recapitulates previously published relationships and doesn’t add anything new, at least nothing the authors remark upon. Thus, it would be more briefly summarized without the need for Figure 6c or 6d. This contrasts with the next section about selective sweep definition being influenced by the new assembly which is new.

– *We used T2T-goat1.0 as the reference genome to call SNPs for population structure analysis. Our goal was to demonstrate the reliability of T2T-goat1.0 genome. The results from NJ, PCA, and ADMIXTURE analyses align with those obtained using ARS1 as the reference genome for SNP calling. This consistency confirms the accuracy of T2T-goat1.0. Additionally, the absence of novel findings suggests that the existing SNP loci are sufficient to differentiate population differences. We moved Figure 6c or 6d to Supplementary Fig. 22 in the revision, and summarized the related results in brief,*

as shown below. Please also see lines 490-495.

“SNPs by *T2T-goat1.0* was reliable for elucidating phylogenetic relationships within the genus *Capra*, with similar results obtained based on *ARS1*. Wild and domestic goats showed different linkage disequilibrium (LD) decay rates (Supplementary Fig. 22), and the analysis of neighbor-joining (NJ) tree, principal component analysis (PCA) and ADMIXTURE distinguished five subgroups in domesticated goats based on the geographical origin, namely, Europe, Africa, East Asia, South Asia, and the Middle East (Supplementary Fig. 22).”

[3.37] Line 457 NJ tree wasn't “used to divide” rather “A neighbor-joining (NJ) tree distinguished two major branches corresponding to wild and domestic goats”.

– *We have modified the sentence in the revised manuscript, according to your suggestion. We updated the manuscript with brief description related to NJ tree, as shown in the response above.*

[3.38] Line 464 I think it would be clearer to state “LD decay rate in wild goats was slower relative to domestic goats, with the East Asian...”

– *We have updated the sentence with brief description according to your suggestion. We updated the manuscript with brief description related to NJ tree, as shown in the response [3.36] above.*

[3.39] Line 474 “commonly found in ARS1” should be “found in common with ARS1”

– *Done. Please see lines 503-504.*

[3.40] Line 475 “previously reported genes” should have references

– *We have added references in the revised manuscript. Please see line 505.*

[3.41] Line 482 why is this remarkable? Suggest just removing that word and stating that “the presence of selective signal was supported by overlapping signals by the π ratio statistic.

– *We have updated this sentence according to your suggestion. Please see below and lines 511-512.*

“The presence of selective signal was supported by overlapping signals by the π ratio statistic (Fig. 5d).”

[3.42] Line 492 “investigated the domestication of SVs”, how does one domesticate an SV? Maybe this should be “We further investigated SVs found in domestication-associated genomic regions in bezoars and domesticated goat populations.”

– *Revised as suggested. Please see lines 521-522.*

[3.43] Line 496 The list of genes is far shorter than 63, so maybe should say “including”

or “for example” for the genes that are selected to be presented. Otherwise, why are these particular genes shown?

– *Corrected to “For example”. Please see below and line 525-528.*

“These 63 genes were associated with phenotypic characteristics, for example, immune activity (*MUC6*, *GIMAP8*, *CNTNAP2*, *KCND3*, and *NBEAL1*), stress resistance (*ABCC4*, *PSIP1*, and *TOMM70*), and sperm flagellar function (*SPAG16*), and were classified into three groups.”

[3.44] Line 500 not clear why one would fail to detect F_{ST} when mapping to the new assembly vs ARS1 for “group 2” genes? Apparently the genes are present in ARS1 so why would using that as reference fail to identify them?

– *For the domestication selection, selective signals not found on ARS1 are located in the PUR region. Due to missing or poor assembly quality on ARS1, these new selective sites cannot be detected with ARS1 as reference. Meanwhile, top 1% selective signals were used to show the sites under strong selection. With T2T-goat1.0 as reference, more variations had been discovered, predominantly due to the presence of PURs and lower error rates, for example, 25,397,794 high-quality SNPs by T2T-goat1.0 vs. 24,238,138 SNPs by ARS1. F_{ST} values were calculated in 50-kb sliding window. Many PURs, especially the PURs without genes, are not under selection and results into lower F_{ST} values calculated based on novel SNPs. Addition of more signals with lower F_{ST} values makes some genes in non-PURs (i.e., “group 2” genes) passing through the cutoff of top 1% F_{ST} values. Therefore, more SNPs with T2T-goat1.0 as reference make the selection analysis more accurate, with not only selected genes in PURs but also selected “group 2” genes in non-PURs.*

[3.45] Line 517 mildly odd that the number of selective signals is nearly identical to those for domestication, a bit surprising I guess (no change needed, just noting this)

– *In the analysis of domestication and trait selection, we used the same window size (50 kb) and step sizes (10kb). Also, the top 1% of F_{ST} and XP-CLR values were considered as putative selective sweeps associated with domestication and cashmere traits, based on the similar SNPs and SVs, resulting in a roughly similar number of selection signals.*

[3.46] Line 524 It is not clear why one would find so much genomic region in selective sweep only when using ARS1 as reference, if the hypothesis is that regions missing in that assembly are a primary source of novel selection detection. If the analysis is this sensitive to reference genome, one wonders how reliable any of the identified regions might be? The results suggest that selective sweep analysis may not be completely substituting for a GWAS in a population segregating the trait, even if previously identified genes fall in some of those regions.

– *The PUR regions constitute only a small portion of the entire genome, with most genomic regions being well-assembled in ARS1. Therefore, we reported the improvement of T2T-goat1.0 on cashmere selection, with 274 novel selection regions*

and 72 genes unveiled in PURs, among which some genes are significantly associated with cashmere trait, for example, ABCC4 gene (IMCG12g00092). We observed the improvement of T2T-goat1.0 as a reference, and identified a large portion of selection signals in common between the two references, on this selective sweep analysis. Top 1% XP-CLR scores is selected as a cutoff to determine the selected regions across the chromosomes. The addition of new SNPs and corrected SNPs with T2T-goat1.0 as a reference reorganized the selected regions for the calculation of XP-CLR scores, and placed some unselected regions by ARS1 into the category of the selected ones. However, the signals between T2T-goat1.0 and ARS1 are reliable and match well in the newly added Supplementary Fig. 25b.

Supplementary Fig. 25b Selective signals between cashmere and non-cashmere goats based on top 1% XP-CLR values are compared across the chromosomes between with ARS1 and T2T-goat1.0 as references.

– In contrast, genome wide association study (GWAS) is another widely used approach to compare the genomes from many different individuals to find genetic markers associated with a particular phenotype. GWAS and selective sweep are applied in the different circumstances, and selective sweep analysis cannot be completely substituting for a GWAS in a population segregating the trait. If the cashmere trait data was collected for each individual in a population with segregation, GWAS might be a good strategy to discover cashmere-related genes, but in this study, our selective sweep might be better to discover the selective genomic regions.

[3.47] Line 531 allele frequency differences between cashmere/non-cashmere

populations can be breed effects not necessarily indicative of association with the trait, as population stratification is inherent in the approach taken.

– *Good comment again! We agreed that the selected genes could be from the genetic basis between cashmere and non-cashmere populations, i.e., breed effects, which indicated that they are not necessarily associated with the cashmere trait. As mentioned in the response [3.7] above, selective sweep is a method to search for the fixed or selected alleles that might be genetic hitchhiking, in a population against the other one. It depends on the divergence in allele frequency between two populations genetically or geographically isolated. A total of 2692 selective signals covering 1440 genes were determined based on the top 1% cutoff, based on a total of 25,397,794 SNPs by T2T-goat1.0. Among them, 274 selective signals covering 72 genes were located in the previously unresolved regions (PURs) that are not included in those by ARS1, and added by using T2T-goat1.0 as reference. Based on these selective regions, it enabled us to pinpoint the candidate genes that are with significant different alleles between cashmere and non-cashmere goats. However, these selective regions or genes do not necessarily represent an association with the cashmere trait, and they might be from the genetic background of these two goat populations, which is a result of adaption to their environments, or partially with contribution of genetic shift.*

These selected genomic regions with allele differences between these two populations are a result of the adaption to environment and selection, rather than the association with cashmere trait in GWAS. However, selective sweep is an effective method to search for the selective regions and genes, which has been used widely in the previous studies¹⁵. Many selective regions and genes are indeed associated with cashmere trait, such as, the KRT genes on both CHI5 and CHI19, and the FGF genes on both CHI5 and CHI20. And our candidate genes could be very useful for further studies on cashmere selection, especially those newly ones in PURs.

[3.48] Line 554 neither the cattle nor gerbil genomes referenced are complete seamless mammal genomes. The gerbil assembly has complete centromeres but was not purported to be T2T. There are some non-human primate assemblies that are T2T status in the NCBI database but so far no publications that I am aware of. I believe that a mouse T2T assembly has been published but didn't have time to search for the reference. Might be safer to say that this assembly is one of the first mammal genomes to achieve complete, seamless assembly following the first human T2T assembly and use the Science reference only. For example, there are plenty of assemblies of cattle published that are as good as the one mentioned in reference 45, also many other species have assemblies of similar quality.

– *We updated the sentence in the revised manuscript. Please see below and lines 590-591.*

“To date, T2T-goat1.0 is one of the first mammal genomes to achieve complete, seamless assembly following the first human T2T assembly⁵.”

[3.49] Line 569 I don't understand this statement since none of the three references report T2T genome assemblies.

– *We removed the references to avoid the misunderstanding in the revised manuscript. Please see below and lines 604-606.*

“The conservation of centromeric satellites could be further investigated on acrocentric chromosomes in the T2T genome assemblies of Bovidae species, for example, cattle, sheep, and deer.”

[3.50] Line 571 It isn't clear why the authors originally used a merged assembly method when parental sequence was available. In general of late, the data types reportedly used would likely have assembled the Y chromosome just using the trio approach in hifiasm that they employed in merged mode instead of trio mode; certainly likely would have resolved the PAR regions of X and Y chromosome.

– *Due to the high repetitiveness and homologous regions (PARs) with the X chromosome (i.e., PARs), the complete assembly of the Y chromosome is extremely challenging. The trio approach is more powerful to assembly Y chromosome⁶¹, for example, ChrY in humans⁸. “The Y chromosome was assembled using a trio-binning strategy and hash tables of *k*-mers from the parents' short reads”, in T2T-goat1.0, although autosomes in T2T-goat1.0 were assembled based on a merged assembly method. The method for Y chromosome assembly is shown as follows.*

“Y-chromosome assembly

The Y chromosome was assembled using a trio-binning strategy and hash tables of *k*-mers from the parents' short reads. In brief, 21-mer libraries were constructed from MGI short reads (~120× coverage) for the buck individual for the T2T genome assembly and its parents using the Jellyfish⁶² (v2.3.0). Furthermore, the paternal and maternal unique 21-mers were identified if a 21-mer was found only in either the father or mother, respectively, and had a sequence depth of >3 and <120 (the average depth across the whole genome) in the T2T buck individual. The ultralong ONT reads were chosen to determine paternal and maternal origination with more paternal and maternal 21-mers respectively. The paternal ONT reads sequenced from the autosomes were subsequently removed out to retrieve those potentially from the paternal X and Y chromosomes, which were used to construct the assembly graph of the Y chromosome using the Nextgraph module in NextDenovo (v2.5.2). The Y chromosome assembly was further improved by filling gaps and correcting possible assembly errors according to the Y contigs by the trio-binning model of HiFiasm (v0.16.1-r375) based on HiFi reads and 31-mers from the T2T buck parents. The Y chromosome was combined with all the autosomes and the X chromosome to obtain the genome assembly GV5.”

– *We added more discussions to address the trio assembly for Y chromosome in animals, as shown below. Please also see lines 608-613.*

“Due to the high repetitiveness and homologous regions with the X chromosome (i.e.,

PARs), the complete assembly of the Y chromosome is extremely challenging. Since the trio-binning method with parental sequencing data was reported for haplotype-resolved assembly⁶¹, Y-chromosome assemblies turned to combining long reads and parental reads. The trio-binned reads enabled the T2T assembly of Y chromosomes, for example, Y-chromosome assemblies of human^{8,63} and apes⁶⁴.”

[3.51] Line 581 I think this may be a misstatement, it is unlikely that there aren't multicopy protein-coding gene families elsewhere in the genome, perhaps this was meant to say “found on the Y chromosome”? Certainly the immune gene clusters would count as multicopy protein coding gene families for example.

– *What we meant to say is that no other multicopy protein-coding gene families were found on the Y chromosome. We have revised the sentence accordingly. Please below and lines 621-622.*

“No other multicopy protein-coding gene families were identified on the Y chromosome in *T2T-goat1.0*.”

Methods :

[3.52] Line 643 “CTAB method” needs reference.

– *Reference added, please see below and line 684.*

“High-molecular-weight genomic DNA was extracted based on the CTAB method⁶⁵”.

[3.53] Line 644 “manufacturer's”

– *Revised. Please see line 685.*

[3.54] Line 991-993 The BioProject indicates “will be available on 2026-01-14”

– *We have released all the genome assemblies and raw data already for the reviewing of the revised manuscript. Please see Data Availability for the accession numbers.*

[3.55] Line 997 the link for software at github returns “page not found”

– *Accessible now.*

1. Chen, J. et al. A complete telomere-to-telomere assembly of the maize genome. *Nat. Genet.* **55**, 1221-1231 (2023).
2. Hu, J., Fan, J., Sun, Z. & Liu, S. NextPolish: a fast and efficient genome polishing tool for long-read assembly. *Bioinformatics* **36**, 2253-2255 (2020).
3. Hu, J. et al. NextPolish2: a repeat-aware polishing tool for genomes assembled using HiFi long reads. *bioRxiv*, 2023.04. 26.538352 (2023).
4. Yang, C. et al. The complete and fully-phased diploid genome of a male Han Chinese. *Cell Res.* **33**, 745–761 (2023).

5. Nurk, S. et al. The complete sequence of a human genome. *Science* **376**, 44-53 (2022).
6. Vollger, M.R. et al. Segmental duplications and their variation in a complete human genome. *Science* **376**, eabj6965 (2022).
7. Išerić, H., Alkan, C., Hach, F. & Numanagić, I. Fast characterization of segmental duplication structure in multiple genome assemblies. *Algorithms Mol. Biol.* **17**, 4 (2022).
8. Rhie, A. et al. The complete sequence of a human Y chromosome. *Nature* **621**, 344-360 (2023).
9. Hamilton, C. et al. Copy number variation of testis-specific protein, Y-encoded (TSPY) in 14 different breeds of cattle (*Bos taurus*). *Sex. Dev.* **3**, 205-213 (2009).
10. Yue, X.-P. et al. Copy number variations of the extensively amplified Y-linked genes, HSFY and ZNF280BY, in cattle and their association with male reproductive traits in Holstein bulls. *BMC Genomics* **15**, 1-12 (2014).
11. Wilkerson, A.J.P. et al. Gene discovery and comparative analysis of X-degenerate genes from the domestic cat Y chromosome. *Genomics* **92**, 329-338 (2008).
12. Li, R. et al. A Hu sheep genome with the first ovine Y chromosome reveal introgression history after sheep domestication. *Sci. China Life Sci.* **64**, 1116-1130 (2021).
13. Bickhart, D.M. et al. Single-molecule sequencing and chromatin conformation capture enable *de novo* reference assembly of the domestic goat genome. *Nat. Genet.* **49**, 643-650 (2017).
14. Zheng, Z. et al. The origin of domestication genes in goats. *Sci. Adv.* **6**, eaaz5216 (2020).
15. Cai, Y. et al. Ancient genomes reveal the evolutionary history and origin of cashmere-producing goats in China. *Mol. Biol. Evol.* **37**, 2099-2109 (2020).
16. Aganezov, S. et al. A complete reference genome improves analysis of human genetic variation. *Science* **376**, eabl3533 (2022).
17. Luo, L.-Y. et al. Telomere-to-telomere sheep genome assembly reveals new variants associated with wool fineness trait. *bioRxiv* (2024).
18. Iannuzzi, L., Di Meo, G.P. & Perucatti, A. G-and R-banded prometaphase karyotypes in goat (*Capra hircus* L.). *Caryologia* **49**, 267-277 (1996).
19. Chen, Q.M. et al. Identification of genomic characteristics and selective signals in a Du'an goat flock. *Animals* **10**, 994 (2020).
20. Chebii, V.J. et al. Genome-wide analysis of Nubian Ibex reveals candidate positively selected genes that contribute to its adaptation to the desert environment. *Animals (Basel)* **10**, 2181 (2020).
21. Guo, J.Z. et al. Identification and population genetic analyses of copy number variations in six domestic goat breeds and Bezoar ibexes using next-generation sequencing. *BMC Genomics* **21**, 840 (2020).
22. Chowdhury, S.M.Z.H. et al. Whole genome analysis of Black Bengal goat from Savar Goat Farm, Bangladesh. *BMC Res. Notes* **12**, 687 (2019).
23. Siddiki, A.Z. et al. The genome of the Black Bengal goat (*Capra hircus*). *BMC Res. Notes* **12**, 362 (2019).
24. Mollah, M. et al. Whole genome sequence and genome-wide distributed single nucleotide polymorphisms (SNPs) of the Black Bengal goat. *F1000Res.* **8**(2019).
25. Bhat, B. et al. Changthangi Pashmina goat genome: sequencing, assembly, and annotation. *Front. Genet.* **12**, 695178 (2021).
26. Cao, Y.H. et al. Genetic basis of phenotypic differences between Chinese Yunling black goats and Nubian goats revealed by allele-specific expression in their F1 hybrids. *Front. Genet.*

- 10**(2019).
27. Gao, J. et al. Genomic characteristics and selection signatures in indigenous Chongming White goat (*Capra hircus*). *Front. Genet.* **11**, 901 (2020).
 28. Kim, J.Y. et al. Discovery of genomic characteristics and selection signatures in Korean indigenous goats through comparison of 10 goat breeds. *Front. Genet.* **10**, 699 (2019).
 29. Wang, J.J. et al. Genomic signatures of selection associated with litter size trait in Jining Gray goat. *Front. Genet.* **11**, 286 (2020).
 30. Signer-Hasler, H. et al. Runs of homozygosity in Swiss goats reveal genetic changes associated with domestication and modern selection. *Genet. Sel. Evol.* **54**, 6 (2022).
 31. Tang, Q.Z. et al. Comparative transcriptomics of 5 high-altitude vertebrates and their low-altitude relatives. *Gigascience* **6**, gix105 (2017).
 32. Zhang, B. et al. Genome-wide definition of selective sweeps reveals molecular evidence of trait-driven domestication among elite goat (*Capra* species) breeds for the production of dairy, cashmere, and meat. *Gigascience* **7**, giy105 (2018).
 33. Wu, D.D. et al. Convergent genomic signatures of high-altitude adaptation among domestic mammals. *Natal. Sci. Rev.* **7**, 952-963 (2020).
 34. Grossen, C., Guillaume, F., Keller, L.F. & Croll, D. Purging of highly deleterious mutations through severe bottlenecks in Alpine ibex. *Nat. Commun.* **11**, 1001 (2020).
 35. Henkel, J. et al. Selection signatures in goats reveal copy number variants underlying breed-defining coat color phenotypes. *PLoS Genet.* **15**, e1008536 (2019).
 36. Chen, L. et al. Large-scale ruminant genome sequencing provides insights into their evolution and distinct traits. *Science* **364**, eaav6202 (2019).
 37. Daly, K.G. et al. Ancient goat genomes reveal mosaic domestication in the Fertile Crescent. *Science* **361**, 85-87 (2018).
 38. Zheng, Z.Q. et al. The origin of domestication genes in goats. *Sci. Adv.* **6**, eaaz5216 (2020).
 39. Lai, F.N. et al. Whole-genome scanning for the litter size trait associated genes and SNPs under selection in dairy goat (*Capra hircus*). *Sci. Rep.* **6**, 38096 (2016).
 40. Menzi, F. et al. Genomic amplification of the caprine *EDNRA* locus might lead to a dose dependent loss of pigmentation. *Sci. Rep.* **6**, 28438 (2016).
 41. Fushan, A.A. et al. Gene expression defines natural changes in mammalian lifespan. *Aging Cell* **14**, 352-365 (2015).
 42. Chen, H.L. et al. Alterations of mRNA and lncRNA profiles associated with the extracellular matrix and spermatogenesis in goats. *Anim. Biosci.* **35**, 544-555 (2022).
 43. Yang, C. et al. Effects of dietary *Macleaya cordata* extract inclusion on transcriptomes and inflammatory response in the lower gut of early weaned goats. *Anim. Feed Sci. Technol.* **272**, 114792 (2021).
 44. Su, R. et al. Screening the key genes of hair follicle growth cycle in Inner Mongolian Cashmere goat based on RNA sequencing. *Arch. Anim. Breed.* **63**, 155-164 (2020).
 45. Frattini, S. et al. Genome-wide analysis of DNA methylation in hypothalamus and ovary of *Capra hircus*. *BMC Genomics* **18**, 476 (2017).
 46. Nocelli, C. et al. Shedding light on cashmere goat hair follicle biology: from morphology analyses to transcriptomic landscape. *BMC Genomics* **21**, 458 (2020).
 47. Zhang, J.P., Deng, C.C., Li, J.L. & Zhao, Y.J. Transcriptome-based selection and validation of optimal house-keeping genes for skin research in goats (*Capra hircus*). *BMC Genomics* **21**, 493

- (2020).
48. Su, F. et al. Genome-wide analysis on the landscape of transcriptomes and their relationship with DNA methylomes in the hypothalamus reveals genes related to sexual precocity in Jining Gray Goats. *Front. Endocrinol.* **9**, 501 (2018).
 49. Ling, Y.H. et al. Identification of lncRNAs by RNA sequencing analysis during *in vivo* pre-implantation developmental transformation in the goat. *Front. Genet.* **10**, 1040 (2019).
 50. Luigi-Sierra, M.G. et al. A protein-coding gene expression atlas from the brain of pregnant and non-pregnant goats. *Front. Genet.* **14**, 1114749 (2023).
 51. He, C.S. et al. Integrative analysis of lncRNA-miRNA-mRNA regulatory network reveals the key lncRNAs implicated potentially in the differentiation of adipocyte in goats. *Front. Physiol.* **13**, 900179 (2022).
 52. Sun, Z.P. et al. Characterization of circular RNA profiles of oviduct reveal the potential mechanism in prolificacy trait of goat in the estrus cycle. *Front. Physiol.* **13**, 990691 (2022).
 53. Shen, J.Y. et al. Integrated transcriptome analysis reveals roles of long non-coding RNAs (lncRNAs) in caprine skeletal muscle mass and meat quality. *Funct. Integr. Genom.* **23**, 63 (2023).
 54. Tian, H.B. et al. Knockout of stearoyl-CoA desaturase 1 decreased milk fat and unsaturated fatty acid contents of the goat model generated by CRISPR/Cas9. *J. Agric. Food Chem.* **70**, 4030-4043 (2022).
 55. Guan, D.L. et al. Analyzing the genomic and transcriptomic architecture of milk traits in Murciano-Granadina goats. *J. Anim. Sci. Biotechnol.* **11**, 35 (2020).
 56. Dong, Y. et al. Sequencing and automated whole-genome optical mapping of the genome of a domestic goat (*Capra hircus*). *Nat. Biotechnol.* **31**, 135-141 (2013).
 57. Wang, A.L. et al. Transcriptome analysis reveals potential immune function-related regulatory genes/pathways of female Lubo goat submandibular glands at different developmental stages. *PeerJ* **8**, e9947 (2020).
 58. Xu, T.S. et al. Landscape of alternative splicing in *Capra_hircus*. *Sci. Rep.* **8**, 15128 (2018).
 59. Zhang, Y.J. et al. Transcriptome profiling reveals transcriptional and alternative splicing regulation in the early embryonic development of hair follicles in the cashmere goat. *Sci. Rep.* **9**, 17735 (2019).
 60. Shumate, A., Wong, B., Pertea, G. & Pertea, M. Improved transcriptome assembly using a hybrid of long and short reads with StringTie. *PLoS Comput. Biol.* **18**, e1009730 (2022).
 61. Koren, S. et al. *De novo* assembly of haplotype-resolved genomes with trio binning. *Nat. Biotechnol.* **36**, 1174-1182 (2018).
 62. Marçais, G. & Kingsford, C. A fast, lock-free approach for efficient parallel counting of occurrences of *k*-mers. *Bioinformatics* **27**, 764-770 (2011).
 63. He, Y. et al. T2T-YAO: A Telomere-to-Telomere assembled diploid reference genome for Han Chinese. *Genom. Proteom. Bioinf.* **21**, 1085-1100 (2023).
 64. Makova, K.D. et al. The complete sequence and comparative analysis of ape sex chromosomes. *Nature* **630**, 401-411 (2024).
 65. Guha, P., Das, A., Dutta, S. & Chaudhuri, T.K. A rapid and efficient DNA extraction protocol from fresh and frozen human blood samples. *J. Clin. Laboratory Anal.* **32**, e22181 (2018).

RESPONSE TO REVIEWERS' COMMENTS

Reviewer #2 (Remarks to the Author):

The explanations are relatively complete, and the entire process is clearer and more credible. It is foreseeable that this genome assembly will become an important reference sequence in goat molecular research. Additionally, I apologize for mistakenly writing "tandem duplication" as "random duplication" in the previous round of review, which might have caused confusion. However, you provided a convincing response. Overall, this is a rigorous and meaningful work.

– Thank you very much for all your suggestions and comments. The "tandem duplication" is used in the revision.

Reviewer #3 (Remarks to the Author):

The authors have responded appropriately to all reviewer comments in my view. The few rebuttals appear to represent differences of opinion not problems with the manuscript, e.g. the value of selective sweep versus GWAS analysis. The manuscript appears better focused and more readable with the reduction in figures although there are still 30 figure panels in the main body.

– Thank you very much for all your suggestions and comments.

Reviewer #3 (Remarks on code availability):

I checked that the code is available and briefly reviewed the scripts. The authors do not really intend for it to be reused as there are various means to achieve the same thing and they aren't reporting their pipeline just the results. The actual software (mappers, variant detectors, etc) seems appropriate but I did not attempt to make it run or anything like that.

– Thank you for your review. We developed a genome polish pipeline, and released it in GitHub and Zenodo. The DOI for Zenodo has been cited in the revision. We tested the pipeline running, and it could run smoothly. In addition to the polish pipeline, we also provided more codes to show the assembly and annotation of our T2T-goat1.0, including assembly, annotation, centromere identification, genome assessment, etc. These scripts may serve as a reference to make use of them, together with our detailed descriptions in Supplementary Methods.

To be reused conveniently, we added more details and information in GitHub (<https://github.com/Wuhui2024/CAU-T2T-Goat>), and the codes are placed in the eight directories, including assembly, polish pipeline, annotation, genome assessment, centromere identification, SV calling based on long reads, SNP calling based on short reads, and population structure and selection analysis. The whole pipeline is summarized in a new figure in GitHub, as shown below.